# Nash Regret Guarantees for Linear Bandits

**Ayush Sawarni**
Indian Institute of Science
Bangalore
sawarniayush@gmail.com

**Soumyabrata Pal**
Google Research
Bangalore
soumyabrata@google.com

**Siddharth Barman**
Indian Institute of Science
Bangalore
barman@iisc.ac.in

## Abstract

We obtain essentially tight upper bounds for a strengthened notion of regret in the stochastic linear bandits framework. The strengthening—referred to as Nash regret—is defined as the difference between the (a priori unknown) optimum and the geometric mean of expected rewards accumulated by the linear bandit algorithm. Since the geometric mean corresponds to the well-studied Nash social welfare (NSW) function, this formulation quantifies the performance of a bandit algorithm as the collective welfare it generates across rounds. NSW is known to satisfy fairness axioms and, hence, an upper bound on Nash regret provides a principled fairness guarantee.

We consider the stochastic linear bandits problem over a horizon of $\mathsf{T}$ rounds and with set of arms $\mathcal{X}$ in ambient dimension $d$. Furthermore, we focus on settings in which the stochastic reward—associated with each arm in $\mathcal{X}$—is a non-negative, $\nu$-sub-Poisson random variable. For this setting, we develop an algorithm that achieves a Nash regret of $O\left(\sqrt{\frac{d\nu}{\mathsf{T}}}\log(\mathsf{T}|\mathcal{X}|)\right)$. In addition, addressing linear bandit instances in which the set of arms $\mathcal{X}$ is not necessarily finite, we obtain a Nash regret upper bound of $O\left(\frac{d^{\frac{5}{4}}\nu^{\frac{1}{2}}}{\sqrt{\mathsf{T}}}\log(\mathsf{T})\right)$. Since bounded random variables are sub-Poisson, these results hold for bounded, positive rewards. Our linear bandit algorithm is built upon the successive elimination method with novel technical insights, including tailored concentration bounds and the use of sampling via John ellipsoid in conjunction with the Kiefer-Wolfowitz optimal design.

## 1 Introduction

Bandit optimization is a prominent framework for sequential decision making and has several applications across multiple domains, such as healthcare [26, 23, 27] and advertising [22]. In this framework, we have a set of arms (possible actions) with unknown means and a time horizon. The goal is to sequentially pull the arms such that the regret—which is a notion of loss defined over the bandit instance—is minimized.

We consider settings wherein the stochastic rewards generated by a sequential algorithm induces welfare across a population of agents. Specifically, there are $\mathsf{T}$ agents, arriving one per round; in particular, the reward accrued at each round $t \in [\mathsf{T}]$ corresponds to the value accrued by the $t^{\text{th}}$ agent. Indeed, such a welfarist connection exists in various applications of the bandit framework. Consider, for instance, the classic context of drug trials [24]: Suppose there are $\mathsf{T}$ patients and several available drugs. In each round $t \in [\mathsf{T}]$, one of the available drugs is administered to the $t^{\text{th}}$ patient. Subsequently, the reward accrued at the $t^{\text{th}}$ round corresponds to the efficacy of the administered drug to the $t^{\text{th}}$ patient. In such a setting, fairness is a fundamental consideration. That is, in addition to cumulative efficacy, individual effectiveness of the drugs is quite important.

37th Conference on Neural Information Processing Systems (NeurIPS 2023).

A central notion in the bandit literature is that of average regret, defined as the difference between the (a priori unknown) optimum and the arithmetic mean of the expected rewards (accumulated by the algorithm) [4]. However, average regret fails to capture the fairness criterion that the rewards should be balanced (across the agents) and not just cumulatively high. From a welfarist viewpoint, the standard notion of (average) regret equates the algorithm's performance to the social welfare it induces. Social welfare is defined as the sum of agent's rewards [19] and can be high among a set of agents even if a fraction of them receive indiscriminately low rewards. For instance, in the drug-trials example provided above, high average efficacy (i.e., high social welfare) does not rule out a severely ineffective outcome for a subset of agents.

Given that average regret is defined using the sum of expected rewards, this notion inherits this utilitarian limitation of social welfare. In summary, in welfare-inducing contexts, a bandit algorithm with low average regret is not guaranteed to induce fair outcomes across rounds.

Addressing this issue and with the overarching aim of achieving fairness across rounds (i.e., across agents that receive rewards from a bandit algorithm), the current work considers a strengthened notion of regret. The strengthening—referred to as Nash regret—is defined as the difference between the (a priori unknown) optimum and the geometric mean of expected rewards induced by the bandit algorithm. It is relevant to note that the geometric mean (of rewards) corresponds to the Nash social welfare (NSW) function [19]. This welfare function has been extensively studied in mathematical economics (see, e.g., [19]) and is known to satisfy fundamental fairness axioms, including the Pigou-Dalton transfer principle, scale invariance, and independence of unconcerned agents. Hence, by definition, Nash regret quantifies the performance of a bandit algorithm as the NSW it generates.

Quantitatively speaking, in order for the geometric mean (i.e., the NSW) to be large, the expected reward at every round should be large enough. The AM-GM inequality also highlights that Nash regret is a more demanding objective that average regret.

We obtain novel results for Nash regret in the stochastic linear bandits framework. In this well-studied bandit setup each arm corresponds to a $d$-dimensional vector $x$ (an arm-specific context) and the unknown arm means are modelled to be a linear function of $x$. With a focus on average regret, stochastic linear bandits have been extensively studied in the past decade [1, 9, 21]. The current paper extends the line of work on linear bandits with fairness and welfare considerations.

Note that an ostensible approach for minimizing Nash regret is to take the logarithm of the observed rewards and, then, solve the average regret problem. However, this approach has the following shortcomings: (i) Taking log implies the modified rewards can have a very large range possibly making the regret vacuous, and (ii) This approach leads to a multiplicative guarantee and not an additive one. In a recent work of [5], the authors study Nash regret in the context of stochastic multi-armed bandits (with bounded rewards) and provide optimal guarantees. The current work notably generalizes this prior work to linear bandits.

## 1.1 Our Contributions and Techniques

We consider the stochastic linear bandits setting with a set of arms $\mathcal{X}$ over a finite horizon of $\mathsf{T}$ rounds. Since we consider the welfarist viewpoint, we assume that the rewards across all the rounds are positive and, in particular, model the distribution of the arm rewards to be $\nu$-sub-Poisson, for parameter $\nu \in \mathbb{R}_+$. Our goal is to minimize the Nash regret $\mathrm{NR_T}$.

We develop a novel algorithm LINNASH that obtains essentially optimal Nash regret guarantees for this setting. Specifically, for a finite set of arms $\mathcal{X} \subset \mathbb{R}^d$, our algorithm LINNASH achieves Nash regret $\mathrm{NR_T} = O\left(\sqrt{\frac{d\nu}{\mathsf{T}}} \log(\mathsf{T}|\mathcal{X}|)\right)$. For infinite sets of arms, a modified version of LINNASH achieves Nash regret $\mathrm{NR_T} = O\left(\frac{d^{\frac{5}{4}}\nu^{\frac{1}{2}}}{\sqrt{\mathsf{T}}} \log(\mathsf{T})\right)$.

Recall that Nash regret is a strengthening of the average regret; the AM-GM inequality implies that, for any bandit algorithm, the Nash regret is at least as much as its average regret. Hence, in the linear bandits context, the known $\Omega\left(d/\sqrt{\mathsf{T}}\right)$ lower bound on average regret (see [17], Chapter 24) holds

for Nash regret as well.[1] This observation implies that, up to a logarithmic factor, our upper bound on Nash regret is tight with respect to the number of rounds $\mathsf{T}$. We also note that for instances in which the number of arms $|\mathcal{X}| = \omega(2^d)$, the Nash-regret dependence on $d$ has a slight gap. Tightening this gap is an interesting direction of future work.

We note that bounded, positive random variables are sub-Poisson (Lemma 1). Hence, our results hold for linear bandit instances wherein the stochastic rewards are bounded and positive. This observation also highlights the fact that the current work is a generalization of the result obtained in [5]. In addition, notice that, by definition, Poisson distributions are 1-sub-Poisson. Hence, our guarantees further hold of rewards that are not necessarily sub-Gaussian. Given the recent interest in obtaining regret guarantees beyond sub-Gaussian rewards [18, 3], our study of sub-Poisson rewards is interesting in its own right.[2]

Our linear bandit algorithm, LINNASH, has two parts. In the first part, we develop a novel approach of sampling arms such that in expectation the reward obtained is a linear function of the center of John Ellipsoid [13]. Such a strategy ensures that the expected reward in any round of the first part is sufficiently large. The second part of LINNASH runs in phases of exponentially increasing length. In each phase, we sample arms according to a distribution that is obtained as a solution of a concave optimization problem, known as D-optimal design. We construct confidence intervals at each phase and eliminate sub-optimal arms. A key novelty in our algorithm and analysis is the use of confidence widths that are estimate dependent. We define these widths considering multiplicative forms of concentration bounds and crucially utilize the sub-Poisson property of the rewards. The tail bounds we develop might be of independent interest.

## 1.2 Other Related Work

There has been a recent surge in interest to achieve fairness guarantees in the context of multi-armed bandits; see, e.g., [14, 7, 20, 6, 12]. However, these works mostly consider fairness across arms and, in particular, impose fairness constraints that require each arm to be pulled a pre-specified fraction of times. By contrast, our work considers fairness across rounds.

*Alternative Regret Formulations.* In the current work, for the welfare computation, each agent $t$'s value is considered as the expected reward in round $t$. One can formulate stronger notions of regret by, say, considering the expectation of the geometric mean of the rewards, rather than the geometric mean of the expectations. However, as discussed in [5], it is not possible to obtain non-trivial guarantees for such reformulations in general: every arm must be pulled at least once. Hence, if one considers the realized rewards (and not their expectations), even a single pull of a zero-reward arm will render the geometric mean zero.

## 2 Problem Formulation and Preliminaries

We will write $[m]$ to denote the set $\{1, 2, \ldots, m\}$. For a matrix $\mathbf{X}$, let $\mathsf{Det}(\mathbf{X})$ to denote the determinant of $\mathbf{X}$. For any discrete probability distribution $\lambda$ with sample space $\Omega$, write $\mathsf{Supp}(\lambda) \triangleq \{x \in \Omega : \Pr_{X \sim \lambda}\{X = x\} > 0\}$ to denote the points for which the probability mass assigned by $\lambda$ is positive. For a vector $\mathbf{a} \in \mathbb{R}^d$ and a positive definite matrix $\mathbf{V} \in \mathbb{R}^{d \times d}$, we will denote $||a||_{\mathbf{V}} := \sqrt{a^T \mathbf{V} a}$. Finally, let $\mathcal{B} := \{x \in \mathbb{R}^d \mid ||x||_2 = 1\}$ be the $d$-dimensional unit ball.

We address the problem of stochastic linear bandits with a time horizon of $\mathsf{T} \in \mathbb{Z}_+$ rounds. Here, an online algorithm (decision maker) is given a set of arms $\mathcal{X} \subset \mathbb{R}^d$. Each arm corresponds to a $d$-dimensional vector. Furthermore, associated with each arm $x \in \mathcal{X}$, we have a stochastic reward $r_x \in \mathbb{R}_+$. In the linear bandits framework, the expected value of the reward $r_x$ is modeled to be a linear function of $x \in \mathbb{R}^d$. In particular, there exists an unknown parameter vector $\theta^* \in \mathbb{R}^d$ such that, for each $x \in \mathcal{X}$, the associated reward's expected value $\mathbb{E}[r_x] = \langle x, \theta^* \rangle$. Given the focus on welfare contexts, we will, throughout, assume that the rewards are positive, $r_x > 0$, for all $x \in \mathcal{X}$.

The online algorithm (possibly randomized) must sequentially select an arm $X_t$ in each round $t \in [\mathsf{T}]$ and, then, it observes the corresponding (stochastic) reward $r_{X_t} > 0$.[3] For notational convenience,

---

[1]This lower bound on average regret is obtained for instances in which the set of arms $\mathcal{X}$ are the corners of a hypercube [17].

[2]An intersection between sub-Gaussian and sub-Poisson distribution classes is identified in Lemma 2.

[3]Note that, for a randomized online algorithm, the selected arm $X_t$ is a random variable.

we will write $r_t$ to denote $r_{X_t}$. In particular, if in round $t$ the selected arm $X_t = x$, then the expected reward is $\langle x, \theta^* \rangle$, i.e., $\mathbb{E}[r_t \mid X_t = x] = \langle x, \theta^* \rangle$. We will, throughout, use $x^*$ to denote the optimal arm, $x^* = \arg\max_{x \in \mathcal{X}} \langle x, \theta^* \rangle$ and $\widehat{\theta}$ to denote estimator of $\theta^*$.

In the stochastic linear bandits framework, our overarching objective is to minimize the Nash regret, defined as follows:

$$\text{NR}_\text{T} := \max_{x \in \mathcal{X}} \langle x, \theta^* \rangle - \left( \prod_{t=1}^{\text{T}} \mathbb{E}[\langle X_t, \theta^* \rangle] \right)^{1/\text{T}} \tag{1}$$

Note that the definition of Nash regret is obtained by applying the Nash social welfare (geometric mean) onto ex ante rewards, $\mathbb{E}\left[\langle X_t, \theta^* \rangle\right]$,[4] accrued across the $\text{T}$ rounds.

## 2.1 Sub-Poisson Rewards

In order to model the environment with positive rewards ($r_x > 0$), we assume that the rewards $r_x$ associated with the arms $x \in \mathcal{X}$ are $\nu$-*sub Poisson*, for some parameter $\nu > 0$. Formally, their moment-generating function satisfies the following bound

$$\mathbb{E}\left[e^{\lambda\, r_x}\right] \leq \exp\left(\nu^{-1}\mathbb{E}[r_x]\left(e^{\nu\lambda} - 1\right)\right) = \exp\left(\nu^{-1}\langle x, \theta^* \rangle \left(e^{\nu\lambda} - 1\right)\right) \text{ for all } \lambda \in \mathbb{R}. \tag{2}$$

Note that a Poisson random variable is 1-sub Poisson. To highlight the generality of $\nu$-sub-Poisson distributions, we note that bounded, non-negative random variables are sub-Poisson (Lemma 1). Further, in Lemma 2, we establish a connection between non-negative sub-Gaussian and sub-Poisson random variables.

**Lemma 1.** *Any non-negative random variable $X \in [0, \text{B}]$ is $\text{B}$-sub-Poisson, i.e., if mean $\mathbb{E}[X] = \mu$, then for all $\lambda \in \mathbb{R}$, we have $\mathbb{E}[e^{\lambda X}] \leq \exp\left(B^{-1}\mu\left(e^{B\lambda} - 1\right)\right)$.*

**Lemma 2.** *Let $X$ be a non-negative sub-Gaussian random variable $X$ with mean $\mu = \mathbb{E}[X]$ and sub-Gaussian norm $\sigma$. Then, $X$ is also $\left(\frac{\sigma^2}{\mu}\right)$-sub-Poisson.*

The proofs of Lemmas 1 and 2 appear in Appendix A. Lemma 2 has useful instantiations. In particular, the lemma implies that the half-normal random variable, with variance of $\sigma$, is also a $(C\sigma)$-sub-Poisson, where $C$ is a constant (independent of distribution parameters). Similarly, for other well-studied, positive sub-Gaussian random variables (including truncated and folded normal distributions), the sub-Poisson parameter is small.

Next, we discuss the necessary preliminaries for our algorithm and analysis.

## 2.2 Optimal Design.

Write $\Delta(\mathcal{X})$ to denote the probability simplex associated with the set of arms $\mathcal{X}$. Let $\lambda \in \Delta(\mathcal{X})$ be such a probability distribution over the arms, with $\lambda_x$ denoting the probability of selecting arm $x$. The following optimization problem, defined over the set of arms $\mathcal{X}$, is well-known and is referred to as the G-optimal design problem.

$$\text{Minimize } g(\lambda) \triangleq \max_{x \in \mathcal{X}} ||x||^2_{\mathbf{U}(\lambda)^{-1}}, \text{ where } \lambda \in \Delta(\mathcal{X}) \text{ and } \mathbf{U}(\lambda) = \sum_{x \in \mathcal{X}} \lambda_x x x^T \tag{3}$$

The solution to (3) provides the optimal sequence of arm pulls (for a given budget of rounds) to minimize the confidence width of the estimated rewards for all arms $x \in \mathcal{X}$. The G-optimal design problem connects to the following optimization problem (known as D-optimal design problem):

$$\text{Maximize } f(\lambda) \triangleq \log \text{Det}(\mathbf{U}(\lambda)), \text{ where } \lambda \in \Delta(\mathcal{X}) \text{ and } \mathbf{U}(\lambda) = \sum_{x \in \mathcal{X}} \lambda_x x x^T \tag{4}$$

The lemma below provides an important result of Kiefer and Wolfowitz [15].

**Lemma 3** (Kiefer-Wolfowitz). *If the set $\mathcal{X}$ is compact and $\mathcal{X}$ spans $\mathbb{R}^d$, then there exists $\lambda^* \in \Delta(\mathcal{X})$ supported over at most $d(d+1)/2$ arms such that $\lambda^*$ minimizes the objective in equation (3) with $g(\lambda^*) = d$. Furthermore, $\lambda^*$ is also a maximizer of the D-optimal design objective, i.e., $\lambda^*$ maximizes the function $f(\lambda) = \log \text{Det}(\mathbf{U}(\lambda))$ subject to $\lambda \in \Delta(\mathcal{X})$.*

---

[4]Here, the expectation is with respect to the random variable $X_t$.

---

**Algorithm 1** `GenerateArmSequence` (Subroutine to generate Arm Sequence)

---

**Input:** Arm set $\mathcal{X}$ and sequence length $\widetilde{\mathsf{T}} \in \mathbb{Z}_+$.

1: Find the probability distribution $\lambda \in \Delta(\mathcal{X})$ by maximizing the following objective

$$\log \mathsf{Det}(\mathbf{U}(\lambda_0)) \text{ subject to } \lambda_0 \in \Delta(\mathcal{X}) \text{ and } \mathsf{Supp}(\lambda_0) \leq d(d+1)/2 \qquad (5)$$

2: Initialize multiset $\mathcal{S} = \emptyset$ and set $\mathcal{A} = \mathsf{Supp}(\lambda)$. Also, initialize count $c_z = 0$, for each arm $z \in \mathcal{A}$.
3: Compute distribution $U$ as described in Section 3.1.
4: **for** $i = 1$ to $\widetilde{\mathsf{T}}$ **do**
5:     With probability $1/2$ set flag = SAMPLE-U, otherwise, set flag = D/G-OPT.
6:     **if** flag = SAMPLE-U or $\mathcal{A} = \emptyset$ **then**
7:         Sample an arm $\widehat{x}$ from the distribution $U$, and update multiset $\mathcal{S} \leftarrow \mathcal{S} \cup \{\widehat{x}\}$.
8:     **else if** flag = D/G-OPT **then**
9:         Pick the next arm $z$ in $\mathcal{A}$ (round robin).
10:        Update multiset $\mathcal{S} \leftarrow \mathcal{S} \cup \{z\}$ and increment count $c_z \leftarrow c_z + 1$.
11:        If $c_z \geq \lceil \lambda_z \widetilde{\mathsf{T}}/3 \rceil$, then update $\mathcal{A} \leftarrow \mathcal{A} \setminus \{z\}$.
12:     **end if**
13: **end for**
14: **return** multiset $\mathcal{S}$

---

At several places in our algorithm, our goal is to find a probability distribution that minimizes the non-convex optimization problem (3). However, instead we will maximize the concave function $f(\lambda) = \log \mathsf{Det}(\mathbf{U}(\lambda))$ over $\lambda \in \Delta(\mathcal{X})$. The Frank-Wolfe algorithm, for instance, can be used to solve the D-optimal design problem (4) and compute $\lambda^*$ efficiently ([17], Chapter 21). Lemma 3 ensures that this approach works, since the G-optimal and the D-optimal design problems have the same optimal solution $\lambda^* \in \Delta(\mathcal{X})$, which satisfies $\mathsf{Supp}(\lambda^*) \leq d(d+1)/2$.[5]

### 2.3 John Ellipsoid.

For any convex body $K \subset \mathbb{R}^d$, a John ellipsoid is an ellipsoid with maximal volume that can be inscribed within $K$. It is known that $K$ itself is contained within the John Ellipsoid dilated by a factor of $d$. Formally,[6]

**Lemma 4** ([11]). *Let $K \subset \mathbb{R}^d$ be a convex body (i.e., a compact, convex set with a nonempty interior). Then, there exists an ellipsoid $E$ (called the John ellipsoid) that satisfies $E \subseteq K \subseteq c + d(E - c)$. Here, $c \in \mathbb{R}^d$ denotes the center of $E$ and $c + d(E - c)$ refers to the (dialated) set $\{c + d(x - c) : x \in E\}$.*

## 3 Our Algorithm LINNASH and Main Results

In this section, we detail our algorithm LINNASH (Algorithm 2), and establish an upper bound on the Nash regret achieved by this algorithm. Subsection 3.1 details Part I of LINNASH and related analysis. Then, Subsection 3.2 presents and analyzes Part II of the algorithm. Using the lemmas from these two subsections, the regret bound for the algorithm is established in Subsection 3.3.

### 3.1 Part I: Sampling via John Ellipsoid and Kiefer-Wolfowitz Optimal Design

As mentioned previously, Nash regret is a more challenging objective than average regret: if in any round $t \in [\mathsf{T}]$, the expected[7] reward $\mathbb{E}[r_t]$ is zero (or very close to zero), then geometric mean $(\prod_{t=1}^{\mathsf{T}} \mathbb{E}[r_{X_t}])^{1/\mathsf{T}}$ goes to zero, even if the expected rewards in the remaining rounds are large. Hence, we need to ensure that in every round $t \in [\mathsf{T}]$, specifically the rounds in the beginning of the algorithm, the expected rewards are bounded from below. In [5], this problem was tackled for stochastic multi-armed bandits (MAB) by directly sampling each arm uniformly at random in the initial rounds. Such a sampling ensured that, in each of those initial rounds, the expected reward is bounded from below by the average of the expected rewards. While such a uniform sampling

---

[5]Even though the two optimization problems (3) and (4) share the optimal solution, the optimal objective function values can be different.

[6]The ellipsoid $E$ considered in Lemma 4 is also the ellipsoid of maximal volume contained in $K$ [11].

[7]Here, the expectation is over randomness in algorithm and the reward noise.

---

**Algorithm 2** LINNASH (Nash Regret Algorithm for Finite Set of Arms)

---

**Input:** Arm set $\mathcal{X}$ and horizon of play $T$.

1: Initialize matrix $\mathbf{V} \leftarrow [0]_{d,d}$ and number of rounds $\widetilde{\mathsf{T}} = 3\sqrt{\mathsf{T}d\nu\log(\mathsf{T}|\mathcal{X}|)}$.
   ```
   Part I
   ```
2: Generate arm sequence $\mathcal{S}$ for the first $\widetilde{\mathsf{T}}$ rounds using Algorithm 1.
3: **for** $t = 1$ to $\widetilde{T}$ **do**
4:    Pull the next arm $X_t$ from the sequence $\mathcal{S}$, observe corresponding reward $r_t$, and update $\mathbf{V} \leftarrow \mathbf{V} + X_t X_t^T$
5: **end for**
6: Set estimate $\widehat{\theta} := \mathbf{V}^{-1}\left(\sum_{t=1}^{\widetilde{\mathsf{T}}} r_t X_t\right)$
7: Compute confidence bounds $\mathrm{LNCB}(x, \widehat{\theta}, \widetilde{\mathsf{T}}/3)$ and $\mathrm{UNCB}(x, \widehat{\theta}, \widetilde{\mathsf{T}}/3)$, for all $x \in \mathcal{X}$ (see equation (7))
8: Set $\widetilde{\mathcal{X}} = \left\{x \in \mathcal{X} : \mathrm{UNCB}(x, \widehat{\theta}, \widetilde{\mathsf{T}}/3) \geq \max_{z \in \mathcal{X}} \mathrm{LNCB}(z, \widehat{\theta}, \widetilde{\mathsf{T}}/3)\right\}$ and initialize $\mathsf{T}' = \frac{2}{3}\widetilde{\mathsf{T}}$
   ```
   Part II
   ```
9: **while** end of time horizon $\mathsf{T}$ is reached **do**
10:    Initialize $V = [0]_{d,d}$ to be an all zeros $d \times d$ matrix and $s = [0]_d$ to be an all-zeros vector.
       `// Beginning of new phase.`
11:    Find the probability distribution $\lambda \in \Delta(\widetilde{\mathcal{X}})$ by maximizing the following objective

$$\log \mathsf{Det}(\mathbf{U}(\lambda_0)) \text{ subject to } \lambda_0 \in \Delta(\widetilde{\mathcal{X}}) \text{ and } \mathsf{Supp}(\lambda_0) \leq d(d+1)/2. \tag{6}$$

12:    **for** each arm $a$ in $\mathsf{Supp}(\lambda)$ **do**
13:       Pull arm $a$ for the next $\lceil \lambda_a \mathsf{T}'\rceil$ rounds. Update $\mathbf{V} \leftarrow \mathbf{V} + \lceil \lambda_a \mathsf{T}'\rceil \cdot aa^T$.
14:       Observe $\lceil \lambda_a \mathsf{T}'\rceil$ corresponding rewards $z_1, z_2, \ldots$ and update $s \leftarrow s + (\sum_j z_j)a$.
15:    **end for**
16:    Set estimate $\widehat{\theta} = \mathbf{V}^{-1}s$ and compute $\mathrm{LNCB}(x, \widehat{\theta}, \mathsf{T}')$ and $\mathrm{UNCB}(x, \widehat{\theta}, \mathsf{T}')$, for all $x \in \mathcal{X}$ (see equation (7))
17:    Set $\widetilde{\mathcal{X}} = \left\{x \in \widetilde{\mathcal{X}} : \mathrm{UNCB}(x, \widehat{\theta}, \mathsf{T}') \geq \max_{z \in \mathcal{X}} \mathrm{LNCB}(z, \widehat{\theta}, \mathsf{T}')\right\}$. `// End of phase.`
18:    Update $\mathsf{T}' \leftarrow 2\,\mathsf{T}'$.
19: **end while**

---

strategy is reasonable for the MAB setting, it can be quite unsatisfactory in the current context of linear bandits. To see this, consider a linear bandit instance in which, all—except for one—arms in $\mathcal{X}$ are orthogonal to $\theta^*$. Here, a uniform sampling strategy will lead to an expected reward of $\langle x^*, \theta^*\rangle/|\mathcal{X}|$, which can be arbitrarily small for large cardinality $\mathcal{X}$.

To resolve this issue we propose a novel approach in the initial $\widetilde{\mathsf{T}} := 3\sqrt{\mathsf{T}d\nu\log(\mathsf{T}|\mathcal{X}|)}$ rounds. In particular, we consider the convex hull of the set of arms $\mathcal{X}$—denoted as $\mathrm{cvh}(\mathcal{X})$— and find the center $c \in \mathbb{R}^d$ of the John ellipsoid $E$ for the convex hull $\mathrm{cvh}(\mathcal{X})$. Since $E \subseteq \mathrm{cvh}(\mathcal{X})$, the center $c$ of the John ellipsoid is contained within $\mathrm{cvh}(\mathcal{X})$ as well. Furthermore, via Carathéodory's theorem [10], we can conclude that the center $c$ can be expressed as a convex combination of at most $(d+1)$ points in $\mathcal{X}$. Specifically, there exists a size-$(d+1)$ subset $\mathcal{Y} := \{y_1, \ldots, y_{d+1}\} \subseteq \mathcal{X}$ and convex coefficients $\alpha_1, \ldots, \alpha_{d+1} \in [0,1]$ such that $c = \sum_{i=1}^{d+1} \alpha_i y_i$ with $\sum_{i=1}^{d+1} \alpha_i = 1$. Therefore, the convex coefficients induce a distribution $U \in \Delta(\mathcal{X})$ of support size $d+1$ and with $\mathbb{E}_{x \sim U}[x] = c$.

Lemma 5 below asserts that sampling according to the distribution $U$ leads to an expected reward that is sufficiently large. Hence, $U$ is used in the subroutine `GenerateArmSequence` (Algorithm 1).

In particular, the purpose of the subroutine is to carefully construct a sequence (multiset) of arms $\mathcal{S}$, with size $|\mathcal{S}| = \widetilde{\mathsf{T}}$ and to be pulled in the initial $\widetilde{\mathsf{T}}$ rounds. The sequence $\mathcal{S}$ is constructed such that (i) upon pulling arms from $\mathcal{S}$, we have a sufficiently large expected reward in each pull, and (ii) we obtain an initial estimate of the inner product of the unknown parameter vector $\theta^*$ with all arms in $\mathcal{X}$. Here, objective (i) is achieved by considering the above-mentioned distribution $U$. Now, towards the objective (ii), we compute distribution $\lambda \in \Delta(\mathcal{X})$ by solving the optimization problem (also known as the D-optimal design problem) stated in equation (5).

We initialize sequence $\mathcal{S} = \emptyset$ and run the subroutine `GenerateArmSequence` for $\widetilde{\mathsf{T}}$ iterations. In each iteration (of the for-loop in Line 4), with probability $1/2$, we sample an arm according to the distribution $U$ (Line 7) and include it in $\mathcal{S}$. Also, in each iteration, with remaining probability $1/2$, we consider the computed distribution $\lambda$ and, in particular, pick arms $z$ from the support of $\lambda$ in a round-robin manner. We include such arms $z$ in $\mathcal{S}$ while ensuring that, at the end of the subroutine,

each such arm $z \in \mathsf{Supp}(\lambda)$ is included at least $\lceil \lambda_z \widetilde{\mathsf{T}}/3 \rceil$ times. We return the curated sequence of arms $\mathcal{S}$ at the end of the subroutine.

Our main algorithm LINNASH (Algorithm 2) first calls subroutine `GenerateArmSequence` to generated the sequence $\mathcal{S}$. Then, the algorithm LINNASH sequentially pulls the arms $X_t$ from $\mathcal{S}$, for $1 \le t \le \widetilde{\mathsf{T}}$ rounds. For these initial $\widetilde{\mathsf{T}} = |\mathcal{S}|$ rounds, let $r_t$ denote the noisy, observed rewards. Using these $\widetilde{\mathsf{T}}$ observed rewards, the algorithm computes the ordinary least squares (OLS) estimate $\widehat{\theta}$ (see Line 6 in Algorithm 2); in particular, $\widehat{\theta} := (\sum_{t=1}^{\widetilde{\mathsf{T}}} X_t X_t^T)^{-1}(\sum_{t=1}^{\widetilde{\mathsf{T}}} r_t X_t)$. The algorithm uses the OLS estimate $\widehat{\theta}$ to eliminate several low rewarding arms (in Lines 7 and 8 in Algorithm 2). This concludes Part I of the algorithm LINNASH.

Before detailing Part II (in Subsection 3.2), we provide a lemma to be used in the analysis of Part I of LINNASH.

**Lemma 5.** *Let $c \in \mathbb{R}^d$ denote the center of a John ellipsoid for the convex hull $\mathrm{cvh}(\mathcal{X})$ and let $U \in \Delta(\mathcal{X})$ be a distribution that satisfies $\mathbb{E}_{x \sim U}\, x = c$. Then, it holds that*

$$\mathbb{E}_{x \sim U}[\langle x, \theta^* \rangle] \ge \frac{\langle x^*, \theta^* \rangle}{(d+1)}.$$

*Proof.* Lemma 4 ensures that there exists a positive definite matrix $\mathbf{H}$ with the property that

$$\left\{ x \in \mathbb{R}^d : \sqrt{(x-c)^T \mathbf{H}(x-c)} \le 1 \right\} \subseteq \mathrm{cvh}(\mathcal{X}) \subseteq \left\{ x \in \mathbb{R}^d : \sqrt{(x-c)^T \mathbf{H}(x-c)} \le d \right\}.$$

Now, write $y := c - \frac{x^* - c}{d}$ and note that

$$\sqrt{(y-c)^T \mathbf{H}(y-c)} = \sqrt{\frac{(x^*-c)^T \mathbf{H}(x^*-c)}{d^2}} \le 1 \qquad\qquad \text{(since } x^* \in \mathrm{cvh}(\mathcal{X}))$$

Therefore, $y \in \mathrm{cvh}(\mathcal{X})$. Recall that, for all arms $x \in \mathcal{X}$, the associated reward $(r_x)$ is non-negative and, hence, the rewards' expected value satisfies $\langle x, \theta^* \rangle \ge 0$. This inequality and the containment $y \in \mathrm{cvh}(\mathcal{X})$ give us $\langle y, \theta^* \rangle \ge 0$. Substituting $y = c - \frac{x^* - c}{d}$ in the last inequality leads to $\langle c, \theta^* \rangle \ge \langle x^*, \theta^* \rangle/(d+1)$. Given that $\mathbb{E}_{x \sim U}\,[x] = c$, we obtain the desired inequality $\mathbb{E}_{x \sim U}\langle x, \theta^* \rangle = \langle c, \theta^* \rangle \ge \frac{\langle x^*, \theta^* \rangle}{(d+1)}$. $\qquad\square$

Note that at each iteration of the subroutine `GenerateArmSequence`, with probability $1/2$, we insert an arm into $\mathcal{S}$ that is sampled according to $U$. Using this observation and Lemma 5, we obtain that, for any round $t \in [\widetilde{\mathsf{T}}]$ and for the random arm $X_t$ pulled from the sequence $\mathcal{S}$ according to our procedure, the observed reward $r_{X_t}$ must satisfy $\mathbb{E}[r_{X_t}] \ge \frac{\langle x^*, \theta^* \rangle}{2(d+1)}$.[8]

Further, recall that in the subroutine `GenerateArmSequence`, we insert arms $x \in \mathsf{Supp}(\lambda)$ at least $\lceil \lambda_x \widetilde{\mathsf{T}}/3 \rceil$ times, where $\lambda$ corresponds to the solution of D-optimal design problem defined in equation (5). Therefore, we can characterize the confidence widths of the estimated rewards for each arm in $\mathcal{X}$ computed using the least squares estimate $\widehat{\theta}$ computed in Line 7 in Algorithm 2.

Broadly speaking, we can show that all arms with low expected reward (less than a threshold) also have an estimated reward at most twice the true reward. On the other hand, high rewarding arms must have an estimated reward to be within a factor of 2 of the true reward. Thus, based on certain high probability confidence bounds (equation (7)), we can eliminate arms in $\mathcal{X}$ with true expected reward less than some threshold, with high probability.

## 3.2 Part II: Phased Elimination via Estimate Dependent Confidence Widths

Note that while analyzing average regret via confidence bound algorithms, it is quite common to use, for each arm $x$, a confidence width (interval) that does not depend on $x$'s estimated reward. This is a reasonable design choice for bounding average regret, since the regret incurred at each round is the sum of confidence intervals that grow smaller with the round index and, hence, this choice

---

[8]Here, the expectation is over both the randomness in including arm $X_t$ in $\mathcal{S}$ and the noise in the reward.

leads to a small average regret. However, for the analysis of the Nash regret, a confidence width that is independent of the estimated reward can be highly unsatisfactory: the confidence width might be larger than the optimal $\langle x^*, \theta^* \rangle$. This can in turn allow an arm with extremely low reward to be pulled leading to the geometric mean going to zero. In order to alleviate this issue, it is vital that our confidence intervals are reward dependent. This in turn, requires one to instantiate concentration bounds similar to the multiplicative version of the standard Chernoff bound. In general, multiplicative forms of concentration bounds are much stronger than the additive analogues [16]. In prior work [5] on Nash regret for the stochastic multi-armed bandits setting, such concentration bounds were readily available through the multiplicative version of the Chernoff bound. However, in our context of linear bandits, the derivation of analogous concentration bounds (and the associated confidence widths) is quite novel and requires a careful use of the sub-Poisson property.

In particular, we use the following confidence bounds (with estimate dependent confidence widths) in our algorithm. We define the lower and upper confidence bounds considering any arm $x$, any least squares estimator $\phi$ (of $\theta^*$), and $t$ the number of observations used to compute the estimator $\phi$. That is, for any triple $(x, \phi, t) \in \mathcal{X} \times \mathbb{R}^d \times [\mathsf{T}]$, we define Lower Nash Confidence Bound (LNCB) and Upper Nash Confidence Bound (UNCB) as follows:

$$\text{LNCB}(x, \phi, t) := \langle x, \phi \rangle - 6\sqrt{\frac{\langle x, \phi \rangle \nu d \log(\mathsf{T}|\mathcal{X}|)}{t}}$$

$$\text{UNCB}(x, \phi, t) := \langle x, \phi \rangle + 6\sqrt{\frac{\langle x, \phi \rangle \nu d \log(\mathsf{T}|\mathcal{X}|)}{t}}. \tag{7}$$

As mentioned previously, the confidence widths in equation (7) are estimate dependent.

Next, we provide a high level overview of Part II in Algorithm 2. This part is inspired from the phased elimination algorithm for the average regret ([17], Chapter 21); a key distinction here is to use the Nash confidence bounds defined in (7). Part II in Algorithm 2 begins with the set of arms $\widetilde{\mathcal{X}} \subseteq \mathcal{X}$ obtained after an initial elimination of low rewarding arms in Part I . Subsequently, Part II runs in phases of exponentially increasing length and eliminates sub-optimal arms in every phase.

Suppose at the beginning of the $\ell^{\text{th}}$ phase, $\widetilde{\mathcal{X}}$ is the updated set of arms. We solve the D-optimal design problem (see (6)) corresponding to $\widetilde{\mathcal{X}}$ to obtain a distribution $\lambda \in \Delta(\widetilde{\mathcal{X}})$. For the next $O(d^2 + 2^\ell \widetilde{\mathsf{T}})$ rounds, we pull arms $a$ in the support of $\lambda$ (Line 13): each arm $a \in \mathsf{Supp}(\lambda)$ is pulled $\lceil \lambda_a \mathsf{T}' \rceil$ times where $\mathsf{T}' = O(2^\ell \widetilde{\mathsf{T}})$. Using the data covariance matrix and the observed noisy rewards, we recompute: (1) an improved estimate $\widehat{\theta}$ (of $\theta^*$) and (2) improved confidence bounds for every surviving arm. Then, we eliminate arms based on the confidence bounds and update the set of surviving arms (Lines 16 and 17).

The following lemma provides the key concentration bound for the least squares estimate.

**Lemma 6.** *Let $x_1, x_2, \ldots, x_s \in \mathbb{R}^d$ be a fixed set of vectors and let $r_1, r_2, \ldots, r_s$ be independent $\nu$-sub-Poisson random variables satisfying $\mathbb{E} r_s = \langle x_s, \theta^* \rangle$ for some unknown $\theta^*$. Further, let matrix $\mathbf{V} = \sum_{j=1}^s x_j x_j^T$ and $\widehat{\theta} = \mathbf{V}^{-1}\left(\sum_j r_j x_j\right)$ be the least squares estimator of $\theta^*$. Consider any $z \in \mathbb{R}^d$ with the property that $z^T \mathbf{V}^{-1} x_j \leq \gamma$ for all $j \in [s]$. Then, for any $\delta \in [0, 1]$ we have*

$$\mathbb{P}\left\{\langle z, \widehat{\theta} \rangle \geq (1+\delta)\langle z, \theta^* \rangle\right\} \leq \exp\left(-\frac{\delta^2 \langle z, \theta^* \rangle}{3\nu\gamma}\right) \quad and \tag{8}$$

$$\mathbb{P}\left\{\langle z, \widehat{\theta} \rangle \leq (1-\delta)\langle z, \theta^* \rangle\right\} \leq \exp\left(-\frac{\delta^2 \langle z, \theta^* \rangle}{2\nu\gamma}\right). \tag{9}$$

Lemma 6 is established in Appendix B. Using this lemma, we can show that the optimal arm $x^*$ is never eliminated with high probability.

**Lemma 7.** *Consider any bandit instance in which for the optimal arm $x^* \in \mathcal{X}$ we have $\langle x^*, \theta^* \rangle \geq 192\sqrt{\frac{d\nu}{\mathsf{T}}}\log(\mathsf{T}|\mathcal{X}|)$. Then, with probability at least $\left(1 - \frac{4\log \mathsf{T}}{\mathsf{T}}\right)$, the optimal arm $x^*$ always exists in the surviving set $\widetilde{\mathcal{X}}$ in Part I and in every phase in Part II of Algorithm 2.*

Finally, using Lemmas 6 and 7, we show that, with high probability, in every phase of Part II all the surviving arms $x \in \widetilde{\mathcal{X}}$ have sufficiently high reward means.

**Lemma 8.** *Consider any phase $\ell$ in Part II of Algorithm 2 and let $\widetilde{\mathcal{X}}$ be the surviving set of arms at the beginning of that phase. Then, with $\widetilde{\mathsf{T}} = \sqrt{d\nu\mathsf{T}\log(\mathsf{T}|\mathcal{X}|)}$, we have*

$$\Pr\left\{\langle x, \theta^*\rangle \geq \langle x^*, \theta^*\rangle - 25\sqrt{\frac{3d\nu\langle x^*, \theta^*\rangle \log\left(\mathsf{T}|\mathcal{X}|\right)}{2^\ell \cdot \widetilde{\mathsf{T}}}} \text{ for all } x \in \widetilde{\mathcal{X}}\right\} \geq 1 - \frac{4\log\mathsf{T}}{\mathsf{T}} \quad (10)$$

*Here, $\nu$ is the sub-Poisson parameter of the stochastic rewards.*

The proofs of the Lemmas 7 and 8 are deferred to Appendix C.

## 3.3 Main Result

This section states and provides a proof sketch of the main theorem.[9]

**Theorem 1.** *For any given stochastic linear bandits problem with (finite) set of arms $\mathcal{X} \subset \mathbb{R}^d$, time horizon $\mathsf{T} \in \mathbb{Z}_+$, and $\nu$-sub-Poisson rewards, Algorithm 2 achieves Nash regret*

$$\mathrm{NR}_\mathsf{T} = O\left(\beta\sqrt{\frac{d\,\nu}{\mathsf{T}}}\log(\mathsf{T}|\mathcal{X}|)\right).$$

*Here, $\beta = \max\{1, \langle x^*, \theta^*\rangle \log d\}$, with $x^* \in \mathcal{X}$ denoting the optimal arm and $\theta^*$ the (unknown) parameter vector.*

Note that, in Theorem 1, lower the value of the optimal expected reward, $\langle x^*, \theta^*\rangle$, stronger is the Nash regret guarantee. In particular, with a standard normalization assumption that $\langle x^*, \theta^*\rangle \leq 1$ and for 1-sub Poisson rewards, we obtain a Nash regret of $O\left(\sqrt{\frac{d}{\mathsf{T}}}\log(\mathsf{T}|\mathcal{X}|)\right)$. Also, observe that the regret guarantee provided in Theorem 1 depends logarithmically on the size of $\mathcal{X}$. Hence, the Nash regret is small even when $|\mathcal{X}|$ is polynomially large in $d$.

*Proof Sketch of Theorem 1:* We condition on the event $\mathcal{E}$ defined as the intersection of the (high probability) events defined in Lemmas 7 and 8, respectively. Union bound implies that event $\mathcal{E}^c$ holds with probability at most $O(\mathsf{T}^{-1})$. For Part I, in the first $\widetilde{\mathsf{T}} = 3\sqrt{\mathsf{T}d\nu\log(\mathsf{T}|\mathcal{X}|)}$ rounds, we bound from below the product of the expected rewards, using Lemma 5, as follows

$$\prod_{t=1}^{\widetilde{\mathsf{T}}} \mathbb{E}[\langle X_t, \theta^*\rangle \mid \mathcal{E}]^{\frac{1}{\mathsf{T}}} \geq \left(\frac{\langle x^*, \theta^*\rangle}{2(d+1)}\right)^{\frac{\widetilde{\mathsf{T}}}{\mathsf{T}}} \geq \langle x^*, \theta^*\rangle^{\frac{\widetilde{\mathsf{T}}}{\mathsf{T}}}\left(1 - \frac{\widetilde{\mathsf{T}}\log(2(d+1))}{\mathsf{T}}\right).$$

For remaining rounds, we invoke Lemma 8 (specifically, equation (10)) to obtain

$$\prod_{t=\widetilde{\mathsf{T}}+1}^{\mathsf{T}} \mathbb{E}[\langle X_t, \theta^*\rangle \mid \mathcal{E}]^{\frac{1}{\mathsf{T}}} \geq \langle x^*, \theta^*\rangle^{\frac{\mathsf{T}-\widetilde{\mathsf{T}}}{\mathsf{T}}}\prod_{\ell=1}^{\log\mathsf{T}}\left(1 - 50\frac{|\widetilde{\mathsf{T}} \cdot (2^\ell/3) + 2d^2|}{\mathsf{T}}\sqrt{\frac{3d\nu\log(\mathsf{T}|\mathcal{X}|)}{\langle x^*, \theta^*\rangle 2^\ell \cdot \widetilde{\mathsf{T}}}}\right).$$

The above equations reduce to the following bound on the geometric mean of the expected rewards (across the $\mathsf{T}$ rounds):

$$\prod_{t=1}^{\mathsf{T}} \mathbb{E}[\langle X_t, \theta^*\rangle]^{\frac{1}{\mathsf{T}}} \geq \prod_{t=1}^{\mathsf{T}}\left(\mathbb{E}[\langle X_t, \theta^*\rangle \mid \mathcal{E}]\right)^{\frac{1}{\mathsf{T}}}\Pr(\mathcal{E}) \geq \langle x^*, \theta^*\rangle\left(1 - 100\sqrt{\frac{d\nu}{\mathsf{T}\langle x^*, \theta^*\rangle}}\log(\mathsf{T}|\mathcal{X}|)\right).$$

Therefore, the Nash regret of LINNASH satisfies

$$\mathrm{NR}_\mathsf{T} = \langle x^*, \theta^*\rangle - \left(\prod_{t=1}^{\mathsf{T}} \mathbb{E}[\langle X_t, \theta^*\rangle]\right)^{1/\mathsf{T}} = O\left(\sqrt{\frac{d\nu\langle x^*, \theta^*\rangle}{\mathsf{T}}}\log(\mathsf{T}|\mathcal{X}|)\right).$$

This completes the proof sketch.

---

[9]Due to space restrictions, the complete analysis is deferred to Appendix C

**Computational Efficiency of LinNash.** We note that Algorithm 2 (LinNash) executes in polynomial time. In particular, the algorithm calls the subroutine GenerateArmSequence in Part I for computing the John Ellipsoid. Given a set of arm vectors as input, this ellipsoid computation can be performed efficiently (see Chapter 3 in [25]). In fact, for our purposes an approximate version of the John Ellipsoid suffices, and such an approximation can be found much faster [8]; specifically, in time $O(|\mathcal{X}|^2 d)$. Furthermore, the algorithm solves the D-optimal design problem, once in Part I and at most $O(\log \mathsf{T})$ times in Part II. The D-optimal design is a concave maximization problem, which can be efficiently solved using, say, the Frank-Wolfe algorithm with rank-1 updates. Each iteration takes $O(|\mathcal{X}|^2)$ time, and the total number of iterations is at most $O(d)$ (see, e.g., Chapter 21 of [17] and Chapter 3 in [25]). Overall, we get that LinNash is a polynomial-time algorithm.

## 4   Extension of Algorithm LinNash for Infinite Arms

The regret guarantee in Theorem 1 depends logarithmically on $|\mathcal{X}|$. Such a dependence makes the guarantee vacuous when the set of arms $\mathcal{X}$ is infinitely large (or even $|\mathcal{X}| = \Omega(2^{\sqrt{\mathsf{T} d^{-1}}})$). To resolve this limitation, we extend LinNash with a modified confidence width that depends only on the largest estimated reward $\gamma := \max_{x \in \mathcal{X}} \langle x, \widehat{\theta} \rangle$. Specifically, we consider the confidence width $16\sqrt{\frac{\gamma\, d^{\frac{5}{2}}\, \nu\, \log(\mathsf{T})}{\mathsf{T}'}}$, for all the arms, and select the set of surviving arms in each phase (of Part II of the algorithm for infinite arms) as follows:

$$\widetilde{\mathcal{X}} = \left\{ x \in \mathcal{X} : \langle x, \widehat{\theta} \rangle \geq \gamma - 16\sqrt{\frac{\gamma\, d^{\frac{5}{2}}\, \nu\, \log(\mathsf{T})}{\mathsf{T}'}} \right\} \tag{11}$$

See Algorithm 3 in Appendix D for details. The theorem below is the main result of this section.

**Theorem 2.** *For any given stochastic linear bandits problem with set of arms $\mathcal{X} \subset \mathbb{R}^d$, time horizon $\mathsf{T} \in \mathbb{Z}_+$, and $\nu$-sub-Poisson rewards, Algorithm 2 achieves Nash regret*

$$\mathsf{NR}_\mathsf{T} = O\left( \beta \frac{d^{\frac{5}{4}}\sqrt{\nu}}{\sqrt{\mathsf{T}}} \log(\mathsf{T}) \right),$$

*Here, $\beta = \max\{1,\ \langle x^*, \theta^* \rangle \log d\}$, with $x^* \in \mathcal{X}$ denoting the optimal arm and $\theta^*$ the (unknown) parameter vector.*

Proof of Theorem 2 and a detailed regret analysis of Algorithm 3 can be found in Appendix D.

## 5   Conclusion and Future Work

Fairness and welfare considerations have emerged as a central design objectives in online decision-making contexts. Motivated, broadly, by such considerations, the current work addresses the notion of Nash regret in the linear bandits framework. We develop essentially tight Nash regret bounds for linear bandit instances with a finite number of arms.

In addition, we extend this guarantee to settings wherein the number of arms is infinite. Here, our regret bound scales as $d^{5/4}$, where $d$ is the ambient dimension. Note that, for linear bandits with infinite arms, [1] obtains a bound of $d/\sqrt{\mathsf{T}}$ for average regret. We conjecture that a similar dependence should be possible for Nash regret as well and pose this strengthening as a relevant direction of future work. Another important direction would be to study Nash regret for more other bandit frameworks (such as contextual bandits and combinatorial bandits) and Markov Decision Processes (MDPs).

## Acknowledgments and Disclosure of Funding

Siddharth Barman's reserach is supported by a SERB Core research grant (CRG/2021/006165).

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

**Limitations:** The main contributions of our works are theoretical. From a theoretical point of view, the limitations of our paper are discussed in Section 5. In particular, we believe that tightening the gap between the upper and lower bounds in Nash regret for an infinite set of arms will require novel and non-trivial algorithmic ideas - we leave this as an important direction of future work.

**Broader Impact:** Due to the theoretical nature of this work, we do not foresee any adverse societal impact of this work.

## A    Proof of Lemmas 1 and 2

**Lemma 1.** *Any non-negative random variable $X \in [0, \mathsf{B}]$ is $\mathsf{B}$-sub-Poisson, i.e., if mean $\mathbb{E}[X] = \mu$, then for all $\lambda \in \mathbb{R}$, we have $\mathbb{E}[e^{\lambda X}] \leq \exp\left(B^{-1}\mu\left(e^{B\lambda} - 1\right)\right)$.*

*Proof.* For random variable $X$ we have

$$
\begin{aligned}
\mathbb{E}\left[\exp\left(\lambda X\right)\right] &= \mathbb{E}\left[\exp\left(\lambda B \frac{X}{B} + (1 - \frac{X}{B})0\right)\right] \\
&\leq \mathbb{E}\left[\frac{X}{B}e^{(\lambda B)} + \left(1 - \frac{X}{B}\right)e^0\right] && \text{(due to convexity of } e^x) \\
&= 1 + \frac{\mathbb{E}\left[X\right]}{\mathsf{B}}\left(e^{\lambda \mathsf{B}} - 1\right) \\
&\leq 1 + \frac{\mu}{\mathsf{B}}\left(e^{\lambda \mathsf{B}} - 1\right) \\
&\leq \exp\left(\frac{\mu}{\mathsf{B}}\left(e^{\lambda \mathsf{B}} - 1\right)\right).
\end{aligned}
$$

$\square$

**Lemma 2.** *Let $X$ be a non-negative sub-Gaussian random variable $X$ with mean $\mu = \mathbb{E}[X]$ and sub-Gaussian norm $\sigma$. Then, $X$ is also $\left(\frac{\sigma^2}{\mu}\right)$-sub-Poisson.*

*Proof.* Since $X$ is a $\sigma$-sub-Gaussian random variable, for any non-negative scalar $s \geq 0$, we have

$$
\begin{aligned}
\mathbb{E}[e^{sX}] &\leq \exp\left(s\mu + \frac{(s\sigma)^2}{2}\right) \\
&= \exp\left(\frac{\mu^2}{\sigma^2}\left(\frac{s\sigma^2}{\mu} + \frac{1}{2}\left(\frac{s\sigma^2}{\mu}\right)^2\right)\right)
\end{aligned}
\tag{12}
$$

The fact that $X$ is a positive random variable implies that the mean $\mu > 0$. Also, the considered scalar $s \geq 0$ and, hence, the term $\frac{s\sigma^2}{\mu} > 0$. Also, recall that $e^x \geq 1 + x + \frac{x^2}{2}$, for any non-negative $x$. Using these observations and equation (12), we obtain

$$
\mathbb{E}[e^{sX}] \leq \exp\left(\frac{\mu^2}{\sigma^2}\left(e^{\frac{s\sigma^2}{\mu}} - 1\right)\right)
\tag{13}
$$

For random variable $X$, inequality (13) ensures that the required mgf bound (equation (2)) holds for all non-negative $s$ and with sub-Poisson parameter equal to $\frac{\sigma^2}{\mu}$.

We next complete the proof by showing that the mgf bound holds for negative $s$ as well. Towards this, write $B := \frac{\sigma^2}{\mu}$ and define random variable $Y := \mathbf{1}_{\{X \leq B\}} X + \mathbf{1}_{\{X > B\}} B$. Note that $Y$ is a positive, bounded random variable. Furthermore, for any negative $s$, we have $\exp(sY) \geq \exp(sX)$. Therefore, for a negative $s$, it holds that $\mathbb{E}\left[\exp(sX)\right] \leq \mathbb{E}\left[\exp(sY)\right]$. Since positive random variable $Y \in [0, B]$, the mgf bound obtained in Lemma 1 gives us

$$
\mathbb{E}[e^{sX}] \leq \mathbb{E}\left[e^{sY}\right] \leq \exp\left(\frac{\mu}{\mathsf{B}}\left(e^{s\mathsf{B}} - 1\right)\right).
$$

Since $B := \frac{\sigma^2}{\mu}$, the mgf bound (equation (2)) on $X$ holds for negative $s$ as well. This, overall, shows that $X$ is a $\left(\frac{\sigma^2}{\mu}\right)$-sub-Poission random variable. The lemma stands proved. $\square$

# B  Proof of Concentration Bounds

**Lemma 6.** *Let $x_1, x_2, \ldots, x_s \in \mathbb{R}^d$ be a fixed set of vectors and let $r_1, r_2, \ldots, r_s$ be independent $\nu$-sub-Poisson random variables satisfying $\mathbb{E}r_s = \langle x_s, \theta^* \rangle$ for some unknown $\theta^*$. Further, let matrix $\mathbf{V} = \sum_{j=1}^{s} x_j x_j^T$ and $\widehat{\theta} = \mathbf{V}^{-1} \left( \sum_j r_j x_j \right)$ be the least squares estimator of $\theta^*$. Consider any $z \in \mathbb{R}^d$ with the property that $z^T \mathbf{V}^{-1} x_j \leq \gamma$ for all $j \in [s]$. Then, for any $\delta \in [0, 1]$ we have*

$$\mathbb{P}\left\{ \langle z, \widehat{\theta} \rangle \geq (1 + \delta)\langle z, \theta^* \rangle \right\} \leq \exp\left( -\frac{\delta^2 \langle z, \theta^* \rangle}{3\nu\gamma} \right) \ \textit{and} \tag{8}$$

$$\mathbb{P}\left\{ \langle z, \widehat{\theta} \rangle \leq (1 - \delta)\langle z, \theta^* \rangle \right\} \leq \exp\left( -\frac{\delta^2 \langle z, \theta^* \rangle}{2\nu\gamma} \right). \tag{9}$$

*Proof.* We use the Chernoff method to get an upper bound on the desired probabilities, as shown below

$$
\begin{aligned}
\mathbb{P}\left\{ \langle z, \widehat{\theta} \rangle \geq (1 + \delta)\langle z, \theta^* \rangle \right\} &= \mathbb{P}\left( \exp(c\,\langle z, \widehat{\theta} \rangle) \geq \exp(c(1 + \delta)\langle z, \theta^* \rangle) \right) \quad \text{(for some constant } c) \\
&\leq \frac{\mathbb{E}[\exp\left( c\, z^T \mathbf{V}^{-1} \left( \sum_t r_t x_t \right) \right)]}{\exp(c\,(1 + \delta)\langle z, \theta^* \rangle)} \\
&= \frac{\prod_{t=1}^{s} \mathbb{E}[\exp\left( c\, r_t \mathbf{V}^{-1} x_t \right)]}{\exp(c\,(1 + \delta)\langle z, \theta^* \rangle)} \quad (r_t\text{'s are independent}) \\
&\leq \frac{\prod_{t=1}^{s} \exp\left( \frac{\mathbb{E}[r_t]}{\nu} \left( e^{c\nu z^T \mathbf{V}^{-1} x_t} - 1 \right) \right)}{\exp(c\,(1 + \delta)\langle z, \theta^* \rangle)} \quad (r_t \text{ is sub Poisson}) \\
&= \exp\left( -c\langle z, \theta^* \rangle(1 + \delta) + \sum_{t=1}^{s} \frac{\langle x, \theta^* \rangle}{\nu} \left( e^{c\,\nu z^T \mathbf{V}^{-1} x_t} - 1 \right) \right).
\end{aligned}
$$

Substituting $c = \frac{\log(1+\delta)}{\nu\gamma}$, we get

$$\mathbb{P}\left\{ \langle z, \widehat{\theta} \rangle \geq (1 + \delta)\langle z, \theta^* \rangle \right\} \leq \exp\left( -\frac{\langle z, \theta^* \rangle}{\nu\gamma}(1 + \delta) \log(1 + \delta) + \sum_{t=1}^{s} \frac{\langle x_t, \theta^* \rangle}{\nu} \left( (1 + \delta)^{\frac{1}{\gamma} z^T \mathbf{V}^{-1} x_t} - 1 \right) \right). \tag{14}$$

Since $\frac{1}{\gamma} z^T \mathbf{V}^{-1} x_t \leq 1$ we have $(1 + \delta)^{\frac{1}{\gamma} z^T \mathbf{V}^{-1} x_t} \leq 1 + \delta \cdot \frac{1}{\gamma} z^T \mathbf{V}^{-1} x_t$. Substituting in (14) we get

$$
\begin{aligned}
&\mathbb{P}\left\{ \langle z, \widehat{\theta} \rangle \geq (1 + \delta)\langle z, \theta^* \rangle \right\} \\
&\leq \exp\left( -\frac{1}{\nu\gamma} \langle z, \theta^* \rangle(1 + \delta) \log(1 + \delta) + \sum_{t=1}^{s} \langle x_t, \theta^* \rangle \cdot \frac{\delta}{\nu\gamma} z^T \mathbf{V}^{-1} x_t \right) \\
&= \exp\left( -\frac{1}{\nu\gamma} \langle z, \theta^* \rangle(1 + \delta) \log(1 + \delta) + \frac{\delta}{\nu\gamma} \sum_{t=1}^{s} \theta^{*T} x_t x_t^T \mathbf{V}^{-1} z \right) \quad \text{(rearranging terms)} \\
&= \exp\left( -\frac{1}{\nu\gamma} \langle z, \theta^* \rangle(1 + \delta) \log(1 + \delta) + \frac{\delta}{\nu\gamma} \langle z, \theta^* \rangle \right). \quad (\sum_{t=1}^{s} x_t x_t^T = \mathbf{V})
\end{aligned}
$$

Using the logarithmic inequality $\log(1 + \delta) \geq \frac{2\delta}{2+\delta}$, we further simplify as

$$
\begin{aligned}
\mathbb{P}\left\{ \langle z, \widehat{\theta} \rangle \geq (1 + \delta)\langle z, \theta^* \rangle \right\} &\leq \exp\left( -\frac{\langle z, \theta^* \rangle}{\nu\gamma} \left( (1 + \delta) \log(1 + \delta) - \delta \right) \right) \\
&\leq \exp\left( \frac{-\delta^2 \langle z, \theta^* \rangle}{(2 + \delta)\,\nu\gamma} \right) \\
&\leq \exp\left( \frac{-\delta^2 \langle z, \theta^* \rangle}{3\nu\gamma} \right). \quad \text{(since } \delta \in [0, 1])
\end{aligned}
$$

Following similar steps and substituting $c = \frac{\log(1-\delta)}{\nu\gamma}$, we obtain a bound on the lower tail (inequality 9):

$$\mathbb{P}\left\{\langle z, \widehat{\theta}\rangle \leq (1-\delta)\langle z, \theta^*\rangle\right\} \leq \exp\left(-\frac{1}{\nu\gamma}\langle z, \theta^*\rangle(1-\delta)\log(1-\delta) - \frac{\delta}{\nu\gamma}\langle z, \theta^*\rangle\right).$$

Now, using the logarithmic inequality $(1-\delta)\log(1-\delta) \geq -\delta + \frac{\delta^2}{2}$, we get

$$\mathbb{P}\left\{\langle z, \widehat{\theta}\rangle \leq (1-\delta)\langle z, \theta^*\rangle\right\} \leq \exp\left(\frac{-\delta^2\langle z, \theta^*\rangle}{2\nu\gamma}\right)$$

$\square$

Combining (9) and (8) we get the following Corollary.

**Corollary 9.** *Using the notations as in Lemma 6, we have*

$$\mathbb{P}\left\{|\langle z, \widehat{\theta}\rangle - \langle z, \theta^*\rangle| \geq \delta\langle z, \theta^*\rangle\right\} \leq 2\exp\left(-\frac{\delta^2\langle z, \theta^*\rangle}{3\gamma}\right). \tag{15}$$

The next two lemmas are variants of Lemma 6 where we bound the error in terms of an upper bound on $\langle z, \theta^*\rangle$.

**Lemma 10.** *Let $x_1, x_2, \ldots, x_s \in \mathbb{R}^d$ be a fixed set of vectors and let $r_1, r_2, \ldots, r_s$ be independent $\nu-$sub Poisson random variables satisfying $\mathbb{E}r_s = \langle x_s, \theta^*\rangle$ for some unknown $\theta^*$. In that case, let matrix $\mathbf{V} = \sum_{j=1}^{s} x_j x_j^T$ and $\widehat{\theta} = \mathbf{V}^{-1}\left(\sum_j r_j x_j\right)$ be the least squares estimator of $\theta^*$. Consider any $z \in \mathbb{R}^d$ that satisfies $z^T\mathbf{V}^{-1}x_j \leq \gamma$ for all $j \in [s]$ and $\langle z, \theta^*\rangle \leq \alpha$. Then for any $\delta \in [0, 1]$ we have*

$$\mathbb{P}\left\{\langle z, \widehat{\theta}\rangle \geq (1+\delta)\alpha\right\} \leq e^{-\frac{\delta^2\alpha}{3\gamma\nu}}. \tag{16}$$

*Proof.* Following the same approach as in the proof of Lemma 6, we have

$$\mathbb{P}\left\{\langle z, \widehat{\theta}\rangle \geq (1+\delta)\alpha\right\} \leq \frac{\mathbb{E}[\exp(c\, z^T\mathbf{V}^{-1}\left(\sum_t r_t x_t\right))]}{\exp(c\,(1+\delta)\alpha)}$$

$$\leq \exp\left(-c\alpha(1+\delta) + \sum_{t=1}^{s} \frac{\langle x_t, \theta^*\rangle}{\nu}\left(e^{c\nu z^T\mathbf{V}^{-1}x_t} - 1\right)\right)$$

$$(r_t \text{ are sub-poisson and independent})$$

Now, substituting $c = \frac{1}{\nu\gamma}\log(1+\delta))$ and using $(1+\delta)^{\frac{1}{\gamma}z^T\mathbf{V}^{-1}x_t} \leq 1 + \delta \cdot \frac{1}{\gamma}z^T\mathbf{V}^{-1}x_t$ we have

$$\mathbb{P}\left\{\langle z, \widehat{\theta}\rangle \geq (1+\delta)\alpha\right\} \leq \exp\left(-\frac{1}{\gamma\nu}\alpha(1+\delta)\log(1+\delta) + \sum_{t=1}^{s} \frac{\langle x_t, \theta^*\rangle}{\nu}\theta^*\left((1+\delta)^{\frac{1}{\gamma}z^T\mathbf{V}^{-1}x_t} - 1\right)\right)$$

$$\leq \exp\left(-\frac{1}{\nu\gamma}\alpha(1+\delta)\log(1+\delta) + \frac{\delta}{\nu\gamma}\sum_{t=1}^{s}\theta^{*T}x_t x_t^T\mathbf{V}^{-1}Z\right)$$

$$= \exp\left(-\frac{1}{\nu\gamma}\alpha(1+\delta)\log(1+\delta) + \frac{\delta}{\nu\gamma}\langle z, \theta^*\rangle\right)$$

$$\leq \exp\left(-\frac{1}{\nu\gamma}\alpha(1+\delta)\log(1+\delta) + \frac{\delta}{\nu\gamma}\alpha\right) \qquad (\alpha \geq \langle z, \theta^*\rangle)$$

$$\leq \exp\left(\frac{-\delta^2\alpha}{(2+\delta)\nu\gamma}\right) \qquad \left(\text{using } \log(1+\delta) \geq \frac{2\delta}{2+\delta}\right)$$

Since $\delta \in [0, 1]$, we have the desired result. $\square$

**Lemma 11.** *Using the same notations as in Lemma 10, for any $\delta \in [0, 1]$, the following holds*

$$\mathbb{P}\left\{\langle z, \widehat{\theta}\rangle \leq \langle z, \theta^*\rangle - \delta\alpha\right\} \leq \exp\left(-\frac{\delta^2\alpha}{2\gamma\nu}\right) \tag{17}$$

*Proof.* Using steps similar to the previous lemmas, we obtain

$$\mathbb{P}\left\{\langle z,\widehat{\theta}\rangle \leq \langle z,\theta^*\rangle - \delta\alpha\right\} \leq \frac{\mathbb{E}[\exp(c\, z^T \mathbf{V}^{-1}\left(\sum_t r_t x_t\right))]}{\exp(c\left(\langle z,\theta^*\rangle - \delta\alpha\right))}$$

$$\leq \exp\left(c\alpha\delta + c\langle z,\theta^*\rangle + \sum_{t=1}^{s}\frac{\langle x_t,\theta^*\rangle}{\nu}\left(e^{c\nu z^T \mathbf{V}^{-1}x_t}-1\right)\right)$$

($r_t$ are sub-poisson and independent)

Substituting $c = \frac{\log(1-\delta)}{\nu\gamma}$ and simplifing we get

$$\mathbb{P}\left\{\langle z,\widehat{\theta}\rangle \leq \langle z,\theta^*\rangle - \delta\alpha\right\} \leq \exp\left(-\frac{\langle z,\theta^*\rangle}{\nu\gamma}\left(\log\left(1-\delta\right)+\delta\right)+\frac{\alpha}{\nu\gamma}\delta\log\left(1-\delta\right)\right)$$

Note that since $\log(1-\delta) + \delta$ is negative, we can upper bound the above expression by replacing $\langle z,\theta^*\rangle$ with $\alpha$.

$$\mathbb{P}\left\{\langle z,\widehat{\theta}\rangle \leq \langle z,\theta^*\rangle - \delta\alpha\right\} \leq \exp\left(-\frac{\alpha}{\nu\gamma}\left(\log\left(1-\delta\right)+\delta-\delta\log\left(1-\delta\right)\right)\right)$$

$$\leq \exp\left(-\frac{\delta^2\alpha}{2\nu\gamma}\right). \qquad \text{(since } (1-\delta)\log(1-\delta) \geq -\delta + \frac{\delta^2}{2}\text{)}$$

Hence, the lemma stands proved. $\qquad\square$

## C   Regret Analysis of Algorithm 2

We will first define events $E_1$ and $E_2$ for each phase of the algorithm and show that they hold with high probability. We will use the events in the regret analysis.

- Event $E_1$: At the end of Part I, let $\widehat{\theta}$ be the unbiased estimator of $\theta^*$ and $\widetilde{\mathsf{T}}$ be as defined in Algorithm 2. All arms $x \in \mathcal{X}$ with $\langle x,\theta^*\rangle < 10\sqrt{\frac{d\nu\log(\mathsf{T}|\mathcal{X}|)}{\mathsf{T}}}$ satisfy

$$\langle x,\widehat{\theta}\rangle \leq 20\sqrt{\frac{d\nu\log(\mathsf{T}|\mathcal{X}|)}{\mathsf{T}}} \qquad (18)$$

  In addition, all arms $x \in \mathcal{X}$ with $\langle x,\theta^*\rangle \geq 10\sqrt{\frac{d\nu\log(\mathsf{T}|\mathcal{X}|)}{\mathsf{T}}}$ satisfy

$$|\langle x,\theta^*\rangle - \langle x,\widehat{\theta}\rangle| \leq 3\sqrt{\frac{d\nu\langle x,\theta^*\rangle\log(\mathsf{T}|\mathcal{X}|)}{\widetilde{\mathsf{T}}}} \quad \text{and} \qquad (19)$$

$$\frac{1}{2}\langle x,\theta^*\rangle \leq \langle x,\widehat{\theta}\rangle \leq \frac{4}{3}\langle x,\theta^*\rangle. \qquad (20)$$

- Event $E_2$: Let $\widetilde{\mathcal{X}}$ denote the surviving set of arms at the start of a phase in Part II, and $\mathsf{T}'$ be as defined in Algorithm 2. For all phases and for all $x \in \widetilde{\mathcal{X}}$ such that $\langle x,\theta^*\rangle \geq 10\sqrt{\frac{d\nu\log(\mathsf{T}|\mathcal{X}|)}{\mathsf{T}}}$, the estimator $\widehat{\theta}$ (calculated at the end of a phase) satisfies

$$|\langle x,\theta^*\rangle - \langle x,\widehat{\theta}\rangle| \leq 3\sqrt{\frac{d\nu\langle x,\theta^*\rangle\log(\mathsf{T}|\mathcal{X}|)}{\mathsf{T}'}} \quad \text{and} \qquad (21)$$

$$\frac{1}{2}\langle x,\theta^*\rangle \leq \langle x,\widehat{\theta}\rangle \leq \frac{4}{3}\langle x,\theta^*\rangle. \qquad (22)$$

### C.1   Supporting Lemmas

**Lemma 12** (Chernoff Bound). *Let $Z_1,\ldots,Z_n$ be independent Bernoulli random variables. Consider the sum $S = \sum_{r=1}^{n}Z_r$ and let $\mu = \mathbb{E}[S]$ be its expected value. Then, for any $\varepsilon \in [0,1]$, we have*

$$\mathbb{P}\left\{S \leq (1-\varepsilon)\mu\right\} \leq \exp\left(-\frac{\mu\varepsilon^2}{2}\right).$$

**Lemma 13.** *During Part* I, *arms from D-optimal design are added to $S$ at least $\widetilde{\mathsf{T}}/3$ times with probability greater than $1 - \frac{1}{\mathsf{T}}$.*

*Proof.* We use Lemma 12 with $Z_i$ as indicator random variables that take value one when an arm from $\mathcal{A}$ (the support of $\lambda$ in the optimal design) is chosen. By setting $\varepsilon = \frac{1}{3}$ and $\mu = \frac{\widetilde{\mathsf{T}}}{2}$, we obtain the required probability bound. $\square$

**Lemma 14.** *Using the notations in Algorithm 1, if the event in Lemma 13 holds, then for each $x \in \mathcal{X}$ and each round $t$ in Part* I *of the algorithm, we have*

$$x^T \mathbf{V}^{-1} X_t \le \frac{3d}{\widetilde{\mathsf{T}}},$$

*where $X_t$ is the arm pulled in round $t$.*

*Proof.* Let $\mathbf{U}(\lambda)$ and $\lambda$ denote the optimal design matrix (as defined in (4)) and the solution to the D-optimal design problem in Algorithm 1, respectively. That is, $\lambda$ is the solution of the optimization problem stated in equation (5) and $\mathbf{U}(\lambda) = \sum_{x \in \mathcal{X}} \lambda_x x x^T$. Lemma 3 implies that $\|x\|_{\mathbf{U}(\lambda)^{-1}} \le \sqrt{d}$ for all $x \in \mathcal{X}$.

Next, note that the construction of the sequence $\mathcal{S}$ in Part I (Subroutine `GenerateArmSequence`) and the event specified in Lemma 13 give us $\mathbf{V} \succ \frac{\widetilde{\mathsf{T}}}{3} \mathbf{U}(\lambda)$. Hence,

$$
\begin{aligned}
x^T \mathbf{V}^{-1} X_t &\le \|x\|_{\mathbf{V}^{-1}} \left\| \mathbf{V}^{-1} X_t \right\|_{\mathbf{V}} && \text{(via Hölder's inequality)} \\
&= \|x\|_{\mathbf{V}^{-1}} \|X_t\|_{\mathbf{V}^{-1}} \\
&\le \|x\|_{\left(\frac{\widetilde{\mathsf{T}}}{3} \mathbf{U}(\lambda)\right)^{-1}} \|X_t\|_{\left(\frac{\widetilde{\mathsf{T}}}{3} \mathbf{U}(\lambda)\right)^{-1}} && \left(\text{since } \mathbf{V} \succ \frac{\widetilde{\mathsf{T}}}{3} \mathbf{U}(\lambda)\right) \\
&= \sqrt{\frac{3}{\widetilde{\mathsf{T}}}} \|x\|_{\mathbf{U}(\lambda)^{-1}} \sqrt{\frac{3}{\widetilde{\mathsf{T}}}} \|X_t\|_{\mathbf{U}(\lambda)^{-1}} \\
&\le \sqrt{\frac{3d}{\widetilde{\mathsf{T}}}} \sqrt{\frac{3d}{\widetilde{\mathsf{T}}}} && \text{(by Lemma 3)} \\
&= \frac{3d}{\widetilde{\mathsf{T}}}.
\end{aligned}
$$

$\square$

The next lemma lower bounds the probability of event $E_1$ (see equations (18), (19), and (20)).

**Lemma 15.** *Event $E_1$ holds with probability at least $1 - \frac{6}{T}$.*

*Proof.* First, consider all arms $x \in \mathcal{X}$ for which $\langle x, \theta^* \rangle < 10\sqrt{\frac{d\nu \log (\mathsf{T}|\mathcal{X}|)}{\mathsf{T}}}$. Here, we invoke Lemma 10, with $\gamma = \frac{3d}{\widetilde{\mathsf{T}}}$ (as derived in Lemma 14), $\alpha = 10\sqrt{\frac{d\nu \log (\mathsf{T}|\mathcal{X}|)}{\mathsf{T}}}$, and $\delta = 1$, to obtain

$$
\begin{aligned}
\mathbb{P}\left\{ \langle x, \widehat{\theta} \rangle \le 20\sqrt{\frac{d\nu \log (\mathsf{T}|\mathcal{X}|)}{\mathsf{T}}} \right\} &\le \exp\left( -\frac{\delta^2 \alpha}{3\gamma \nu} \right) \\
&\le \exp\left( -\frac{10\sqrt{\frac{d\nu \log (\mathsf{T}|\mathcal{X}|)}{\mathsf{T}}} 3\sqrt{T d\nu \log(\mathsf{T}|\mathcal{X}|)}}{3\nu d} \right) \\
&\le \frac{1}{\mathsf{T}|\mathcal{X}|} && (23)
\end{aligned}
$$

Next, we consider arms $x \in \mathcal{X}$ such that $\langle x, \theta^* \rangle \ge 10\sqrt{\frac{d\nu \log (\mathsf{T}|\mathcal{X}|)}{\mathsf{T}}}$ and for such arms establish equations (19) and (20). Towards this, we invoke Lemma 6, with parameters $\gamma = \frac{3d}{\widetilde{\mathsf{T}}}$ and $\delta =$

$3\sqrt{\frac{d\nu \log(\mathsf{T}|\mathcal{X}|)}{\langle x,\theta^*\rangle\widetilde{\mathsf{T}}}}$. It is relevant to note that here $\delta \in [0,1]$ – this containment follows from the condition $\langle x,\theta^*\rangle \geq 10\sqrt{\frac{d\nu \log(\mathsf{T}|\mathcal{X}|)}{\mathsf{T}}}$ and $\widetilde{\mathsf{T}} = 3\sqrt{\mathsf{T}d\nu \log(\mathsf{T}|\mathcal{X}|)}$. Therefore,

$$\mathbb{P}\left\{|\langle x,\theta^*\rangle - \langle x,\widehat{\theta}\rangle| \geq 3\sqrt{\frac{d\nu\langle x,\theta^*\rangle \log(\mathsf{T}|\mathcal{X}|)}{\widetilde{\mathsf{T}}}}\right\} = \mathbb{P}\left\{|\langle x,\theta^*\rangle - \langle x,\widehat{\theta}\rangle| \geq \delta\langle x,\theta^*\rangle\right\}$$

$$\left(\text{since } \delta = 3\sqrt{\tfrac{d\nu \log(\mathsf{T}|\mathcal{X}|)}{\langle x,\theta^*\rangle\widetilde{\mathsf{T}}}}\right)$$

$$\leq 2\exp\left(-\frac{\frac{9d\nu \log(\mathsf{T}|\mathcal{X}|)}{\langle x,\theta^*\rangle\widetilde{\mathsf{T}}}\,\langle x,\theta^*\rangle}{3\nu\frac{3d}{\widetilde{\mathsf{T}}}}\right)$$

$$\text{(Lemma 6)}$$

$$= \frac{2}{\mathsf{T}|\mathcal{X}|} \tag{24}$$

For establishing equation (20), we invoke Lemma 6 again, now with $\gamma = \frac{3d}{\widetilde{\mathsf{T}}}$ and $\delta = \frac{1}{3}$:

$$\mathbb{P}\left\{\langle x,\widehat{\theta}\rangle \geq \frac{4}{3}\langle x,\theta^*\rangle\right\} \leq \exp\left(-\frac{3\sqrt{\mathsf{T}\nu d \log(\mathsf{T}|\mathcal{X}|)}\,\langle x,\theta^*\rangle}{27\nu d}\right)$$

$$\leq \exp\left(-\frac{3\sqrt{\mathsf{T}\nu d \log(\mathsf{T}|\mathcal{X}|)}\,10\sqrt{\frac{d\nu \log(\mathsf{T}|\mathcal{X}|)}{\mathsf{T}}}}{27\nu d}\right)$$

$$\leq \frac{1}{\mathsf{T}|\mathcal{X}|} \tag{25}$$

Similarly, with $\delta = \frac{1}{2}$, Lemma 6 gives us

$$\mathbb{P}\left\{\langle x,\widehat{\theta}\rangle \leq \frac{1}{2}\langle x,\theta^*\rangle\right\} \leq \frac{1}{\mathsf{T}|\mathcal{X}|} \tag{26}$$

Finally, we combine (23), (24), (25) and (26), and apply a union bound over all arms in $\mathcal{X}$. Then, conditioning on the event in Lemma 13 leads to the stated probability bound. The lemma stands proved. $\qquad\square$

The next lemma shows that event $E_2$ (see equations (21) and (22)) holds with high probability

**Lemma 16.** *Event $E_2$ holds with probability at least $1 - \frac{3\log\mathsf{T}}{T}$.*

*Proof.* Consider any phase in Part II and let $\mathbf{U}(\lambda)$ be the optimal design matrix obtained after solving the D-optimal design problem at the start of the phase. By Lemma 3, for all $x,z \in \widetilde{\mathcal{X}}$ we have

$$z^T\mathbf{V}^{-1}x \leq \|z\|_{\mathbf{V}^{-1}}\left\|\mathbf{V}^{-1}x\right\|_{\mathbf{V}} \qquad \text{(via Hölder's inequality)}$$

$$\leq \|z\|_{\mathbf{V}^{-1}}\|x\|_{\mathbf{V}^{-1}}$$

$$\leq \sqrt{\frac{d}{\mathsf{T}'}}\sqrt{\frac{d}{\mathsf{T}'}} = \frac{d}{\mathsf{T}'}$$

First, we address equation (21). In particular, we instantiate Lemma 6 with $\delta = 3\sqrt{\frac{d\nu \log(\mathsf{T}|\mathcal{X}|)}{\langle x,\theta^*\rangle\mathsf{T}'}}$ and $\gamma = \frac{d}{\mathsf{T}'}$. Note that given the lower bound on $\langle x,\theta^*\rangle$ and the inequality $\mathsf{T}' \geq 2\sqrt{\mathsf{T}d\nu \log(\mathsf{T}|\mathcal{X}|)}$ ensure that $\delta$ lies in $[0,1]$. Hence, substituting these values of $\delta$ and $\gamma$ in Lemma 6, we obtain

$$\mathbb{P}\left\{|\langle x,\theta^*\rangle - \langle x,\widehat{\theta}\rangle| \geq 3\sqrt{\frac{d\nu\langle x,\theta^*\rangle \log(\mathsf{T}|\mathcal{X}|)}{\mathsf{T}'}}\right\} \leq 2\exp\left(-\frac{\frac{9d\nu \log(\mathsf{T}|\mathcal{X}|)}{\langle x,\theta^*\rangle\mathsf{T}'}\cdot\langle x,\theta^*\rangle}{3\frac{d\nu}{\mathsf{T}'}}\right)$$

$$\leq \frac{2}{(\mathsf{T}|\mathcal{X}|)^3}$$

Next, following a similar approach as in the proof of Lemma 15, we use Lemma 6 with $\delta = \frac{1}{3}$ and $\delta = \frac{1}{2}$ to establish the upper and lower bounds of equation (22), respectively. Applying a union bound across arms in $\widetilde{\mathcal{X}}$ and over all—at most $\log \mathsf{T}$—phases, we obtain the desired probability bound of $1 - \frac{3 \log \mathsf{T}}{\mathsf{T}}$. $\qquad\square$

**Corollary 17.**

$$\mathbb{P}\left\{E_1 \cap E_2\right\} \geq 1 - \frac{4 \log \mathsf{T}}{\mathsf{T}}.$$

*Proof.* From Lemma 15 we have $\mathbb{P}\left\{E_1\right\} \geq 1 - \frac{6}{\mathsf{T}}$. Furthermore, from Lemma 16 we have $\mathbb{P}\left\{E_2\right\} \geq 1 - \frac{3 \log \mathsf{T}}{\mathsf{T}}$. Applying a union bound on the complements of these two events establishes the corollary. $\qquad\square$

**Lemma 18.** *Consider any bandit instance with $\langle x^*, \theta^* \rangle \geq 192 \sqrt{\frac{d\nu \log (\mathsf{T}|\mathcal{X}|)}{\mathsf{T}}}$. If event $E_1$ holds, then any arm with mean $\langle x, \theta^* \rangle \leq 10 \sqrt{\frac{d\nu \log(\mathsf{T}|\mathcal{X}|)}{\mathsf{T}}}$ is eliminated after Part I of Algorithm 2.*

*Proof.* We will show that in the given bandit instance and under the event $E_1$, for each arm $x \in \mathcal{X}$ with mean $\langle x, \theta^* \rangle \leq 10 \sqrt{\frac{d\nu \log(\mathsf{T}|\mathcal{X}|)}{\mathsf{T}}}$ the upper Nash confidence bound (see equation (7)) is less than the lower confidence bound of the optimal arm $x^*$. Hence, all such arms $x$ are eliminated from consideration in Line 8 of Algorithm 2. This will establish the lemma.

The upper Nash confidence bound of arm $x$ at the end of Part I is defined as

$$\mathrm{UNCB}\left(x, \widehat{\theta}, \widetilde{\mathsf{T}}/3\right) = \langle x, \widehat{\theta} \rangle + 6 \sqrt{\frac{3\langle x, \widehat{\theta}\rangle\, d\,\nu\,\log (\mathsf{T}|\mathcal{X}|)}{\widetilde{\mathsf{T}}}}$$

$$\leq 20 \sqrt{\frac{d\,\nu\,\log(\mathsf{T}|\mathcal{X}|)}{\mathsf{T}}} + 6\sqrt{\frac{3\langle x, \widehat{\theta}\rangle\, d\,\nu\,\log (\mathsf{T}|\mathcal{X}|)}{\widetilde{\mathsf{T}}}} \qquad \text{(via event } E_1\text{)}$$

$$\leq 20 \sqrt{\frac{d\nu \log(\mathsf{T}|\mathcal{X}|)}{\mathsf{T}}} + 6\sqrt{\frac{3 \cdot 20 \sqrt{\frac{d\nu \log(\mathsf{T}|\mathcal{X}|)}{\mathsf{T}}} d\nu \log (\mathsf{T}|\mathcal{X}|)}{3\sqrt{\mathsf{T}\nu d \log(\mathsf{T}|\mathcal{X}|)}}}$$

$$\text{(substituting } \widetilde{\mathsf{T}}\text{)}$$

$$\leq 47 \sqrt{\frac{d\nu \log(\mathsf{T}|\mathcal{X}|)}{\mathsf{T}}} \qquad\qquad\qquad (27)$$

In the given bandit instance and under event $E_1$, for the optimal arm $x^*$, we have

$$\langle x^*, \widehat{\theta} \rangle \leq \langle x^*, \theta^* \rangle + 3 \sqrt{\frac{d\nu \langle x^*, \theta^* \rangle \log (\mathsf{T}|\mathcal{X}|)}{\widetilde{\mathsf{T}}}}$$

$$= \langle x^*, \theta^* \rangle \left(1 + 3 \sqrt{\frac{d\nu \log (\mathsf{T}|\mathcal{X}|)}{\langle x^*, \theta^* \rangle 3 \sqrt{\mathsf{T} d\,\nu\, \log(\mathsf{T}|\mathcal{X}|)}}}\right) \qquad \text{(substituting } \widetilde{\mathsf{T}}\text{)}$$

$$\leq \langle x^*, \theta^* \rangle \left(1 + 3 \sqrt{\frac{d\nu \log (\mathsf{T}|\mathcal{X}|)}{192 \sqrt{\frac{d\nu \log(\mathsf{T}|\mathcal{X}|)}{\mathsf{T}}} 3 \sqrt{\mathsf{T}\nu d \log(\mathsf{T}|\mathcal{X}|)}}}\right)$$

$$\text{(using } \langle x^*, \theta^* \rangle \geq 192 \sqrt{\frac{d\nu \log (\mathsf{T}|\mathcal{X}|)}{\mathsf{T}}}\text{)}$$

$$= \frac{17}{16}\langle x^*, \theta^* \rangle. \qquad\qquad\qquad (28)$$

Therefore, the lower Nash confidence bound of $x^*$ satisfies

$$\mathrm{LNCB}\left(x^*, \widehat{\theta}, \widetilde{\mathsf{T}}/3\right) = \langle x^*, \widehat{\theta} \rangle - 6 \sqrt{\frac{3\langle x^*, \widehat{\theta}\rangle\, d\,\nu\, \log (\mathsf{T}|\mathcal{X}|)}{\widetilde{\mathsf{T}}}}$$

$$\geq \langle x^*, \theta^* \rangle - 3\sqrt{\frac{d\,\nu\,\langle x^*, \theta^* \rangle \log\left(\mathsf{T}|\mathcal{X}|\right)}{\widetilde{\mathsf{T}}}} - 6\sqrt{\frac{3\langle x^*, \widehat{\theta} \rangle\,d\,\nu\,\log\left(\mathsf{T}|\mathcal{X}|\right)}{\widetilde{\mathsf{T}}}}$$

$$\text{(via (19) in event } E_1\text{)}$$

$$\geq \langle x^*, \theta^* \rangle - \left(3 + 6\sqrt{\frac{51}{16}}\right)\sqrt{\frac{d\,\nu\,\langle x^*, \theta^* \rangle \log\left(\mathsf{T}|\mathcal{X}|\right)}{\widetilde{\mathsf{T}}}}$$

$$\text{(since } \langle x^*, \widehat{\theta} \rangle \leq \tfrac{17}{16}\langle x^*, \theta^* \rangle \text{ via (28))}$$

$$\geq \langle x^*, \theta^* \rangle \left(1 - 14\sqrt{\frac{d\nu\,\log\left(\mathsf{T}|\mathcal{X}|\right)}{\langle x^*, \theta^* \rangle \widetilde{\mathsf{T}}}}\right)$$

$$\geq \langle x^*, \theta^* \rangle \left(1 - 14\sqrt{\frac{d\nu\,\log\left(\mathsf{T}|\mathcal{X}|\right)}{192\sqrt{\frac{d\nu\,\log(\mathsf{T}|\mathcal{X}|)}{\mathsf{T}}}\,3\sqrt{\mathsf{T}d\nu\log(\mathsf{T}|\mathcal{X}|)}}}\right)$$

$$\geq \frac{5}{12}\langle x^*, \theta^* \rangle$$

$$\geq 80\sqrt{\frac{d\nu\log(\mathsf{T}|\mathcal{X}|)}{\mathsf{T}}} \tag{29}$$

Equations (29) and (27) imply

$$\mathrm{UNCB}\left(x, \widehat{\theta}, \widetilde{\mathsf{T}}/3\right) < \mathrm{LNCB}\left(x^*, \widehat{\theta}, \widetilde{\mathsf{T}}/3\right) \tag{30}$$

As mentioned previously, Line 8 in Algorithm 2 eliminates all arms $x$ that satisfy inequality (30). Hence, the lemma stands proved $\qquad\square$

## C.2 Proofs of Lemmas 7 and 8

**Lemma 7.** *Consider any bandit instance in which for the optimal arm $x^* \in \mathcal{X}$ we have $\langle x^*, \theta^* \rangle \geq 192\sqrt{\frac{d\,\nu}{\mathsf{T}}}\log(\mathsf{T}|\mathcal{X}|)$. Then, with probability at least $\left(1 - \frac{4\log \mathsf{T}}{\mathsf{T}}\right)$, the optimal arm $x^*$ always exists in the surviving set $\widetilde{\mathcal{X}}$ in Part I and in every phase in Part II of Algorithm 2.*

*Proof.* We will show that, under events $E_1$ and $E_2$, throughout the execution of Algorithm 2 the UNCB of the optimal arm $x^*$ is never less than the LNCB of any arm $x$. Hence, then the optimal arm $x^*$ never satisfies the elimination criterion in Algorithm 2 and, hence, $x^*$ always exists in the surviving set of arms.

First, we consider arms $x$ with the property that $\langle x, \theta^* \rangle < 10\sqrt{\frac{d\nu\log\left(\mathsf{T}|\mathcal{X}|\right)}{\mathsf{T}}}$. For any such arm $x$, at the end of Part I of the algorithm we have

$$\mathrm{LNCB}\left(x, \widehat{\theta}, \widetilde{\mathsf{T}}/3\right) \leq \mathrm{UNCB}\left(x, \widehat{\theta}, \widetilde{\mathsf{T}}/3\right) \underset{\text{via (30)}}{<} \mathrm{LNCB}\left(x^*, \widehat{\theta}, \widetilde{\mathsf{T}}/3\right) \leq \mathrm{UNCB}\left(x^*, \widehat{\theta}, \widetilde{\mathsf{T}}/3\right).$$

Hence, at the end of Part I, arm $x^*$ is not eliminated via the LNCB of any $x$ which satisfies $\langle x, \theta^* \rangle < 10\sqrt{\frac{d\nu\log\left(\mathsf{T}|\mathcal{X}|\right)}{\mathsf{T}}}$. Further, note that, under event $E_1$, such arms are eliminated at the end of Part I (Lemma 18). Hence, the LNCB of such arms are not even considered in the phases of Part II.

To complete the proof, we next show that the UNCB of the optimal arm $x^*$ is at least the LNCB of all arms $x$ which bear $\langle x, \theta^* \rangle \geq 10\sqrt{\frac{d\nu\log\left(\mathsf{T}|\mathcal{X}|\right)}{\mathsf{T}}}$. Below, we will consider the Nash confidence bounds for a general $\mathsf{T}'$. Replacing $\mathsf{T}'$ by $\widetilde{\mathsf{T}}$ gives us the desired confidence-bounds comparison for the end of Part I – this repetition is omitted.

Under events $E_1$ and $E_2$, for any arm $x$ with $\langle x, \theta^* \rangle \geq 10\sqrt{\frac{d\nu \log (\mathsf{T}|\mathcal{X}|)}{\mathsf{T}}}$, it holds that

$$\mathrm{LNCB}(x, \widehat{\theta}, \mathsf{T}') = \langle x, \widehat{\theta} \rangle - 6\sqrt{\frac{\langle x, \widehat{\theta} \rangle \, d \, \nu \, \log (\mathsf{T}|\mathcal{X}|)}{\mathsf{T}'}}$$

$$\leq \langle x, \theta^* \rangle + 3\sqrt{\frac{d \, \nu \, \langle x, \theta^* \rangle \log (\mathsf{T}|\mathcal{X}|)}{\mathsf{T}'}} - 6\sqrt{\frac{\langle x, \widehat{\theta} \rangle \, d\nu \, \log (\mathsf{T}|\mathcal{X}|)}{\mathsf{T}'}} \quad \text{(via (21))}$$

$$\leq \langle x, \theta^* \rangle - \left( \frac{6}{\sqrt{2}} - 3 \right) \sqrt{\frac{d\nu \langle x, \theta^* \rangle \log (\mathsf{T}|\mathcal{X}|)}{\mathsf{T}'}} \quad (\langle x, \widehat{\theta} \rangle \geq \tfrac{1}{2} \langle x, \theta^* \rangle \text{ via (22)})$$

$$\leq \langle x, \theta^* \rangle. \tag{31}$$

Complementarily, for optimal arm $x^*$ we have

$$\mathrm{UNCB}(x^*, \widehat{\theta}, \mathsf{T}') = \langle x^*, \widehat{\theta} \rangle + 6\sqrt{\frac{\langle x^*, \widehat{\theta} \rangle \, d \, \nu \, \log (\mathsf{T}|\mathcal{X}|)}{\mathsf{T}'}}$$

$$\geq \langle x^*, \theta^* \rangle - 3\sqrt{\frac{d\nu \langle x^*, \theta^* \rangle \log (\mathsf{T}|\mathcal{X}|)}{\mathsf{T}'}} + 6\sqrt{\frac{\langle x^*, \widehat{\theta} \rangle \, d \, \nu \, \log (\mathsf{T}|\mathcal{X}|)}{\mathsf{T}'}}$$

$$\geq \langle x^*, \theta^* \rangle + \left( \frac{6}{\sqrt{2}} - 3 \right) \sqrt{\frac{d\nu \, \langle x^*, \theta^* \rangle \log (\mathsf{T}|\mathcal{X}|)}{\mathsf{T}'}} \quad (\text{since } \langle x^*, \widehat{\theta} \rangle \geq \tfrac{\langle x^*, \theta^* \rangle}{2})$$

$$\geq \langle x^*, \theta^* \rangle \tag{32}$$

Since $\langle x^*, \theta^* \rangle \geq \langle x, \theta^* \rangle$ for all arms $x$, inequalities (31) and (32) lead to the confidence-bounds comparison:

$$\mathrm{UNCB}(x^*, \widehat{\theta}, \mathsf{T}') \geq \mathrm{LNCB}(x, \widehat{\theta}, \mathsf{T}').$$

Hence, if events $E_1$ and $E_2$ hold, then the optimal arm $x^*$ is never eliminated from Algorithm 2. Further, Corollary 17 ensures that the events $E_1$ and $E_2$ hold with probability at least $1 - \frac{4 \log \mathsf{T}}{\mathsf{T}}$. Hence, the lemma stands proved. $\square$

**Lemma 8.** *Consider any phase $\ell$ in Part* II *of Algorithm 2 and let $\widetilde{\mathcal{X}}$ be the surviving set of arms at the beginning of that phase. Then, with $\widetilde{\mathsf{T}} = \sqrt{d\nu \mathsf{T} \log(\mathsf{T}|\mathcal{X}|)}$, we have*

$$\Pr \left\{ \langle x, \theta^* \rangle \geq \langle x^*, \theta^* \rangle - 25\sqrt{\frac{3d\nu \langle x^*, \theta^* \rangle \log (\mathsf{T}|\mathcal{X}|)}{2^{\ell} \cdot \widetilde{\mathsf{T}}}} \text{ for all } x \in \widetilde{\mathcal{X}} \right\} \geq 1 - \frac{4 \log \mathsf{T}}{\mathsf{T}} \tag{10}$$

*Here, $\nu$ is the sub-Poisson parameter of the stochastic rewards.*

*Proof.* For the analysis, assume that events $E_1$ and $E_2$ hold. Lemma 7 ensures that the optimal arm is contained in the surviving set of arms $\widetilde{\mathcal{X}}$. Furthermore, if an arm $x \in \widetilde{\mathcal{X}}$ at the beginning of the $\ell^{\text{th}}$ phase, then it must be the case that arm $x$ was not eliminated in the previous phase (which executed for $\mathsf{T}'/2$ rounds); in particular, we have $\mathrm{UNCB}(x, \widehat{\theta}, \mathsf{T}'/2) \geq \mathrm{LNCB}(x^*, \widehat{\theta}, \mathsf{T}'/2)$. This inequality reduces to

$$\langle x, \widehat{\theta} \rangle + 6\sqrt{\frac{\langle x, \widehat{\theta} \rangle \, d \, \nu \, \log (\mathsf{T}|\mathcal{X}|)}{\frac{\mathsf{T}'}{2}}} \geq \langle x^*, \widehat{\theta} \rangle - 6\sqrt{\frac{\langle x^*, \widehat{\theta} \rangle \, d \, \nu \, \log (\mathsf{T}|\mathcal{X}|)}{\frac{\mathsf{T}'}{2}}}.$$

Rearranging the terms, we obtain

$$\langle x, \widehat{\theta} \rangle \geq \langle x^*, \widehat{\theta} \rangle - 6\sqrt{\frac{\langle x^*, \widehat{\theta} \rangle \, d \, \nu \, \log (\mathsf{T}|\mathcal{X}|)}{\frac{\mathsf{T}'}{2}}} - 6\sqrt{\frac{\langle x, \widehat{\theta} \rangle \, d \, \nu \, \log (\mathsf{T}|\mathcal{X}|)}{\frac{\mathsf{T}'}{2}}}$$

$$\geq \langle x^*, \widehat{\theta} \rangle - 6\sqrt{\frac{4\langle x^*, \theta^* \rangle \, d \, \nu \, \log (\mathsf{T}|\mathcal{X}|)}{3\mathsf{T}'/2}} - 6\sqrt{\frac{4\langle x, \theta^* \rangle \, d \, \nu \log (\mathsf{T}|\mathcal{X}|)}{3\mathsf{T}'/2}}$$

$$\quad (\langle x, \widehat{\theta} \rangle \leq \tfrac{4}{3} \langle x, \theta^* \rangle \text{ via (22)})$$

$$\geq \langle x^*, \widehat{\theta} \rangle - 20\sqrt{\frac{\langle x^*, \theta^* \rangle \, d \, \nu \, \log (\mathsf{T}|\mathcal{X}|)}{\mathsf{T}'}}.$$

Further, invoking equation (21) for $x^*$ leads to

$$\langle x, \theta^* \rangle \geq \langle x^*, \theta^* \rangle - 20\sqrt{\frac{\langle x^*, \theta^* \rangle \, d \, \nu \, \log(\mathsf{T}|\mathcal{X}|)}{\mathsf{T}'}} - 3\sqrt{\frac{\langle x^*, \theta^* \rangle \, d \, \nu \, \log(\mathsf{T}|\mathcal{X}|)}{\frac{\mathsf{T}'}{2}}}$$

$$\geq \langle x^*, \theta^* \rangle - 25\sqrt{\frac{\langle x^*, \theta^* \rangle \, d \, \nu \, \log(\mathsf{T}|\mathcal{X}|)}{\mathsf{T}'}}.$$

Substituting $\mathsf{T}' = 2^\ell \widetilde{\mathsf{T}}/3$, the above inequality reduces to the desired bound in (10). From Corollary 17, we have that the events $E_1$ and $E_2$ hold with probability at least $1 - \frac{4\log \mathsf{T}}{\mathsf{T}}$. Hence, the lemma stands proved. $\square$

## C.3 Proof of Theorem 1

**Theorem 1.** *For any given stochastic linear bandits problem with (finite) set of arms $\mathcal{X} \subset \mathbb{R}^d$, time horizon $\mathsf{T} \in \mathbb{Z}_+$, and $\nu$-sub-Poisson rewards, Algorithm 2 achieves Nash regret*

$$\mathrm{NR}_\mathsf{T} = O\left(\beta \sqrt{\frac{d \, \nu}{\mathsf{T}}} \log(\mathsf{T}|\mathcal{X}|)\right).$$

*Here, $\beta = \max\{1, \langle x^*, \theta^* \rangle \log d\}$, with $x^* \in \mathcal{X}$ denoting the optimal arm and $\theta^*$ the (unknown) parameter vector.*

*Proof.* We will assume, without loss of generality, that $\langle x^*, \theta^* \rangle \geq 192\sqrt{\frac{d \, \nu}{\mathsf{T}}} \log(\mathsf{T}|\mathcal{X}|)$, otherwise the stated Nash Regret bound directly holds (see equation (1)). Write $E$ to denote the 'good' event identified in Lemma 8; the lemma ensures that $\mathbb{P}\{E\} \geq 1 - \frac{4\log \mathsf{T}}{\mathsf{T}}$.

During Part I of Algorithm 2, the product of expected rewards, conditioned on $E$, satisfies

$$\prod_{t=1}^{\widetilde{\mathsf{T}}} \mathbb{E}[\langle X_t, \theta^* \rangle \mid E]^{\frac{1}{\mathsf{T}}} \geq \left(\frac{\langle x^*, \theta^* \rangle}{2(d+1)}\right)^{\frac{\widetilde{\mathsf{T}}}{\mathsf{T}}} \quad \text{(via Lemma 5)}$$

$$= \langle x^*, \theta^* \rangle^{\frac{\widetilde{\mathsf{T}}}{\mathsf{T}}}\left(1 - \frac{1}{2}\right)^{\frac{\log(2(d+1))\widetilde{\mathsf{T}}}{\mathsf{T}}}$$

$$\geq \langle x^*, \theta^* \rangle^{\frac{\widetilde{\mathsf{T}}}{\mathsf{T}}}\left(1 - \frac{\log(2(d+1))\widetilde{\mathsf{T}}}{\mathsf{T}}\right).$$

For analyzing Part II, we will utilize Lemma 8. Write $\mathcal{B}_\ell$ to denote all the rounds $t$ that belong to $\ell^{\text{th}}$ phase (in Part II). Also, let $\mathsf{T}'_\ell$ denote the associated phase-length parameter, i.e., $\mathsf{T}'_\ell = 2^\ell \widetilde{\mathsf{T}}/3$. Note that in each phase $\ell$ (i.e., in the for-loop at Line 12 of Algorithm 2), every arm $a$ in $\mathsf{Supp}(\lambda)$ (the support of D-optimal design) is pulled $\lceil \lambda_a \mathsf{T}'_\ell \rceil$ times. Given that $|\mathsf{Supp}(\lambda)| \leq d(d+1)/2$, we have $|\mathcal{B}_\ell| \leq \mathsf{T}'_\ell + \frac{d(d+1)}{2}$. By construction $\mathsf{T}'_\ell \geq \frac{d(d+1)}{2}$ and, hence, $|\mathcal{B}_\ell| \leq 2\mathsf{T}'_\ell$. Since the phase length parameter, $\mathsf{T}'_\ell$, doubles after each phase, the algorithm would have at most $\log \mathsf{T}$ phases. Hence, the product of expected rewards in Part II satisfies

$$\prod_{t=\widetilde{\mathsf{T}}+1}^{\mathsf{T}} \mathbb{E}[\langle X_t, \theta^* \rangle \mid E]^{\frac{1}{\mathsf{T}}} = \prod_{\mathcal{B}_\ell} \prod_{t \in \mathcal{B}_\ell} \mathbb{E}[\langle X_t, \theta^* \rangle \mid E]^{\frac{1}{\mathsf{T}}}$$

$$\geq \prod_{\mathcal{B}_\ell}\left(\langle x^*, \theta^* \rangle - 25\sqrt{\frac{d \, \nu \, \langle x^*, \theta^* \rangle \log(\mathsf{T}|\mathcal{X}|)}{\mathsf{T}'_\ell}}\right)^{\frac{|\mathcal{B}_\ell|}{\mathsf{T}}} \quad \text{(Lemma 8)}$$

$$\geq \langle x^*, \theta^* \rangle^{\frac{\mathsf{T}-\widetilde{\mathsf{T}}}{\mathsf{T}}} \prod_{\ell=1}^{\log \mathsf{T}}\left(1 - 25\sqrt{\frac{d \, \nu \, \log(\mathsf{T}|\mathcal{X}|)}{\langle x^*, \theta^* \rangle \mathsf{T}'_\ell}}\right)^{\frac{|\mathcal{B}_\ell|}{\mathsf{T}}}$$

$$\geq \langle x^*, \theta^* \rangle^{\frac{\mathsf{T}-\widetilde{\mathsf{T}}}{\mathsf{T}}} \prod_{\ell=1}^{\log \mathsf{T}}\left(1 - 50\frac{|\mathcal{B}_\ell|}{\mathsf{T}}\sqrt{\frac{d \, \nu \, \log(\mathsf{T}|\mathcal{X}|)}{\langle x^*, \theta^* \rangle \mathsf{T}'_\ell}}\right).$$

The last inequality follows from the fact that $(1-x)^r \geq (1-2rx)$, for any $r \in [0,1]$ and $x \in [0,1/2]$. Note that the term $\sqrt{\frac{d\nu \log(\mathsf{T}|\mathcal{X}|)}{\langle x^*, \theta^*\rangle \mathsf{T}'_\ell}} \leq 1/2$, since $\langle x^*, \theta^*\rangle \geq 192\sqrt{\frac{d\nu}{\mathsf{T}}}\log(\mathsf{T}|\mathcal{X}|)$ along with $\mathsf{T}'_\ell \geq 2\sqrt{\mathsf{T}d\nu \log \mathsf{T}|\mathcal{X}|}$ and $\mathsf{T} \geq e^4$. We further simplify the expression as follows

$$\prod_{\ell=1}^{\log \mathsf{T}} \left(1 - 50\frac{|\mathcal{B}_\ell|}{\mathsf{T}}\sqrt{\frac{d\,\nu\,\log(\mathsf{T}|\mathcal{X}|)}{\langle x^*, \theta^*\rangle \mathsf{T}'_\ell}}\right) \geq \prod_{\ell=1}^{\log \mathsf{T}} \left(1 - 100\frac{\sqrt{\mathsf{T}'_\ell}}{\mathsf{T}}\sqrt{\frac{d\,\nu\,\log(\mathsf{T}|\mathcal{X}|)}{\langle x^*, \theta^*\rangle}}\right)$$

$$\text{(since } |\mathcal{B}_\ell| \leq 2\mathsf{T}'_\ell\text{)}$$

$$\geq 1 - \frac{100}{\mathsf{T}}\sqrt{\frac{d\,\nu\log(\mathsf{T}|\mathcal{X}|)}{\langle x^*, \theta^*\rangle}}\left(\sum_{\ell=1}^{\log \mathsf{T}} \sqrt{\mathsf{T}'_\ell}\right)$$

$$\text{(since } (1-a)(1-b) \geq 1 - a - b \quad \text{for } a, b \geq 0\text{)}$$

$$\geq 1 - \frac{100}{\mathsf{T}}\sqrt{\frac{d\,\nu\,\log(\mathsf{T}|\mathcal{X}|)}{\langle x^*, \theta^*\rangle}}\left(\sqrt{\mathsf{T}\log\mathsf{T}}\right)$$

$$\text{(via Cauchy-Schwarz inequality)}$$

$$\geq 1 - 100\sqrt{\frac{d\nu}{\mathsf{T}\langle x^*, \theta^*\rangle}}\log(\mathsf{T}|\mathcal{X}|).$$

Combining the lower bound for the expected rewards in the two parts we get

$$\prod_{t=1}^{\mathsf{T}} \mathbb{E}[\langle X_t, \theta^*\rangle]^{\frac{1}{\mathsf{T}}} \geq \prod_{t=1}^{\mathsf{T}} \left(\mathbb{E}[\langle X_t, \theta^*\rangle \mid E]\,\mathbb{P}\{E\}\right)^{\frac{1}{\mathsf{T}}}$$

$$\geq \langle x^*, \theta^*\rangle\left(1 - \frac{\log(2(d+1))\widetilde{\mathsf{T}}}{\mathsf{T}}\right)\left(1 - 100\sqrt{\frac{d\nu}{\mathsf{T}\langle x^*, \theta^*\rangle}}\log(\mathsf{T}|\mathcal{X}|)\right)\mathbb{P}\{E\}$$

$$\geq \langle x^*, \theta^*\rangle\left(1 - \frac{\log(2(d+1))\widetilde{\mathsf{T}}}{\mathsf{T}} - 100\sqrt{\frac{d\nu}{\mathsf{T}\langle x^*, \theta^*\rangle}}\log(\mathsf{T}|\mathcal{X}|)\right)\mathbb{P}\{E\}$$

$$\geq \langle x^*, \theta^*\rangle\left(1 - \frac{\log(2(d+1))\widetilde{\mathsf{T}}}{\mathsf{T}} - 100\sqrt{\frac{d\nu}{\mathsf{T}\langle x^*, \theta^*\rangle}}\log(\mathsf{T}|\mathcal{X}|)\right)\left(1 - \frac{4\log\mathsf{T}}{\mathsf{T}}\right)$$

$$\geq \langle x^*, \theta^*\rangle\left(1 - \frac{\log(2(d+1))3\sqrt{\mathsf{T}d\nu\log(\mathsf{T}|\mathcal{X}|)}}{\mathsf{T}} - 100\sqrt{\frac{d\nu}{\mathsf{T}\langle x^*, \theta^*\rangle}}\log(\mathsf{T}|\mathcal{X}|) - \frac{4\log\mathsf{T}}{\mathsf{T}}\right)$$

$$\geq \langle x^*, \theta^*\rangle - 100\sqrt{\frac{\langle x^*, \theta^*\rangle d\,\nu}{\mathsf{T}}}\log(\mathsf{T}|\mathcal{X}|) - 6\langle x^*, \theta^*\rangle\sqrt{\frac{d\,\nu\,\log(\mathsf{T}|\mathcal{X}|)}{\mathsf{T}}}\log(2(d+1)).$$

Therefore, the Nash Regret can be bounded as

$$\mathrm{NR}_{\mathsf{T}} = \langle x^*, \theta^*\rangle - \left(\prod_{t=1}^{\mathsf{T}} \mathbb{E}[\langle X_t, \theta^*\rangle]\right)^{1/\mathsf{T}}$$

$$\leq 100\sqrt{\frac{\langle x^*, \theta^*\rangle d\,\nu}{\mathsf{T}}}\log(\mathsf{T}|\mathcal{X}|) + 6\sqrt{\frac{d\,\nu\,\log(\mathsf{T}|\mathcal{X}|)}{\mathsf{T}}}\log(2(d+1))\langle x^*, \theta^*\rangle \qquad (33)$$

$$\leq \left(100\sqrt{\langle x^*, \theta^*\rangle} + 6\log(2(d+1))\langle x^*, \theta^*\rangle\right)\sqrt{\frac{d\nu}{\mathsf{T}}}\log(\mathsf{T}|\mathcal{X}|) \qquad (34)$$

Hence, with $\beta = \max\left\{1, \sqrt{\langle x^*, \theta^*\rangle}, \langle x^*, \theta^*\rangle \log d\right\} = \max\{1, \langle x^*, \theta^*\rangle \log d\}$, from equation (34) we obtain the desired bound on Nash regret $\mathrm{NR}_{\mathsf{T}} = O\left(\beta\sqrt{\frac{d\,\nu}{\mathsf{T}}}\log(\mathsf{T}|\mathcal{X}|)\right)$. The theorem stands proved. $\qquad \square$

## D  Algorithm 3 and Regret Analysis

**Algorithm 3** LINNASH (Nash Confidence Bound Algorithm for Infinite Set of Arms)

**Input:** Arm set $\mathcal{X}$ and horizon of play $T$.

1: Initialize matrix $\mathbf{V} \leftarrow [0]_{d,d}$ and number of rounds $\widetilde{T} = 3\sqrt{T d^{2.5} \nu \log(T)}$.
   Part I
2: Generate arm sequence $\mathcal{S}$ for the first $\widetilde{T}$ rounds using Algorithm 1.
3: **for** $t = 1$ to $\widetilde{T}$ **do**
4:     Pull the next arm $X_t$ from the sequence $S$.
5:     Observe reward $r_t$ and update $\mathbf{V} \leftarrow \mathbf{V} + X_t X_t^T$
6: **end for**
7: Set estimate $\widehat{\theta} := \mathbf{V}^{-1}\left(\sum_{t=1}^{\widetilde{T}} r_t X_t\right)$
8: Find $\gamma = \max_{z \in \mathcal{X}}\langle z, \widehat{\theta}\rangle$
9: Update $\widetilde{\mathcal{X}} \leftarrow \{x \in \mathcal{X} : \langle x, \widehat{\theta}\rangle \geq \gamma - 16\sqrt{\frac{3\,\gamma\,d^{\frac{5}{2}}\,\nu\,\log(T)}{\widetilde{T}}}\}$
10: $T' \leftarrow \frac{2}{3}\widetilde{T}$
    Part II
11: **while** end of time horizon $T$ is reached **do**
12:     Initialize $V = [0]_{d,d}$ to be an all zeros $d \times d$ matrix and $s = [0]_d$ to be an all-zeros vector.
        // Beginning of new phase.
13:     Find the probability distribution $\lambda \in \Delta(\widetilde{\mathcal{X}})$ by maximizing the following objective

$$\log \mathsf{Det}(\mathbf{V}(\lambda)) \text{ subject to } \lambda \in \Delta(\widetilde{\mathcal{X}}) \text{ and } \mathsf{Supp}(\lambda) \leq d(d+1)/2. \qquad (35)$$

14:     **for** each arm $a$ in $\mathsf{Supp}(\lambda)$ **do**
15:         Pull arm $a$ for the next $\lceil \lambda_a T' \rceil$ rounds.
16:         Observe rewards and Update $\mathbf{V} \leftarrow \mathbf{V} + \lceil \lambda_a T' \rceil \cdot aa^T$
17:         Observe $\lceil \lambda_a T' \rceil$ corresponding rewards $z_1, z_2, \ldots$ and update $s \leftarrow s + (\sum_j z_j)a$.
18:     **end for**
19:     Estimate $\widehat{\theta} = \mathbf{V}^{-1}\left(\sum_{t \in \mathcal{E}} r_t X_t\right)$
20:     Find $\gamma = \max_{z \in \mathcal{X}}\langle z, \widehat{\theta}\rangle$
21:     $\widetilde{\mathcal{X}} \leftarrow \{x \in \mathcal{X} : \langle x, \widehat{\theta}\rangle \geq \gamma - 16\sqrt{\frac{\gamma\,d^{\frac{5}{2}}\,\log(T)}{T'}}\}$
22:     $T' \leftarrow 2 \times T'$   // End of phase.
23: **end while**

---

Instead of ensuring probability bounds on individual arms, we construct a confidence ellipsoid around $\theta^*$. In the context of Algorithm 3, we define the following events for the regret analysis:

$G_1$  In Part I, arms from the D-optimal design are chosen at least $\widetilde{T}/3$ times. If $\langle x^*, \theta^*\rangle \geq 196\sqrt{\frac{d^{2.5}\nu}{\widetilde{T}}}\log T$, then $\widehat{\theta}$ calculated at the end of Part I satisfies

$$\left\|\widehat{\theta} - \theta^*\right\|_{\mathbf{V}} \leq 7\sqrt{\langle x^*, \theta^*\rangle d^{\frac{3}{2}}\nu \log T}.$$

$G_2$  In Part II, for every phase, if $\langle x^*, \theta^*\rangle \geq 196\sqrt{\frac{d^{2.5}\nu}{\widetilde{T}}}\log T$, the estimators $\widehat{\theta}$ satisfy:

$$\left\|\widehat{\theta} - \theta^*\right\|_{\mathbf{V}} \leq 7\sqrt{\langle x^*, \theta^*\rangle d^{\frac{3}{2}}\nu \log T}.$$

Without loss of generality, we assume throughout that $\langle x^*, \theta^*\rangle \geq 196\frac{d^{1.25}\sqrt{\nu}}{\sqrt{T}}\log T$. Otherwise, the regret bound in Theorem 2 trivially holds. Let $\mathcal{B}$ denote the unit ball in $\mathbb{R}^d$. We have

$$\left\|\widehat{\theta} - \theta^*\right\|_{\mathbf{V}} = \left\|\mathbf{V}^{\frac{1}{2}}(\widehat{\theta} - \theta^*)\right\|_2$$
$$= \max_{y \in \mathcal{B}}\langle y, \mathbf{V}^{\frac{1}{2}}(\widehat{\theta} - \theta^*)\rangle.$$

We construct an $\varepsilon$-net for the unit ball, denoted as $\mathcal{C}_\varepsilon$. For any $y \in \mathcal{B}$, we define $y_\varepsilon :=$ $\arg\min_{b \in \mathcal{C}_\varepsilon} \|b - y\|_2$. We can now write

$$
\begin{aligned}
\left\|\widehat{\theta} - \theta^*\right\|_{\mathbf{V}} &= \max_{y \in \mathcal{B}} \langle y - y_\varepsilon, \mathbf{V}^{\frac{1}{2}}(\widehat{\theta} - \theta^*)\rangle + \langle y_\varepsilon, \mathbf{V}^{\frac{1}{2}}(\widehat{\theta} - \theta^*)\rangle \\
&\leq \max_{y \in \mathcal{B}} \|y - y_\varepsilon\|_2 \left\|\mathbf{V}^{\frac{1}{2}}(\widehat{\theta} - \theta^*)\right\|_2 + |\langle y_\varepsilon, \mathbf{V}^{\frac{1}{2}}(\widehat{\theta} - \theta^*)\rangle| \\
&\leq \varepsilon \left\|(\widehat{\theta} - \theta^*)\right\|_{\mathbf{V}} + |\langle y_\varepsilon, \mathbf{V}^{\frac{1}{2}}(\widehat{\theta} - \theta^*)\rangle|.
\end{aligned}
$$

Rearranging, we obtain

$$
\left\|\widehat{\theta} - \theta^*\right\|_{\mathbf{V}} \leq \frac{1}{1 - \varepsilon} |\langle y_\varepsilon \mathbf{V}^{\frac{1}{2}}, \widehat{\theta} - \theta^*\rangle|. \tag{36}
$$

In the following lemmas, we show that $|\langle y_\varepsilon \mathbf{V}^{\frac{1}{2}}, \widehat{\theta} - \theta^*\rangle|$ is small for all values of $y_\varepsilon$.

**Lemma 19.** *Let* $x_1, x_2, \ldots, x_n$ *be a sequence of fixed arm pulls (from a set $\mathcal{X}$) such that each arm $x$ in the support $\lambda$ from D-optimal design (for $\mathcal{X}$) is pulled at least $\lceil \lambda_x \tau \rceil$ times. Consider the matrix* $\mathbf{V} = \sum_{j=1}^n x_j x_j^\mathsf{T}$ *and let $z$ be a vector such that $\|z\|_2 \leq 1$ and $\langle z\mathbf{V}^{\frac{1}{2}}, \theta^*\rangle \geq 6\nu\sqrt{\frac{d}{\tau}} \log(\mathsf{T}|\mathcal{C}_\varepsilon|)$. Then, with probability greater than $1 - \frac{2}{\mathsf{T}|\mathcal{C}_\varepsilon|}$, we have,*

$$
|\langle z\mathbf{V}^{\frac{1}{2}}, \theta^* - \widehat{\theta}\rangle| \leq \left(3\nu\sqrt{\frac{nd}{\tau}} \log(\mathsf{T}|\mathcal{C}_\varepsilon|)\langle x^*, \theta^*\rangle\right)^{\frac{1}{2}}
$$

*Proof.* We begin by utilizing Lemma 6. First, we determine the $\gamma$ parameter in the lemma as follows, for any $t \in [n]$ we have

$$
\begin{aligned}
\left(z\mathbf{V}^{\frac{1}{2}}\right)^T \mathbf{V}^{-1} x_t &\leq \left\|z\mathbf{V}^{\frac{1}{2}}\right\|_{\mathbf{V}^{-1}} \|\mathbf{V}^{-1} x_t\|_{\mathbf{V}} \\
&\leq \|z\|_2 \|x_t\|_{\mathbf{V}^{-1}} \\
&\leq \|x_t\|_{\mathbf{V}^{-1}}. \qquad\text{(since } \|z\|_2 \leq 1\text{)}
\end{aligned}
$$

Let $A_\lambda$ be the optimal design matrix. Since $\mathbf{V} \succ \tau A_\lambda$, we have

$$
\begin{aligned}
\|x_t\|_{\mathbf{V}^{-1}} &\leq \|x_t\|_{\frac{1}{\tau} A_\lambda^{-1}} \\
&\leq \sqrt{\frac{d}{\tau}}. \qquad\text{(by Lemma 3)}
\end{aligned}
$$

Now, we use Corollary 9 with $\gamma = \sqrt{\frac{d}{\tau}}$ and $\delta = \left(3\sqrt{\frac{d}{\tau}} \frac{\nu \log(\mathsf{T}|\mathcal{C}_\varepsilon|)}{\langle z\mathbf{V}^{\frac{1}{2}}, \theta^*\rangle}\right)^{\frac{1}{2}}$. Note that $\delta \in [0, 1]$ since $\langle z\mathbf{V}^{\frac{1}{2}}, \theta^*\rangle \geq 6\sqrt{\frac{d}{\tau}} \nu \log(\mathsf{T}|\mathcal{C}_\varepsilon|)$. We obtain the following probability bound

$$
\mathbb{P}\left\{|\langle z\mathbf{V}^{\frac{1}{2}}, \theta^* - \widehat{\theta}\rangle| \geq \left(3\nu\sqrt{\frac{d}{\tau}} \log(\mathsf{T}|\mathcal{C}_\varepsilon|)\langle z\mathbf{V}^{\frac{1}{2}}, \theta^*\rangle\right)^{\frac{1}{2}}\right\} \leq 2\exp\left(-\frac{3\sqrt{\frac{d}{\tau}} \frac{\nu \log(\mathsf{T}|\mathcal{C}_\varepsilon|)}{\langle z\mathbf{V}^{\frac{1}{2}}, \theta^*\rangle}\langle z\mathbf{V}^{\frac{1}{2}}, \theta^*\rangle}{3\nu\sqrt{\frac{d}{\tau}}}\right)
$$

$$
\leq \frac{2}{\mathsf{T}|\mathcal{C}_\varepsilon|}. \tag{37}
$$

Finally, we establish an upper bound on the term $\langle z\mathbf{V}^{\frac{1}{2}}, \theta^*\rangle$ as follows

$$
\begin{aligned}
\langle z\mathbf{V}^{\frac{1}{2}}, \theta^*\rangle &\leq \|z\|_2 \left\|\mathbf{V}^{\frac{1}{2}}\theta^*\right\|_2 \\
&\leq \sqrt{\theta^{*T}\mathbf{V}\theta^*} \qquad\text{(since } \|z\|_2 \leq 1\text{)} \\
&= \sqrt{\left(\sum_{i \in [n]} \theta^{*T} x_i x_i^T \theta^*\right)} \\
&= \sqrt{n}\langle x^*, \theta^*\rangle. \qquad (\langle x_i, \theta^*\rangle \leq \langle x^*, \theta^*\rangle)
\end{aligned}
$$

Substituting in (37) we get the lemma statement. This completes the proof of the lemma. $\square$

**Lemma 20.** *Consider the same notation as in Lemma 19. If $\langle z\mathbf{V}^{\frac{1}{2}}, \theta^* \rangle \in \left[ 0, 6\nu\sqrt{\frac{d}{\tau}} \log\left(\mathsf{T}|\mathcal{C}_\varepsilon|\right) \right]$, then with probability greater than $1 - \frac{2}{\mathsf{T}|\mathcal{X}|}$ we have*

$$|\langle z\mathbf{V}^{\frac{1}{2}}, \theta^* - \widehat{\theta} \rangle| \leq 12\nu\sqrt{\frac{d}{\tau}} \log\left(\mathsf{T}|\mathcal{C}_\varepsilon|\right).$$

*Proof.* Utilizing Lemma 10, with $\delta = 1$, $\alpha = 6\nu\sqrt{\frac{d}{\tau}} \log\left(\mathsf{T}|\mathcal{C}_\varepsilon|\right)$, and $\gamma = \sqrt{\frac{d}{\tau}}$, we have $\langle z\mathbf{V}^{\frac{1}{2}}, \widehat{\theta} \rangle \leq 12\nu\sqrt{\frac{d}{\tau}} \log\left(\mathsf{T}|\mathcal{C}_\varepsilon|\right)$. Since $\langle z\mathbf{V}^{\frac{1}{2}}, \theta^* \rangle \geq 0$, it follows, with probability greater than $1 - \frac{1}{\mathsf{T}|\mathcal{X}|}$, that

$$\langle z\mathbf{V}^{\frac{1}{2}}, \widehat{\theta} - \theta^* \rangle \leq 12\nu\sqrt{\frac{d}{\tau}} \log\left(\mathsf{T}|\mathcal{C}_\varepsilon|\right).$$

Next, applying Lemma 11 with $\delta = 1$ and $\alpha = 6\nu\sqrt{\frac{d}{\tau}} \log\left(\mathsf{T}|\mathcal{C}_\varepsilon|\right)$, we have, with probability greater than $1 - \frac{1}{\mathsf{T}|\mathcal{X}|}$,

$$\langle z\mathbf{V}^{\frac{1}{2}}, \theta^* - \widehat{\theta} \rangle \leq 6\nu\sqrt{\frac{d}{\tau}} \log\left(\mathsf{T}|\mathcal{C}_\varepsilon|\right) \leq 12\nu\sqrt{\frac{d}{\tau}} \log\left(\mathsf{T}|\mathcal{C}_\varepsilon|\right).$$

Hence, the lemma stands proved. $\qquad\square$

**Lemma 21.** *If $\langle x^*, \theta^* \rangle \geq 196\sqrt{\frac{d^{2.5}\nu}{\mathsf{T}}} \log \mathsf{T}$, then*

$$\mathbb{P}\{G_1\} \geq 1 - \frac{3}{\mathsf{T}} \tag{38}$$

*Proof.* First, we note (from Lemma 13) that arms from the solution of the D-optimal design problem are selected (with probability greater than $1 - \frac{1}{\mathsf{T}}$) at least $\widetilde{\mathsf{T}}/3$ times. Hence, we can use Lemmas 19 and 20 with $\tau = \widetilde{\mathsf{T}}/3$.

Let us consider the case where $\langle y_\varepsilon \mathbf{V}^{\frac{1}{2}}, \theta^* \rangle \geq 6\sqrt{\frac{3d}{\widetilde{\mathsf{T}}}} \log\left(\mathsf{T}|\mathcal{C}_\varepsilon|\right)$. We have that the following holds with probability greater than $1 - \frac{1}{\mathsf{T}|\mathcal{C}_\varepsilon|}$:

$$\left\| \widehat{\theta} - \theta^* \right\|_{\mathbf{V}} \leq \frac{1}{1-\varepsilon} \langle y_\varepsilon \mathbf{V}^{\frac{1}{2}}, \widehat{\theta} - \theta^* \rangle \qquad \text{(from (36))}$$

$$\leq \frac{1}{1-\varepsilon} \left( 3\nu\sqrt{\frac{\widetilde{\mathsf{T}}d}{\frac{\widetilde{\mathsf{T}}}{3}}} \log\left(\mathsf{T}|\mathcal{C}_\varepsilon|\right) \langle x^*, \theta^* \rangle \right)^{\frac{1}{2}} \qquad \text{(using Lemma 19)}$$

$$\leq \frac{1}{1-\varepsilon} \left( 3\sqrt{3d}\,\nu \log\left(\mathsf{T}|\mathcal{C}_\varepsilon|\right) \langle x^*, \theta^* \rangle \right)^{\frac{1}{2}}.$$

Next, we note that $|\mathcal{C}_\varepsilon| \leq \left(\frac{3}{\varepsilon}\right)^d$ [17], and by choosing $\varepsilon = 1/2$ we get

$$\left\| \widehat{\theta} - \theta^* \right\|_{\mathbf{V}} \leq 7 \left( \nu d^{\frac{3}{2}} \log\left(\mathsf{T}\right) \langle x^*, \theta^* \rangle \right)^{\frac{1}{2}}$$

Taking a union bound over all elements in $\mathcal{C}_\varepsilon$ gives a probability bound of $1 - \frac{1}{\mathsf{T}}$.

Now, for the case where $\langle y_\varepsilon \mathbf{V}^{\frac{1}{2}}, \theta^* \rangle \in \left[0, 6\sqrt{\frac{3d}{\widetilde{\mathsf{T}}}} \log\left(\mathsf{T}|\mathcal{C}_\varepsilon|\right)\right]$, substituting $\tau = \widetilde{\mathsf{T}}/3$ in Lemma 20 we have, with probability greater than $1 - \frac{1}{\mathsf{T}|\mathcal{C}_\varepsilon|}$,

$$
\begin{aligned}
\left\|\widehat{\theta} - \theta^*\right\|_{\mathbf{V}} &\leq \frac{1}{1-\varepsilon} \langle y_\varepsilon \mathbf{V}^{\frac{1}{2}}, \widehat{\theta} - \theta^* \rangle \\
&\leq \frac{12\nu}{1-\varepsilon}\sqrt{\frac{d}{\tau}} \log\left(\mathsf{T}|\mathcal{C}_\varepsilon|\right) && \text{(using Lemma 20)} \\
&\leq 24\nu\sqrt{\frac{3d^3}{\widetilde{\mathsf{T}}}} \log\left(\mathsf{T}\right) && \text{(substituting } \varepsilon = 0.5) \\
&\leq 7 \left(d^{\frac{3}{2}}\nu \log\left(\mathsf{T}\right)\langle x^*, \theta^* \rangle\right)^{\frac{1}{2}}
\end{aligned}
$$

The last inequality is due to the fact that $\langle x^*, \theta^* \rangle \geq 196\sqrt{\frac{d^{2.5}\nu}{\mathsf{T}}} \log \mathsf{T}$ and $\widetilde{\mathsf{T}} = 3\sqrt{\mathsf{T}\nu d^{2.5} \log \mathsf{T}}$. We again take a union bound over all elements in $\mathcal{C}_\varepsilon$ to get a probability bound of $1 - \frac{1}{\mathsf{T}}$.

Finally, a union bound over the two cases and the event in Lemma 13 proves the lemma. $\qquad\square$

**Lemma 22.** *If* $\langle x^*, \theta^* \rangle \geq 196\sqrt{\frac{d^{2.5}\nu}{\mathsf{T}}} \log \mathsf{T}$, *then*

$$
\mathbb{P}\{G_2\} \geq 1 - \frac{\log \mathsf{T}}{\mathsf{T}}. \tag{39}
$$

*Proof.* To prove Lemma 22, we follow the same steps as in the proof of Lemma 21. Utilizing Lemma 19 and Lemma 20 with $\tau = \mathsf{T}'$, we establish that for any fixed phase, the following inequality holds with probability greater than $1 - \frac{1}{\mathsf{T}}$:

$$
\left\|\widehat{\theta} - \theta^*\right\|_{\mathbf{V}} \leq 7 \left(d^{\frac{3}{2}}\nu \log \mathsf{T} \langle x^*, \theta^* \rangle\right)^{\frac{1}{2}}.
$$

Taking a union bound over all – at most $\log \mathsf{T}$ – phases in Part II of Algorithm 3 gives us the desired lower bound on $\mathbb{P}\{G_2\}$. $\qquad\square$

**Corollary 23.** *If* $G_1$ *holds, then for all* $x \in \mathcal{X}$, $\widehat{\theta}$ *calculated at the end of Part* I *satisfies*

$$
|\langle x, \widehat{\theta} \rangle - \langle x, \theta^* \rangle| \leq 7\sqrt{\frac{3\langle x^*, \theta^* \rangle d^{2.5}\nu \log \mathsf{T}}{\widetilde{\mathsf{T}}}}
$$

*Consider any phase* $\ell$ *in Part* II. *If* $G_2$ *holds, then for every arm in the surviving arm set* $\widetilde{\mathcal{X}}$, $\widehat{\theta}$ *calculated at the end of the phase satisfies*

$$
|\langle x, \widehat{\theta} \rangle - \langle x, \theta^* \rangle| \leq 7\sqrt{\frac{3\langle x^*, \theta^* \rangle d^{2.5}\nu \log \mathsf{T}}{2^\ell \widetilde{\mathsf{T}}}}.
$$

*Proof.* First we use Hölder's inequality

$$
|\langle x, \theta^* - \widehat{\theta} \rangle| \leq \|x\|_{\mathbf{V}^{-1}} \left\|\theta^* - \widehat{\theta}\right\|_{\mathbf{V}}. \tag{40}
$$

Since $G_1$ holds, arms from the optimal design matrix are selected at least $\widetilde{\mathsf{T}}/3$ times; we have by Lemma 3

$$
\|x\|_{\mathbf{V}^{-1}} \leq \sqrt{\frac{3d}{\widetilde{\mathsf{T}}}}.
$$

Similarly, for every phase in Part II with $\mathsf{T}' = 2^\ell \widetilde{\mathsf{T}}/3$ we have

$$
\|x\|_{\mathbf{V}^{-1}} \leq \sqrt{\frac{d}{\mathsf{T}'}}.
$$

Finally, using bounds on $\left\|\theta^* - \widehat{\theta}\right\|_{\mathbf{V}}$ from events $G_1$ and $G_2$, and substituting in (40), we get the desired bound. $\qquad\square$

**Corollary 24.** *If* $\langle x^*, \theta^* \rangle \geq 196 \sqrt{\frac{d^{2.5}\nu}{\mathsf{T}}} \log \mathsf{T}$

$$\frac{7}{10} \langle x^*, \theta^* \rangle \leq \max_{x \in \mathcal{X}} \langle x, \widehat{\theta} \rangle \leq \frac{13}{10} \langle x^*, \theta^* \rangle$$

*Proof.* Since $\mathsf{T}' \geq 2\widetilde{\mathsf{T}}/3$, via Corollary 23 any $\widehat{\theta}$ calculated in Part I or during any phase of Part II satisfies

$$|\langle x, \widehat{\theta} \rangle - \langle x, \theta^* \rangle| \leq 7 \sqrt{\frac{3\langle x^*, \theta^* \rangle d^{2.5}\nu \log \mathsf{T}}{\widetilde{\mathsf{T}}}}$$

We have

$$\max_{x \in \mathcal{X}} \langle x, \widehat{\theta} \rangle \geq \langle x^*, \widehat{\theta} \rangle$$

$$\geq \langle x^*, \theta^* \rangle - 7 \sqrt{\frac{\langle x^*, \theta^* \rangle d^{2.5}\nu \log \mathsf{T}}{\widetilde{\mathsf{T}}}}$$

$$\geq \langle x^*, \theta^* \rangle \left( 1 - 7 \sqrt{\frac{d^{2.5}\nu \log \mathsf{T}}{\langle x^*, \theta^* \rangle \widetilde{\mathsf{T}}}} \right)$$

$$\geq \frac{7}{10} \langle x^*, \theta^* \rangle \qquad \text{(since } \langle x^*, \theta^* \rangle \geq 196 \sqrt{\frac{d^{2.5}\nu}{\mathsf{T}}} \log \mathsf{T} \text{ and } \widetilde{\mathsf{T}} = 3\sqrt{\mathsf{T}d^{2.5}\nu \log(\mathsf{T})})$$

Now, for any $x \in \mathcal{X}$,

$$\langle x, \widehat{\theta} \rangle \leq \langle x, \theta^* \rangle + 7 \sqrt{\frac{\langle x^*, \theta^* \rangle d^{2.5}\nu \log \mathsf{T}}{\tau}}$$

$$\leq \langle x^*, \theta^* \rangle \left( 1 + 7 \sqrt{\frac{d^{2.5}\nu \log \mathsf{T}}{\langle x^*, \theta^* \rangle \tau}} \right)$$

$$\leq \frac{13}{10} \langle x^*, \theta^* \rangle$$

Hence, the lemma stands proved. $\qquad \square$

**Lemma 25.** *If events $G_1$ and $G_2$ hold then the optimal arm $x^*$ always exists in the surviving set $\widetilde{X}$ in every phase in Part* II *of Algorithm 3*

*Proof.* Let $\tau = \widetilde{\mathsf{T}}/3$ for Part I and $\tau = \mathsf{T}'$ for every phase of Part II. From Corollary 23 we have

$$\langle x^*, \widehat{\theta} \rangle \geq \langle x^*, \theta^* \rangle - 7 \sqrt{\frac{\langle x^*, \theta^* \rangle d^{2.5}\nu \log \mathsf{T}}{\tau}}$$

$$\geq \langle x, \theta^* \rangle - 7 \sqrt{\frac{\langle x^*, \theta^* \rangle d^{2.5}\nu \log \mathsf{T}}{\tau}} \qquad \text{(since } \langle x^*, \theta^* \rangle \geq \langle x, \theta^* \rangle)$$

$$\geq \langle x, \widehat{\theta} \rangle - 14 \sqrt{\frac{\langle x^*, \theta^* \rangle d^{2.5}\nu \log \mathsf{T}}{\tau}} \qquad \text{(using Corollary 23)}$$

$$\geq \langle x, \widehat{\theta} \rangle - 16 \sqrt{\frac{\max_{x \in \widetilde{\mathcal{X}}} \langle x, \theta^* \rangle d^{2.5}\nu \log \mathsf{T}}{\tau}}. \qquad \text{(using Corollary 24)}$$

Hence, the best arm will never satisfy the elimination criteria in Algorithm 3. $\qquad \square$

**Lemma 26.** *Given that events $G_1$ and $G_2$ hold, consider any phase index $\ell$ in Part* II *of Alg. 3. For the surviving set of arms $\widetilde{\mathcal{X}}$ at the beginning of that phase, and for $\widetilde{\mathsf{T}} = \sqrt{d^{2.5}\nu \mathsf{T} \log(\mathsf{T})}$, the following inequality holds for all $x \in \widetilde{\mathcal{X}}$*

$$\langle x, \theta^* \rangle \geq \langle x^*, \theta^* \rangle - 26 \sqrt{\frac{3d^{2.5}\nu \langle x^*, \theta^* \rangle}{2^\ell \cdot \widetilde{\mathsf{T}}}}. \tag{41}$$

*Proof.* Lemma 25 ensures that the optimal arm is contained in the surviving set of arms $\widetilde{\mathcal{X}}$. Furthermore, if an arm $x \in \widetilde{\mathcal{X}}$ is pulled in the $\ell^{\text{th}}$ phase, then it must be the case that arm $x$ was not eliminated in the previous phase (with a phase length parameter $\frac{\mathsf{T}'}{2}$); in particular the arms $x$ does not satisfy the inequality on Line 21 of Algorithm 3. This inequality reduces to

$$\langle x, \widehat{\theta} \rangle \geq \langle x^*, \widehat{\theta} \rangle - 16 \sqrt{\frac{\max_{x \in \widetilde{\mathcal{X}}} \langle x, \widehat{\theta} \rangle \, d^{2.5} \, \nu \, \log\left(\mathsf{T}\right)}{\frac{\mathsf{T}'}{2}}}$$

$$\geq \langle x^*, \widehat{\theta} \rangle - 26 \sqrt{\frac{\langle x^*, \theta^* \rangle \, d^{2.5} \, \nu \, \log\left(\mathsf{T}\right)}{\mathsf{T}'}}$$

(via Corollary 24)

Substituting $\mathsf{T}' = 2^l \widetilde{\mathsf{T}} / 3$ in the above inequality proves the Lemma. $\qquad \square$

**Theorem 2.** *For any given stochastic linear bandits problem with set of arms $\mathcal{X} \subset \mathbb{R}^d$, time horizon $\mathsf{T} \in \mathbb{Z}_+$, and $\nu$-sub-Poisson rewards, Algorithm 2 achieves Nash regret*

$$\mathrm{NR}_\mathsf{T} = O\left(\beta \frac{d^{\frac{5}{4}} \sqrt{\nu}}{\sqrt{\mathsf{T}}} \log(\mathsf{T})\right),$$

*Here, $\beta = \max\{1, \langle x^*, \theta^* \rangle \log d\}$, with $x^* \in \mathcal{X}$ denoting the optimal arm and $\theta^*$ the (unknown) parameter vector.*

*Proof.* Without loss of generality, we assume that $\langle x^*, \theta^* \rangle \geq 196 \sqrt{\frac{d^{2.5} \nu}{\mathsf{T}}} \log \mathsf{T}$. Otherwise, the Nash Regret bound is trivially true. For Part I, the product of expected rewards satisfies

$$\prod_{t=1}^{\widetilde{\mathsf{T}}} \mathbb{E}[\langle X_t, \theta^* \rangle \mid G_1 \cap G_2]^{\frac{1}{\mathsf{T}}} \geq \left(\frac{\langle x^*, \theta^* \rangle}{2(d+1)}\right)^{\frac{\widetilde{\mathsf{T}}}{\mathsf{T}}}$$

(From Lemma 5)

$$= \langle x^*, \theta^* \rangle^{\frac{\widetilde{\mathsf{T}}}{\mathsf{T}}} \left(1 - \frac{1}{2}\right)^{\frac{\log(2(d+1))\widetilde{\mathsf{T}}}{\mathsf{T}}}$$

$$\geq \langle x^*, \theta^* \rangle^{\frac{\widetilde{\mathsf{T}}}{\mathsf{T}}} \left(1 - \frac{\log(2(d+1))\widetilde{\mathsf{T}}}{\mathsf{T}}\right).$$

For Part II, we use Lemma 8. Let $\mathcal{E}_i$ denote the time interval of the $i^{th}$ phase, and let $\mathsf{T}'_i$ be the phase length parameter in that phase. Recall that $|\mathcal{E}_i| \leq \mathsf{T}'_i + \frac{d(d+1)}{2}$. Also, the algorithm runs for at most $\log \mathsf{T}$ phases. Hence, we have

$$\prod_{t=\widetilde{\mathsf{T}}+1}^{\mathsf{T}} \mathbb{E}[\langle X_t, \theta^* \rangle \mid G_1 \cap G_2]^{\frac{1}{\mathsf{T}}} = \prod_{\mathcal{E}_j} \prod_{t \in \mathcal{E}_j} \mathbb{E}[\langle X_t, \theta^* \rangle \mid G_1 \cap G_2]^{\frac{1}{\mathsf{T}}}$$

$$\geq \prod_{\mathcal{E}_j} \left(\langle x^*, \theta^* \rangle - 26 \sqrt{\frac{d^{2.5} \, \nu \, \langle x^*, \theta^* \rangle \log\left(\mathsf{T}\right)}{\mathsf{T}'_j}}\right)^{\frac{|\mathcal{E}_j|}{\mathsf{T}}}$$

$$\geq \langle x^*, \theta^* \rangle^{\frac{\mathsf{T}-\widetilde{\mathsf{T}}}{\mathsf{T}}} \prod_{i=1}^{\log \mathsf{T}} \left(1 - 26 \sqrt{\frac{d^{2.5} \, \nu \, \log\left(\mathsf{T}\right)}{\langle x^*, \theta^* \rangle \mathsf{T}'_j}}\right)^{\frac{|\mathcal{E}_j|}{\mathsf{T}}}$$

$$\geq \langle x^*, \theta^* \rangle^{\frac{\mathsf{T}-\widetilde{\mathsf{T}}}{\mathsf{T}}} \prod_{i=1}^{\log \mathsf{T}} \left(1 - 52 \frac{|\mathcal{E}_j|}{\mathsf{T}} \sqrt{\frac{d^{2.5} \, \nu \, \log\left(\mathsf{T}\right)}{\langle x^*, \theta^* \rangle \mathsf{T}'_j}}\right)$$

The last inequality is due to the fact that $(1-x)^r \geq (1-2rx)$ where $r \in [0,1]$ and $x \in [0,1/2]$. Note that the term $\sqrt{\frac{d^{2.5}\nu \log(\mathsf{T})}{\langle x^*, \theta^*\rangle \mathsf{T}'_j}} \leq 1/2$ for $\langle x^*, \theta^*\rangle \geq 196\sqrt{\frac{d^{2.5}\nu}{\mathsf{T}}}\log \mathsf{T}$, $\mathsf{T}' \geq 2\sqrt{\mathsf{T}d^{2.5}\nu \log \mathsf{T}}$, and $\mathsf{T} \geq e^6$. We can further simplify the expression as follows

$$\prod_{j=1}^{\log \mathsf{T}}\left(1 - 52\frac{|\mathcal{E}_j|}{\mathsf{T}}\sqrt{\frac{d^{2.5}\ \nu\ \log(\mathsf{T})}{\langle x^*, \theta^*\rangle \mathsf{T}'_j}}\right) \geq \prod_{j=1}^{\log \mathsf{T}}\left(1 - 52\frac{\mathsf{T}'_j + \frac{d(d+1)}{2}}{\mathsf{T}}\sqrt{\frac{d^{2.5}\ \nu\ \log(\mathsf{T})}{\langle x^*, \theta^*\rangle \mathsf{T}'_j}}\right)$$

$$\geq \prod_{j=1}^{\log \mathsf{T}}\left(1 - 78\frac{\sqrt{\mathsf{T}'_j}}{\mathsf{T}}\sqrt{\frac{d^{2.5}\ \nu\log(\mathsf{T})}{\langle x^*, \theta^*\rangle}}\right)$$

(assuming $\mathsf{T}'_j \geq d(d+1)$)

$$\geq 1 - 78\frac{1}{\mathsf{T}}\sqrt{\frac{d^{2.5}\ \nu\log(\mathsf{T})}{\langle x^*, \theta^*\rangle}}\left(\sum_{j=1}^{\log \mathsf{T}}\sqrt{\mathsf{T}'_j}\right)$$

$$\geq 1 - 78\frac{1}{\mathsf{T}}\sqrt{\frac{d^{2.5}\ \nu\ \log(\mathsf{T})}{\langle x^*, \theta^*\rangle}}\left(\sqrt{\mathsf{T}\log \mathsf{T}}\right)$$

(using Cauchy Schwarz)

$$\geq 1 - 78\sqrt{\frac{d^{2.5}\nu}{\mathsf{T}\langle x^*, \theta^*\rangle}}\log(\mathsf{T}).$$

Combining the lower bound for rewards in Part I and Part II of the algorithm, we obtain

$$\prod_{t=1}^{\mathsf{T}}\mathbb{E}[\langle X_t, \theta^*\rangle]^{\frac{1}{\mathsf{T}}} \geq \prod_{t=1}^{\mathsf{T}}\left(\mathbb{E}[\langle X_t, \theta^*\rangle \mid G_1 \cap G_2] \cdot \mathbb{P}\{G_1 \cap G_2\}\right)^{\frac{1}{\mathsf{T}}}$$

$$\geq \langle x^*, \theta^*\rangle\left(1 - \frac{\log(2(d+1))\widetilde{\mathsf{T}}}{\mathsf{T}}\right)\left(1 - 78\sqrt{\frac{d^{2.5}\nu}{\mathsf{T}\langle x^*, \theta^*\rangle}}\log(\mathsf{T})\right)\mathbb{P}\{G_1 \cap G_2\}$$

$$\geq \langle x^*, \theta^*\rangle\left(1 - \frac{\log(2(d+1))\widetilde{\mathsf{T}}}{\mathsf{T}} - 78\sqrt{\frac{d^{2.5}\nu}{\mathsf{T}\langle x^*, \theta^*\rangle}}\log(\mathsf{T})\right)\mathbb{P}\{G_1 \cap G_2\}$$

$$\geq \langle x^*, \theta^*\rangle\left(1 - \frac{\log(2(d+1))\widetilde{\mathsf{T}}}{\mathsf{T}} - 78\sqrt{\frac{d^{2.5}\nu}{\mathsf{T}\langle x^*, \theta^*\rangle}}\log(\mathsf{T})\right)\left(1 - \frac{2\log \mathsf{T}}{\mathsf{T}}\right)$$

$$\geq \langle x^*, \theta^*\rangle\left(1 - \frac{\log(2(d+1))3\sqrt{\mathsf{T}d\nu \log(\mathsf{T})}}{\mathsf{T}} - 78\sqrt{\frac{d^{2.5}\nu}{\mathsf{T}\langle x^*, \theta^*\rangle}}\log(\mathsf{T}) - \frac{2\log \mathsf{T}}{\mathsf{T}}\right)$$

$$\geq \langle x^*, \theta^*\rangle - 78\sqrt{\frac{\langle x^*, \theta^*\rangle d^{2.5}\nu}{\mathsf{T}}}\log(\mathsf{T}) - 2\frac{\langle x^*, \theta^*\rangle \log(2(d+1))3\sqrt{d\log(\mathsf{T})}}{\sqrt{\mathsf{T}}}.$$

Hence, the Nash Regret can be bounded as

$$\mathrm{NR}_T = \langle x^*, \theta^*\rangle - \left(\prod_{t=1}^{T}\mathbb{E}[\langle X_t, \theta^*\rangle]\right)^{1/T}$$

$$\leq 78\sqrt{\frac{\langle x^*, \theta^*\rangle d^{2.5}\nu}{\mathsf{T}}}\log(\mathsf{T}) + 2\frac{\langle x^*, \theta^*\rangle \log(2(d+1))3\sqrt{d\nu \log(\mathsf{T})}}{\sqrt{\mathsf{T}}}.$$

The theorem stands proved. $\qquad\qquad\square$

## E  Experiments

We conduct experiments to compare the performance of our algorithm LINNASH with Thompson Sampling on synthetic data. For a comparison, we select Thompson Sampling (Algorithm 1 in

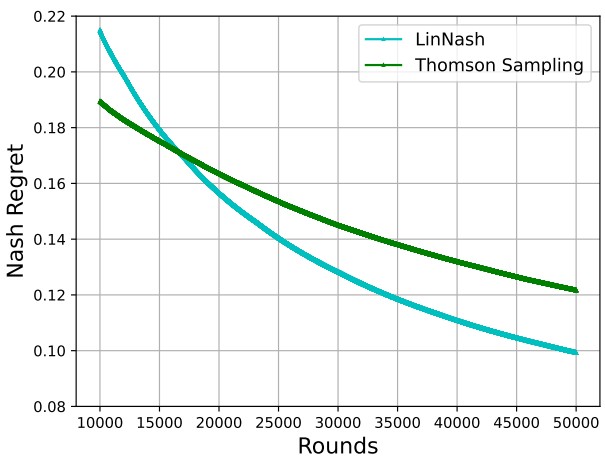

Figure 1: Nash Regret comparison of LINNASH and Thompson Sampling

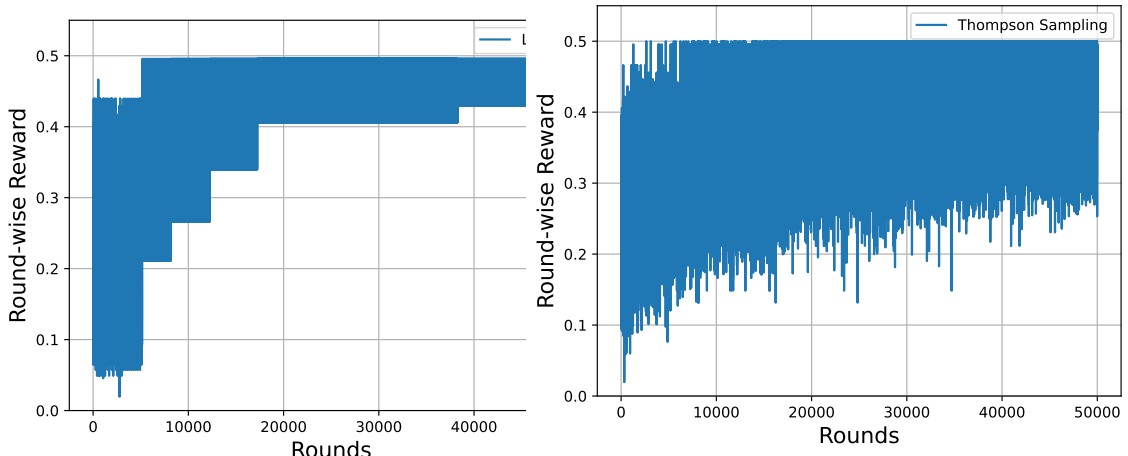

Figure 2: Round-wise reward for LINNASH

Figure 3: Round-wise reward for Thompson Sampling

[2]), instead of UCB/OFUL, since randomization is essential to achieve meaningful Nash Regret guarantees.

We fine-tune the parameters of both algorithms and evaluate their performance in the following experimental setup: We fix the ambient dimension $d = 80$, the number of arms $|\mathcal{X}| = 10000$, and the number of rounds $\mathsf{T} = 50000$. Both the unknown parameter vector, $\theta^*$, and the arm embeddings are sampled from a multivariate Gaussian distribution. Subsequently, the arm embeddings are shifted and normalized to ensure that all mean rewards are non-negative, with the maximum reward mean being set to $0.5$. Upon pulling an arm, we observe a Bernoulli random variable with a probability corresponding to its mean reward.

In this experimental setting, we observe a significant performance advantage of LINNASH over Thompson Sampling. We plot our results in Figure 1, which shows that the Nash regret of LINNASH decreases notably faster than that of Thompson Sampling.

Another notable advantage of LINNASH evident from the experiments is due to successive elimination. The variance in the quality of arms pulled decreases as the number of rounds progresses – see Figures 2 and 3. This is due to the bulk elimination of suboptimal arms at regular intervals. In contrast, Thompson Sampling incurs a large variance in quality of arms being pulled even after several rounds, since no arms are being eliminated at any point.

