**Limitations:** The main contributions of our works are theoretical. From a theoretical point of view, the limitations of our paper are discussed in Section 5. In particular, we believe that tightening the gap between the upper and lower bounds in Nash regret for an infinite set of arms will require novel and non-trivial algorithmic ideas - we leave this as an important direction of future work.

**Broader Impact:** Due to the theoretical nature of this work, we do not foresee any adverse societal impact of this work.

## A  Proof of Concentration Bounds

**Lemma 1.** *Any non-negative random variable $X \in [0, \mathsf{B}]$ is $\mathsf{B}$-sub Poisson, i.e., if mean $\mathbb{E}[X] = \mu$, then for all $\lambda \in \mathbb{R}$, we have $\mathbb{E}[e^{\lambda X}] \leq \exp\left(B^{-1}\mu\left(e^{B\lambda} - 1\right)\right)$.*

*Proof.* For random variable $X$ we have

$$
\begin{aligned}
\mathbb{E}\left[\exp\left(\lambda X\right)\right] &= 1 + \sum_{i=1}^{\infty} \frac{\lambda^i \mathbb{E}\left[X^i\right]}{i!} \\
&\leq 1 + \sum_{i=1}^{\infty} \frac{\lambda^i \mathbb{E}\left[\frac{X}{\mathsf{B}}\mathsf{B}^i\right]}{i!} \\
&= 1 + \frac{\mathbb{E}\left[X\right]}{\mathsf{B}} \sum_{i=1}^{\infty} \frac{\lambda^i \mathsf{B}^i}{i!} \\
&\leq 1 + \frac{\mu}{\mathsf{B}}\left(e^{\lambda \mathsf{B}} - 1\right) \\
&\leq \exp\left(\frac{\mu}{\mathsf{B}}\left(e^{\lambda \mathsf{B}} - 1\right)\right).
\end{aligned}
$$

$\square$

**Lemma 5.** *Let $x_1, x_2, \ldots, x_s \in \mathbb{R}^d$ be a fixed set of vectors and let $r_1, r_2, \ldots, r_s$ be independent $\nu-$sub Poisson random variables satisfying $\mathbb{E}r_s = \langle x_s, \theta^*\rangle$ for some unknown $\theta^*$. In that case, let matrix $\mathbf{V} = \sum_{j=1}^{s} x_j x_j^T$ and $\widehat{\theta} = \mathbf{V}^{-1}\left(\sum_j r_j x_j\right)$ be the least squares estimator of $\theta^*$. Consider any $z \in \mathbb{R}^d$ that satisfies $z^T \mathbf{V}^{-1} x_j \leq \gamma$ for all $j \in [s]$. Then, for any $\delta \in [0, 1]$ we have*

$$
\mathbb{P}\left\{\langle z, \widehat{\theta}\rangle \geq (1+\delta)\langle z, \theta^*\rangle\right\} \leq \exp\left(-\frac{\delta^2 \langle z, \theta^*\rangle}{3\nu\gamma}\right) \text{ and} \tag{8}
$$

$$
\mathbb{P}\left\{\langle z, \widehat{\theta}\rangle \leq (1-\delta)\langle z, \theta^*\rangle\right\} \leq \exp\left(-\frac{\delta^2 \langle z, \theta^*\rangle}{2\nu\gamma}\right) \tag{9}
$$

*Proof.* We use $\mathbf{X}$ to denote a matrix with arm pulls $x_1, x_2, \ldots, x_s$ stacked as rows.

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

$. Clearly, from Lemma 2, we must have that for any $z \in \mathcal{X}$, $\|z\|_{\mathbf{U}(\lambda)^{-1}} \leq d$. By construction of the sequence $\mathcal{S}$ in Step 1 (Subroutine `GenerateArmSequence`), we have $\mathbf{V} \succ \frac{\widetilde{\mathsf{T}}}{3}\mathbf{U}(\lambda)$. Hence

$$
\begin{aligned}
z^T \mathbf{V}^{-1} X_t &\leq \|z\|_{\mathbf{V}^{-1}} \|\mathbf{V}^{-1} X_t\|_{\mathbf{V}} && \text{(By Hölder's inequality)} \\
&= \|z\|_{\mathbf{V}^{-1}} \|X_t\|_{\mathbf{V}^{-1}} \\
&\leq \|z\|_{\left(\frac{\widetilde{\mathsf{T}}}{3}\mathbf{U}(\lambda)\right)^{-1}} \|X_t\|_{\left(\frac{\widetilde{\mathsf{T}}}{3}\mathbf{U}(\lambda)\right)^{-1}} && \text{(since } \mathbf{V} \succ \frac{\widetilde{\mathsf{T}}}{3}\mathbf{U}(\lambda)) \\
&= \sqrt{\frac{3}{\widetilde{\mathsf{T}}}}\|Z\|_{\mathbf{U}(\lambda)^{-1}} \sqrt{\frac{3}{\widetilde{\mathsf{T}}}}\|X_t\|_{\mathbf{U}(\lambda)^{-1}} \\
&\leq \sqrt{\frac{3d}{\widetilde{\mathsf{T}}}}\sqrt{\frac{3d}{\widetilde{\mathsf{T}}}} && \text{(by Lemma 2)} \\
&= \frac{3d}{\widetilde{\mathsf{T}}}.
\end{aligned}
$$

$\square$

**Lemma 14.** *Let $\widehat{\theta}$ be the estimate computed at the end of Part* I *of Algorithm 2. Following holds with probability greater than $1 - \frac{4}{T}$-*

- *All arms $x \in \mathcal{X}$ with $\langle x, \theta^* \rangle \leq 10\sqrt{d\,\nu\mathsf{T}^{-1}\log(\mathsf{T}|\mathcal{X}|)}$ satisfy*

$$\langle x, \widehat{\theta} \rangle \leq 20\sqrt{d\,\nu\mathsf{T}^{-1}\log(\mathsf{T}|\mathcal{X}|)}. \tag{16}$$

- *All arms $x \in \mathcal{X}$ with $\langle x, \theta^* \rangle \geq 10\sqrt{d\,\nu\mathsf{T}^{-1}\log(\mathsf{T}|\mathcal{X}|)}$ satisfy*

$$|\langle x, \theta^* \rangle - \langle x, \widehat{\theta} \rangle| \leq 3\sqrt{\frac{d\langle x, \theta^* \rangle \log(\mathsf{T}|\mathcal{X}|)}{\mathsf{T}'}} \quad and \tag{17}$$

$$\frac{1}{2}\langle x, \theta^* \rangle \leq \langle x, \widehat{\theta} \rangle \leq \frac{4}{3}\langle x, \theta^* \rangle. \tag{18}$$

*Proof.* First, consider the set $\mathcal{X}_{\text{low}}$. We use Lemma 9 for the proof. We set $\gamma = \frac{3d}{\widetilde{\mathsf{T}}}$ (from Lemma 13), $\alpha = 10\sqrt{\frac{d\nu\log(\mathsf{T}\nu|\mathcal{X}|)}{\mathsf{T}}}$ and $\delta = 1$,

$$\mathbb{P}\left\{\langle x, \widehat{\theta} \rangle \leq 20\sqrt{\frac{d\nu\log(\mathsf{T}|\mathcal{X}|)}{\mathsf{T}}}\right\} \leq e^{-\frac{\delta^2\alpha}{3\gamma\nu}}$$

$$\leq \exp\left(-\frac{3\sqrt{\frac{d\nu\log(\mathsf{T}|\mathcal{X}|)}{\mathsf{T}}}3\sqrt{\mathsf{T}d\nu\log(\mathsf{T}|\mathcal{X}|)}}{3\nu d}\right)$$

$$\leq \frac{1}{\mathsf{T}|\mathcal{X}|}.$$

Next, we make use of Lemma 5 for (17). We set $\gamma = \frac{3d}{\widetilde{\mathsf{T}}}$ and $\delta = 3\sqrt{\frac{d\nu\log(\mathsf{T}|\mathcal{X}|)}{\langle x,\theta^*\rangle\widetilde{\mathsf{T}}}}$. Note that since $\langle x, \theta^* \rangle \geq 10\sqrt{d\,\nu\mathsf{T}^{-1}\log(\mathsf{T}|\mathcal{X}|)}$ and $\widetilde{\mathsf{T}} = 3\sqrt{\mathsf{T}d\nu\log(\mathsf{T}|\mathcal{X}|)}$, $\delta$ always lies in $[0,1]$. Hence we can apply Lemma 5 as follows

$$\mathbb{P}\left\{|\langle X, \theta^* \rangle - \langle X, \widehat{\theta} \rangle| \geq 3\sqrt{\frac{\nu d\langle x, \theta^* \rangle \log(\mathsf{T}|\mathcal{X}|)}{\widetilde{\mathsf{T}}}}\right\}$$

$$\leq 2\exp\left(-\frac{\frac{9d\nu\log(\mathsf{T}|\mathcal{X}|)}{\langle x,\theta^*\rangle\widetilde{\mathsf{T}}} \cdot \langle x, \theta^* \rangle}{3\nu\frac{3d}{\widetilde{\mathsf{T}}}}\right)$$

$$= \frac{2}{\mathsf{T}|\mathcal{X}|}.$$

Next, we prove (18). The upper tail is obtained by setting $\gamma = \frac{3d}{\widetilde{\mathsf{T}}}$, $\delta = \frac{1}{3}$ in expression (8) of Lemma 5, we get

$$\mathbb{P}\left\{\langle X, \widehat{\theta} \rangle \geq \frac{4}{3}\langle x, \theta^* \rangle\right\} \leq \exp\left(-\frac{3\sqrt{\mathsf{T}\nu d\log(\mathsf{T}|\mathcal{X}|)} \cdot 10\sqrt{\frac{d\nu\log(\mathsf{T}|\mathcal{X}|)}{\mathsf{T}}}}{27\nu d}\right)$$

$$\text{(Since } \langle x, \theta^* \rangle \geq 10\sqrt{\frac{d\nu\log(\mathsf{T}|\mathcal{X}|)}{\mathsf{T}}}\text{)}$$

$$\leq \frac{1}{\mathsf{T}|\mathcal{X}|}.$$

Similarly substituting $\delta = 1/2$ in expression (9) of Lemma 5 we get

$$\mathbb{P}\left\{\langle X, \widehat{\theta} \rangle \leq \frac{1}{2}\langle x, \theta^* \rangle\right\} \leq \frac{1}{\mathsf{T}|\mathcal{X}|}.$$

Union bound over all arms in $\mathcal{X}$ gives us the required probability bound. □

Next, we look at Part II of Algorithm 2 and show that the event $E_2$ holds with high probability. Note that since we find a sparse $\lambda$ (with support size almost $\frac{d(d+1)}{2}$) in every phase, the phase length is upper bounded as $\mathsf{T}' + \frac{d(d+1)}{2}$.

**Lemma 15.** *Using the notation in Algorithm 2, For all arms $x \in \widetilde{\mathcal{X}}$ with $\langle x, \theta^* \rangle \geq 10 \frac{\sqrt{d\nu \log (\mathsf{T}|\mathcal{X}|)}}{\sqrt{\mathsf{T}}}$, the following holds (for every phase) with probability greater than $1 - \frac{3 \log \mathsf{T}}{\mathsf{T}}$*

$$|\langle x, \theta^* \rangle - \langle x, \widehat{\theta} \rangle| \leq 3 \sqrt{\frac{d\nu \langle x, \theta^* \rangle \log (\mathsf{T}|\mathcal{X}|)}{\mathsf{T}'}} \tag{19}$$

$$\frac{1}{2} \langle x, \theta^* \rangle \leq \langle x, \widehat{\theta} \rangle \leq \frac{4}{3} \langle x, \theta^* \rangle \tag{20}$$

*Proof.* The proof follows the same structure as the proof of Lemma 14. Consider any Phase in Part II andet $\mathbf{U}(\lambda)$ be the optimal design matrix obtained after solving the D-optimal design problem at the start of the phase. Since each arm $a$ in the support of $\lambda$ (denoted by $\mathcal{A}$) is pulled at least $\lceil \lambda_a \mathsf{T}' \rceil$ times, we have $\mathbf{V} \succ \frac{\mathsf{T}'}{3} \mathbf{U}(\lambda)$. Thus by Theorem 2, for $x \in \mathcal{A}$ and all $z \in \widetilde{\mathcal{X}}$ we have

$$z^T \mathbf{V}^{-1} x \leq \|z\|_{\mathbf{V}^{-1}} \|\mathbf{V}^{-1} x\|_{\mathbf{V}} \qquad \text{(By Hölder's inequality)}$$

$$\leq \|z\|_{\mathbf{V}^{-1}} \|x\|_{\mathbf{V}^{-1}} \tag{21}$$

$$\leq \sqrt{\frac{d}{\mathsf{T}'}} \sqrt{\frac{d}{\mathsf{T}'}} = \frac{d}{\mathsf{T}'} \tag{22}$$

Now we use Lemma 5 with $\delta = 3 \sqrt{\frac{d\nu \log (\mathsf{T}|\mathcal{X}|)}{\langle x, \theta^* \rangle \mathsf{T}'}}$ and $\gamma = \frac{d}{\mathsf{T}'}$. Note that given the lower bound on $\langle x, \theta^* \rangle$ and $\mathsf{T}' \geq 2\sqrt{\mathsf{T} d\nu \log(\mathsf{T}|\mathcal{X}|)}$ in every phase, $\delta$ always lies in $[0,1]$. Substituting in Lemma 5, we get

$$\mathbb{P} \left\{ |\langle X, \theta^* \rangle - \langle X, \widehat{\theta} \rangle| \geq 3 \sqrt{\frac{d\nu \langle x, \theta^* \rangle \log (\mathsf{T}|\mathcal{X}|)}{\mathsf{T}'}} \right\} \leq 2 \exp \left( -\frac{\frac{9d \log (\mathsf{T}|\mathcal{X}|)}{\langle x, \theta^* \rangle \mathsf{T}'} \cdot \langle x, \theta^* \rangle}{3 \frac{d}{\mathsf{T}'}} \right)$$

$$\leq \frac{2}{(\mathsf{T}|\mathcal{X}|)^3}$$

Similar to the proof of Lemma 14, we use Lemma 5 with $\delta = \frac{1}{3}$ and $\delta = \frac{1}{2}$ to bound the upper and lower tails of (20) respectively. Furthermore, a union bound across arms in $\mathcal{X}$ and all – at most $\log \mathsf{T}$ – phases gives us the desired probability bound of $1 - \frac{3 \log \mathsf{T}}{\mathsf{T}}$. $\qquad \square$

**Corollary 16.**

$$\mathbb{P} \{E_1 \cap E_2\} \geq 1 - \frac{4 \log \mathsf{T}}{\mathsf{T}}.$$

*Proof.* From Lemma 14 we have $\mathbb{P} \{E_1\} \geq 1 - \frac{4}{\mathsf{T}}$. Furthermore from Lemma 15 we have $\mathbb{P} \{E_2\} \geq 1 - \frac{3 \log \mathsf{T}}{\mathsf{T}}$. Taking union bound over the complements of the two events proves the corollary. $\qquad \square$

**Lemma 17.** *Consider an instance with $\langle x^*, \theta^* \rangle \geq 192 \sqrt{\frac{d\nu \log (\mathsf{T}|\mathcal{X}|)}{\mathsf{T}}}$. If $E_1$ holds, then any arm with mean $\langle x, \theta^* \rangle \leq 10 \sqrt{\frac{d\nu \log(\mathsf{T}|\mathcal{X}|)}{\mathsf{T}}}$ is eliminated after Part I of Algorithm 2.*

*Proof.* From Lemma 14 for any arm with $\langle x, \theta^* \rangle \leq 10\sqrt{\frac{d\nu \log(\mathsf{T}|\mathcal{X}|)}{\mathsf{T}}}$ we have,

$$
\begin{aligned}
\mathrm{UNCB}\left(x, \widehat{\theta}, \widetilde{\mathsf{T}}/3\right) &= \langle x, \widehat{\theta} \rangle + 6\sqrt{\frac{3\langle x, \widehat{\theta} \rangle \, d \, \nu \, \log(\mathsf{T}|\mathcal{X}|)}{\widetilde{\mathsf{T}}}} \\
&\leq 20\sqrt{\frac{d \, \nu \log(\mathsf{T}|\mathcal{X}|)}{T}} + 6\sqrt{\frac{3\langle x, \widehat{\theta} \rangle \, d \, \nu \log(\mathsf{T}|\mathcal{X}|)}{\widetilde{\mathsf{T}}}} \\
&\leq 20\sqrt{\frac{d\nu \log(\mathsf{T}|\mathcal{X}|)}{T}} + 6\sqrt{\frac{3 \cdot 20\sqrt{\frac{d\nu \log(\mathsf{T}|\mathcal{X}|)}{T}} d\nu \log(\mathsf{T}|\mathcal{X}|)}{3\sqrt{\mathsf{T}\nu d \log(\mathsf{T}|\mathcal{X}|)}}} \\
&\qquad\qquad\qquad \text{(via Lemma 14 } \langle x, \widehat{\theta} \rangle \leq 20\sqrt{\frac{d\nu \log(\mathsf{T}|\mathcal{X}|)}{T}}) \\
&\leq 47\sqrt{\frac{d\nu \log(\mathsf{T}|\mathcal{X}|)}{T}}. \tag{23}
\end{aligned}
$$

For the optimal arm $x^*$ we have

$$
\begin{aligned}
\langle x^*, \widehat{\theta} \rangle &\leq \langle x^*, \theta^* \rangle + 3\sqrt{\frac{d\nu \langle x^*, \theta^* \rangle \log(\mathsf{T}|\mathcal{X}|)}{\widetilde{\mathsf{T}}}} = \langle x^*, \theta^* \rangle \left( 1 + 3\sqrt{\frac{d\nu \log(\mathsf{T}|\mathcal{X}|)}{\langle x^*, \theta^* \rangle 3\sqrt{\mathsf{T}d \, \nu \log(\mathsf{T}|\mathcal{X}|)}}} \right) \\
&\qquad\qquad\qquad\qquad \text{(Substituting the value of } \widetilde{\mathsf{T}}) \\
&\leq \langle x^*, \theta^* \rangle \left( 1 + 3\sqrt{\frac{d\nu \log(\mathsf{T}|\mathcal{X}|)}{192\sqrt{\frac{d\nu \log(\mathsf{T}|\mathcal{X}|)}{T}} 3\sqrt{\mathsf{T}\nu d \log(\mathsf{T}|\mathcal{X}|)}}} \right) \\
&= \frac{17}{16}\langle x^*, \theta^* \rangle. \tag{24}
\end{aligned}
$$

This gives us a lower bound on the LNCB of $x^*$

$$
\begin{aligned}
\mathrm{LNCB}\left(x^*, \widehat{\theta}, \widetilde{\mathsf{T}}/3\right) &= \langle x^*, \widehat{\theta} \rangle - 6\sqrt{\frac{3\langle x^*, \widehat{\theta} \rangle \, d \, \nu \log(\mathsf{T}|\mathcal{X}|)}{\widetilde{\mathsf{T}}}} \\
&\geq \langle x^*, \theta^* \rangle - 3\sqrt{\frac{d \, \nu \, \langle x^*, \theta^* \rangle \log(\mathsf{T}|\mathcal{X}|)}{\widetilde{\mathsf{T}}}} - 6\sqrt{\frac{3\langle x^*, \widehat{\theta} \rangle \, d \, \nu \, \log(\mathsf{T}|\mathcal{X}|)}{\widetilde{\mathsf{T}}}} \\
&\qquad\qquad\qquad\qquad\qquad \text{(via Lemma 14)} \\
&\geq \langle x^*, \theta^* \rangle - \left( 3 + 6\sqrt{\frac{51}{16}} \right)\sqrt{\frac{d \, \nu \, \langle x^*, \theta^* \rangle \log(\mathsf{T}|\mathcal{X}|)}{\widetilde{\mathsf{T}}}} \\
&\qquad\qquad\qquad\qquad \text{(since } \langle x^*, \widehat{\theta} \rangle \leq \frac{17}{16}\langle x^*, \theta^* \rangle) \\
&\geq \langle x^*, \theta^* \rangle \left( 1 - 14\sqrt{\frac{d\nu \, \log(\mathsf{T}|\mathcal{X}|)}{\langle x^*, \theta^* \rangle \widetilde{\mathsf{T}}}} \right) \\
&\geq \langle x^*, \theta^* \rangle \left( 1 - 14\sqrt{\frac{d\nu \, \log(\mathsf{T}|\mathcal{X}|)}{192\sqrt{\frac{d\nu \, \log(\mathsf{T}|\mathcal{X}|)}{T}} 3\sqrt{\mathsf{T}d\nu \log(\mathsf{T}|\mathcal{X}|)}}} \right) \\
&\geq \frac{5}{12}\langle x^*, \theta^* \rangle \\
&\geq 80\sqrt{\frac{d\nu \log(\mathsf{T}|\mathcal{X}|)}{\mathsf{T}}}. \tag{25}
\end{aligned}
$$

From (25) and (23) we have

$$
\mathrm{UNCB}\left(x, \widehat{\theta}, \widetilde{\mathsf{T}}/3\right) \leq \mathrm{LNCB}\left(x^*, \widehat{\theta}, \widetilde{\mathsf{T}}/3\right). \tag{26}
$$

$\square$

**Lemma 6.** *The optimal arm $x^*$ always exists in the surviving set $\widetilde{\mathcal{X}}$ in Part I and in every phase in Part II of Algorithm 2 with probability at least $1 - O(\mathsf{T}^{-1} \log \mathsf{T})$.*

*Proof.* Let us assume that events $E_1$ and $E_2$ hold. For any arm $x$ in $\mathcal{X}$ with $\langle x, \theta^* \rangle \geq 10\sqrt{\frac{d\nu \log (\mathsf{T}|\mathcal{X}|)}{\mathsf{T}}}$, we have

$$
\begin{aligned}
\mathrm{LNCB}(x, \widehat{\theta}, \mathsf{T}') = \langle x, \widehat{\theta} \rangle - 6\sqrt{\frac{\langle x, \widehat{\theta} \rangle \, d \, \nu \, \log (\mathsf{T}|\mathcal{X}|)}{\mathsf{T}'}} \\
\leq \langle x, \theta^* \rangle + 3\sqrt{\frac{d \, \nu \, \langle x, \theta^* \rangle \log (\mathsf{T}|\mathcal{X}|)}{\mathsf{T}'}} - 6\sqrt{\frac{\langle x, \widehat{\theta} \rangle \, d\nu \, \log (\mathsf{T}|\mathcal{X}|)}{\mathsf{T}'}} \\
\leq \langle x, \theta^* \rangle - \left( \frac{6}{\sqrt{2}} - 3 \right) \sqrt{\frac{d\nu \langle x, \theta^* \rangle \log (\mathsf{T}|\mathcal{X}|)}{\mathsf{T}'}} \\
\leq \langle x, \theta^* \rangle.
\end{aligned}
$$

Similarly, we have

$$
\begin{aligned}
\mathrm{UNCB}(x^*, \widehat{\theta}, \mathsf{T}') = \langle x^*, \widehat{\theta} \rangle + 6\sqrt{\frac{\langle x^*, \widehat{\theta} \rangle \, d \, \nu \, \log (\mathsf{T}|\mathcal{X}|)}{\mathsf{T}'}} \\
\geq \langle x^*, \theta^* \rangle - 3\sqrt{\frac{d \, \nu \langle x^*, \theta^* \rangle \log (\mathsf{T}|\mathcal{X}|)}{\widetilde{\mathsf{T}}}} + 6\sqrt{\frac{\langle x^*, \widehat{\theta} \rangle \, d \, \nu \log (\mathsf{T}|\mathcal{X}|)}{\mathsf{T}'}} \\
\geq \langle x^*, \theta^* \rangle + \left( \frac{6}{\sqrt{2}} - 3 \right) \sqrt{\frac{d \, \nu \, \langle x^*, \theta^* \rangle \log (\mathsf{T}|\mathcal{X}|)}{\widetilde{\mathsf{T}}}} \\
\geq \langle x^*, \theta^* \rangle.
\end{aligned}
$$

Since $\langle x^*, \theta^* \rangle \geq \langle x, \theta^* \rangle \; \forall x \in \mathcal{X}$, we have $\mathrm{UNCB}(x^*, \widehat{\theta}, \mathsf{T}') \geq \mathrm{LNCB}(x, \widehat{\theta}, \mathsf{T}') \; \forall \mathcal{X}$. From Corollary 16, we have that the events $E_1$ and $E_2$ hold with probability greater than $1 - \frac{4 \log \mathsf{T}}{\mathsf{T}}$. Hence, the lemma stands proven. $\square$

**Lemma 7.** *Consider any phase $\ell$ in Part II of Algorithm 2 and let $\widetilde{\mathcal{X}}$ be the surviving set of arms at the beginning of that phase. Then, with $\widetilde{\mathsf{T}} = \sqrt{d\nu \mathsf{T} \log(\mathsf{T}|\mathcal{X}|)}$, we have*

$$
\Pr \left\{ \langle x, \theta^* \rangle \geq \langle x^*, \theta^* \rangle - 25\sqrt{\frac{3d\nu \langle x^*, \theta^* \rangle \log (\mathsf{T}|\mathcal{X}|)}{2^\ell \cdot \widetilde{\mathsf{T}}}} \text{ for all } x \in \widetilde{\mathcal{X}} \right\} \leq 4\mathsf{T}^{-1} \log \mathsf{T} \qquad (10)
$$

Here, $\nu$ is the sub-Poisson parameter of the stochastic rewards.

*Proof.* Let us assume that events $E_1$ and $E_2$ hold. From the second phase onwards, if an arm is pulled in a phase with phase length parameter $\mathsf{T}'$, then it was not eliminated in the previous phase with phase length parameter $\frac{\mathsf{T}'}{2}$. Additionally, since the best arm is always present in the surviving arm set $\widetilde{\mathcal{X}}$ (via Lemma 6), we have $\mathrm{UNCB}(x, \widehat{\theta}, \mathsf{T}'/2) \geq \mathrm{LNCB}(x^*, \widehat{\theta}, \mathsf{T}'/2)$. That is

$$
\langle x, \widehat{\theta} \rangle + 6\sqrt{\frac{\langle x, \widehat{\theta} \rangle \, d \, \nu \, \log (\mathsf{T}|\mathcal{X}|)}{\frac{\mathsf{T}'}{2}}} \geq \langle x^*, \widehat{\theta} \rangle - 6\sqrt{\frac{\langle x^*, \widehat{\theta} \rangle \, d \, \nu \, \log (\mathsf{T}|\mathcal{X}|)}{\frac{\mathsf{T}'}{2}}}.
$$

Rearranging terms, we get

$$
\begin{aligned}
\langle x, \widehat{\theta} \rangle \geq \langle x^*, \widehat{\theta} \rangle - 6\sqrt{\frac{\langle x^*, \widehat{\theta} \rangle \, d \, \nu \, \log (\mathsf{T}|\mathcal{X}|)}{\frac{\mathsf{T}'}{2}}} - 6\sqrt{\frac{\langle x, \widehat{\theta} \rangle \, d \, \nu \, \log (\mathsf{T}|\mathcal{X}|)}{\frac{\mathsf{T}'}{2}}} \\
\geq \langle x^*, \widehat{\theta} \rangle - 6\sqrt{\frac{4\langle x^*, \theta^* \rangle \, d \, \nu \, \log (\mathsf{T}|\mathcal{X}|)}{\mathsf{T}'}} - 6\sqrt{\frac{4\langle x, \theta^* \rangle \, d \, \nu \log (\mathsf{T}|\mathcal{X}|)}{\mathsf{T}'}} \\
\text{(via Lemma 7 all surviving arms satisfy } \langle x, \widehat{\theta} \rangle \leq \tfrac{4}{3}\langle x, \theta^* \rangle) \\
\geq \langle x^*, \widehat{\theta} \rangle - 20\sqrt{\frac{\langle x^*, \theta^* \rangle \, d \, \nu \, \log (\mathsf{T}|\mathcal{X}|)}{\mathsf{T}'}}.
\end{aligned}
$$

Now using the additive confidence intervals we have,

$$\langle x, \theta^* \rangle \geq \langle x^*, \theta^* \rangle - 20\sqrt{\frac{\langle x^*, \theta^* \rangle \, d \, \nu \, \log\left(\mathsf{T}|\mathcal{X}|\right)}{\mathsf{T}'}} - 3\sqrt{\frac{\langle x^*, \theta^* \rangle \, d \, \nu \, \log\left(\mathsf{T}|\mathcal{X}|\right)}{\frac{\mathsf{T}'}{2}}}$$

$$\geq \langle x^*, \theta^* \rangle - 25\sqrt{\frac{\langle x^*, \theta^* \rangle \, d \, \nu \, \log\left(\mathsf{T}|\mathcal{X}|\right)}{\mathsf{T}'}}.$$

Substituting $\mathsf{T}' = 2^l \widetilde{\mathsf{T}}/3$ in the above inequality proves the Lemma. From Corollary 16, we have that the events $E_1$ and $E_2$ hold with probability greater than $1 - \frac{4\log \mathsf{T}}{\mathsf{T}}$. Hence, the lemma stands proven. □

**Theorem 1.** *Consider the stochastic linear bandits problem over a horizon of $\mathsf{T}$ rounds such that at every round $t \in [\mathsf{T}]$, an arm $X_t \in \mathcal{X} \subset \mathbb{R}^d$ is selected and the corresponding reward $r_t$ is obtained satisfying equation (2). In the setting when $\mathcal{X}$ is finite, Algorithm 2 achieves a Nash regret of*

$$\mathrm{NR}_\mathsf{T} = O\left(\sqrt{\frac{d\nu\langle x^\star, \theta^* \rangle}{\mathsf{T}}} \log(\mathsf{T}|\mathcal{X}|)\right).$$

*Proof.* WLOG we assume that $\langle x^\star, \theta^* \rangle \geq 192\sqrt{\frac{d\,\nu}{\mathsf{T}}} \log(\mathsf{T}|\mathcal{X}|)$, otherwise the Nash Regret bound is trivially true. During Part I of Algorithm 2, the product of expected rewards, conditioned on the event $E_1 \cap E_2$, satisfies

$$\prod_{t=1}^{\widetilde{\mathsf{T}}} \mathbb{E}[\langle X_t, \theta^* \rangle \mid E_1 \cap E_2]^{\frac{1}{\mathsf{T}}} \geq \left(\frac{\langle x^*, \theta^* \rangle}{2(d+1)}\right)^{\frac{\widetilde{\mathsf{T}}}{\mathsf{T}}} \qquad \text{(From Lemma 4)}$$

$$= \langle x^*, \theta^* \rangle^{\frac{\widetilde{\mathsf{T}}}{\mathsf{T}}} \left(1 - \frac{1}{2}\right)^{\frac{\log(2(d+1))\widetilde{\mathsf{T}}}{\mathsf{T}}}$$

$$\geq \langle x^*, \theta^* \rangle^{\frac{\widetilde{\mathsf{T}}}{\mathsf{T}}} \left(1 - \frac{\log(2(d+1))\widetilde{\mathsf{T}}}{\mathsf{T}}\right).$$

For Part II, we use Lemma 7. Let set $\mathcal{E}_i$ denote all $t$ that belong to $i^{th}$ phase and let $\mathsf{T}'_i$ be the phase length parameter in that phase. Since each arm $x$ in $\mathcal{A}$ (the support of D-optimal design) is pulled $\lceil \lambda_x \mathsf{T}'_i \rceil$ times, we have $|\mathcal{E}_i| \leq \mathsf{T}'_i + \frac{d(d+1)}{2}$. Since the phase length parameter doubles after phase, the algorithm would have at most $\log \mathsf{T}$ phases. Hence we have

$$\prod_{t=\widetilde{\mathsf{T}}+1}^{T} \mathbb{E}[\langle X_t, \theta^* \rangle \mid E_1 \cap E_2]^{\frac{1}{\mathsf{T}}} = \prod_{\mathcal{E}_j} \prod_{t \in \mathcal{E}_j} \mathbb{E}[\langle X_t, \theta^* \rangle \mid E_1 \cap E_2]^{\frac{1}{\mathsf{T}}}$$

$$= \prod_{\mathcal{E}_j} \prod_{t \in \mathcal{E}_j} \mathbb{E}[\langle X_t, \theta^* \rangle \mid E_1 \cap E_2]^{\frac{1}{\mathsf{T}}}$$

$$\geq \prod_{\mathcal{E}_j} \left(\langle x^*, \theta^* \rangle - 25\sqrt{\frac{d \, \nu \, \langle x^*, \theta^* \rangle \log\left(\mathsf{T}|\mathcal{X}|\right)}{\mathsf{T}'_j}}\right)^{\frac{|\mathcal{E}_j|}{\mathsf{T}}}$$

$$\geq \langle x^*, \theta^* \rangle^{\frac{T-\widetilde{\mathsf{T}}}{\mathsf{T}}} \prod_{i=1}^{\log T} \left(1 - 25\sqrt{\frac{d \, \nu \, \log\left(\mathsf{T}|\mathcal{X}|\right)}{\langle x^*, \theta^* \rangle \mathsf{T}'_j}}\right)^{\frac{|\mathcal{E}_j|}{\mathsf{T}}}$$

$$\geq \langle x^*, \theta^* \rangle^{\frac{T-\widetilde{\mathsf{T}}}{\mathsf{T}}} \prod_{i=1}^{\log T} \left(1 - 50\frac{|\mathcal{E}_j|}{\mathsf{T}}\sqrt{\frac{d \, \nu \, \log\left(\mathsf{T}|\mathcal{X}|\right)}{\langle x^*, \theta^* \rangle \mathsf{T}'_j}}\right).$$

The last inequality is due to the fact that $(1 - x)^r \geq (1 - 2rx)$ where $r \in [0, 1]$ and $x \in [0, 1/2]$. Note that the term $\sqrt{\frac{d\log\left(\mathsf{T}|\mathcal{X}|\right)}{\langle x^*, \theta^* \rangle \mathsf{T}'_j}} \leq 1/2$ for $\langle x^*, \theta^* \rangle \geq 192\sqrt{\frac{d}{\mathsf{T}}}\log(\mathsf{T}|\mathcal{X}|)$, $\mathsf{T}' \geq 2\sqrt{\mathsf{T}d\log\mathsf{T}|\mathcal{X}|}$ and $\mathsf{T} \geq e^4$. We now further simplify the expression as shown below

$$\prod_{j=1}^{\log T} \left( 1 - 50 \frac{|\mathcal{E}_j|}{T} \sqrt{\frac{d \, \nu \, \log{(T|\mathcal{X}|)}}{\langle x^*, \theta^* \rangle T'_j}} \right) \geq \prod_{j=1}^{\log T} \left( 1 - 50 \frac{T'_j + \frac{d(d+1)}{2}}{T} \sqrt{\frac{d \, \nu \, \log{(T|\mathcal{X}|)}}{\langle x^*, \theta^* \rangle T'_j}} \right)$$

$$\geq \prod_{j=1}^{\log T} \left( 1 - 75 \frac{\sqrt{T'_j}}{T} \sqrt{\frac{d \, \nu \, \log{(T|\mathcal{X}|)}}{\langle x^*, \theta^* \rangle}} \right)$$

$$\text{(assuming } T'_j \geq d(d+1))$$

$$\geq 1 - \frac{75}{T} \sqrt{\frac{d \, \nu \, \log{(T|\mathcal{X}|)}}{\langle x^*, \theta^* \rangle}} \left( \sum_{j=1}^{\log T} \sqrt{T'_j} \right)$$

$$\text{(since } (1-a)(1-b) \geq 1 - a - b \ \forall a, b \geq 0)$$

$$\geq 1 - \frac{75}{T} \sqrt{\frac{d \, \nu \, \log{(T|\mathcal{X}|)}}{\langle x^*, \theta^* \rangle}} \left( \sqrt{T \log T} \right)$$

$$\text{(using Cauchy Schwarz)}$$

$$\geq 1 - 75 \sqrt{\frac{d\nu}{T \langle x^*, \theta^* \rangle}} \log{(T|\mathcal{X}|)}.$$

514 Combining the lower bound for rewards in the two phases, we get

$$\prod_{t=1}^{T} \mathbb{E}[\langle X_t, \theta^* \rangle]^{\frac{1}{T}} \geq \prod_{t=1}^{T} \left( \mathbb{E}[\langle X_t, \theta^* \rangle \mid E_1 \cap E_2] \cdot \mathbb{P}\{E_1 \cap E_2\} \right)^{\frac{1}{T}}$$

$$\geq \langle x^*, \theta^* \rangle \left( 1 - \frac{\log(2(d+1))\widetilde{T}}{T} \right) \left( 1 - 75 \sqrt{\frac{d}{T \langle x^*, \theta^* \rangle}} \log{(T|\mathcal{X}|)} \right) \mathbb{P}\{E_1 \cap E_2\}$$

$$\geq \langle x^*, \theta^* \rangle \left( 1 - \frac{\log(2(d+1))\widetilde{T}}{T} - 75 \sqrt{\frac{d}{T \langle x^*, \theta^* \rangle}} \log{(T|\mathcal{X}|)} \right) \mathbb{P}\{E_1 \cap E_2\}$$

$$\geq \langle x^*, \theta^* \rangle \left( 1 - \frac{\log(2(d+1))\widetilde{T}}{T} - 75 \sqrt{\frac{d}{T \langle x^*, \theta^* \rangle}} \log{(T|\mathcal{X}|)} \right) \left( 1 - \frac{2 \log T}{T} \right)$$

$$\geq \langle x^*, \theta^* \rangle \left( 1 - \frac{\log(2(d+1)) 3 \sqrt{T d \log(T|\mathcal{X}|)}}{T} - 75 \sqrt{\frac{d}{T \langle x^*, \theta^* \rangle}} \log{(T|\mathcal{X}|)} - \frac{2 \log T}{T} \right)$$

$$\geq \langle x^*, \theta^* \rangle - 75 \sqrt{\frac{\langle x^*, \theta^* \rangle d \, \nu}{T}} \log{(T|\mathcal{X}|)} - 6 \sqrt{\frac{d \, \nu \, \log(T|\mathcal{X}|)}{T}} \log(2(d+1)) \langle x^*, \theta^* \rangle.$$

515 Hence the Nash Regret can be bounded as

$$\text{NR}_T = \langle x^*, \theta^* \rangle - \left( \prod_{t=1}^{T} \mathbb{E}[\langle X_t, \theta^* \rangle] \right)^{1/T}$$

$$\leq 75 \sqrt{\frac{\langle x^*, \theta^* \rangle d \, \nu}{T}} \log{(T|\mathcal{X}|)} + 6 \sqrt{\frac{d \, \nu \, \log(T|\mathcal{X}|)}{T}} \log(2(d+1)) \langle x^*, \theta^* \rangle.$$

516 $\qquad\qquad\qquad\qquad\qquad\qquad\qquad\qquad\qquad\qquad\qquad\qquad\qquad\qquad\qquad\qquad\qquad\qquad$ $\square$

## C $\quad \mathcal{X}$ independent Nash regret

518 Instead of working with probability bounds on individual arms, we construct a confidence ellipsoid
519 around $\theta^*$. Using the notations in Algorithm 3, we first define a new set of events for the regret
520 analysis

**Algorithm 3** LINNASH (Nash Confidence Bound Algorithm for Infinite Set of Arms)

**Input:** Arm set $\mathcal{X}$ and horizon of play $T$.

1: Initialize matrix $\mathbf{V} \leftarrow [0]_{d,d}$ and number of rounds $\widetilde{\mathsf{T}} = 3\sqrt{\mathsf{T}d^{2.5}\nu \log(\mathsf{T})}$.
   Part I
2: Generate arm sequence $\mathcal{S}$ for the first $\widetilde{\mathsf{T}}$ rounds using Algorithm 1.
3: **for** $t = 1$ to $\widetilde{\mathsf{T}}$ **do**
4:     Pull the next arm $X_t$ from the sequence $S$.
5:     Observe reward $r_t$ and update $\mathbf{V} \leftarrow \mathbf{V} + X_t X_t^T$
6: **end for**
7: Set estimate $\widehat{\theta} := \mathbf{V}^{-1}\left(\sum_{t=1}^{\widetilde{\mathsf{T}}} r_t X_t\right)$
8: Find $\eta = \max_{z \in \mathcal{X}} \langle z, \widehat{\theta}\rangle$
9: Update $\widetilde{\mathcal{X}} \leftarrow \{x \in \mathcal{X} : \langle x, \widehat{\theta}\rangle \geq \eta - 16\sqrt{\frac{3\,\eta\,d^{\frac{5}{2}}\,\nu\,\log(\mathsf{T})}{\widetilde{\mathsf{T}}}}\}$
10: $\mathsf{T}' \leftarrow \frac{2}{3}\widetilde{\mathsf{T}}$
    Part II
11: **while** end of time horizon $\mathsf{T}$ is reached **do**
12:     Initialize $V = [0]_{d,d}$ to be an all zeros $d \times d$ matrix and $s = [0]_d$ to be an all-zeros vector.
        // Beginning of new phase.
13:     Find the probability distribution $\lambda \in \Delta(\widetilde{\mathcal{X}})$ by maximizing the following objective

$$\log \mathsf{Det}(\mathbf{V}(\lambda)) \text{ subject to } \lambda \in \Delta(\widetilde{\mathcal{X}}) \text{ and } \mathsf{Supp}(\lambda) \leq d(d+1)/2. \tag{27}$$

14:     **for** $a$ in $\mathsf{Supp}(\lambda)$ **do**
15:         Pull $a$ for the next $\lceil \lambda_a \mathsf{T}'\rceil$ rounds.
16:         Observe rewards and Update $\mathbf{V} \leftarrow \mathbf{V} + \lceil \lambda_A \mathsf{T}'\rceil \cdot aa^T$
17:         Observe $\lceil \lambda_a \mathsf{T}'\rceil$ corresponding rewards $z_1, z_2, \ldots$ and update $s \leftarrow s + (\sum_j z_j)a$.
18:     **end for**
19:     Estimate $\widehat{\theta} = \mathbf{V}^{-1}\left(\sum_{t \in \mathcal{E}} r_t X_t\right)$
20:     Find $\eta = \max_{z \in \mathcal{X}} \langle z, \widehat{\theta}\rangle$
21:     $\widetilde{\mathcal{X}} \leftarrow \{x \in \mathcal{X} : \langle x, \widehat{\theta}\rangle \geq \eta - 16\sqrt{\frac{\eta\,d^{\frac{5}{2}}\,\log(\mathsf{T})}{\mathsf{T}'}}\}$
22:     $\mathsf{T}' \leftarrow 2 \times \mathsf{T}'$   // End of phase.
23: **end while**

---

$G_1$  During Part I arms from the D-optimal design are chosen at least $\widetilde{\mathsf{T}}/3$ times. If $\langle x^*, \theta^*\rangle \geq 196\sqrt{\frac{d^{2.5}}{\widetilde{\mathsf{T}}}}\log \mathsf{T}$, then $\widehat{\theta}$ calculated at the end of Part I satisfies,

$$\left\|\widehat{\theta} - \theta^*\right\|_{\mathbf{V}} \leq 7\sqrt{\langle x^*, \theta^*\rangle d^{\frac{3}{2}}\nu \log \mathsf{T}}$$

$G_2$  During Part II, for every phase, if $\langle x^*, \theta^*\rangle \geq 196\sqrt{\frac{d^{2.5}}{\widetilde{\mathsf{T}}}}\log \mathsf{T}$ the estimators $\widehat{\theta}$ satisfy the following

$$\left\|\widehat{\theta} - \theta^*\right\|_{\mathbf{V}} \leq 7\sqrt{\langle x^*, \theta^*\rangle d^{\frac{3}{2}}\nu \log \mathsf{T}}$$

## C.1   Regret Analysis

WLOG let us assume that $\langle x^*, \theta^*\rangle \geq 196\frac{d^{1.25}}{\sqrt{\widetilde{\mathsf{T}}}}\log \mathsf{T}$, otherwise the regret bound is trivially satisfied. Let $\mathcal{B}$ denote the unit ball in $\mathbb{R}^d$, we have

$$\left\|\widehat{\theta} - \theta^*\right\|_{\mathbf{V}} = \left\|\mathbf{V}^{\frac{1}{2}}(\widehat{\theta} - \theta^*)\right\|_2$$
$$= \max_{y \in \mathcal{B}} \langle y, \mathbf{V}^{\frac{1}{2}}(\widehat{\theta} - \theta^*)\rangle$$

We construct an $\varepsilon$-net for the unit ball, which we will refer to as $\mathcal{C}_\varepsilon$. We define $y_\varepsilon = \arg\min_{b \in \mathcal{B}} \|b - y\|_2$

$$\left\|\widehat{\theta} - \theta^*\right\|_\mathbf{V} = \max_{y \in \mathcal{B}} \langle y - y_\varepsilon, \mathbf{V}^{\frac{1}{2}}(\widehat{\theta} - \theta^*)\rangle + \langle y_\varepsilon, \mathbf{V}^{\frac{1}{2}}(\widehat{\theta} - \theta^*)\rangle$$

$$\leq \max_{y \in \mathcal{B}} \|y - y_\varepsilon\|_2 \left\|\mathbf{V}^{\frac{1}{2}}(\widehat{\theta} - \theta^*)\right\|_2 + \langle y_\varepsilon, \mathbf{V}^{\frac{1}{2}}(\widehat{\theta} - \theta^*)\rangle$$

$$\leq \varepsilon \left\|(\widehat{\theta} - \theta^*)\right\|_\mathbf{V} + \langle y_\varepsilon, \mathbf{V}^{\frac{1}{2}}(\widehat{\theta} - \theta^*)\rangle$$

Rearranging we get

$$\left\|\widehat{\theta} - \theta^*\right\|_\mathbf{V} \leq \frac{1}{1-\varepsilon}\langle y_\varepsilon \mathbf{V}^{\frac{1}{2}}, \widehat{\theta} - \theta^*\rangle \tag{28}$$

In the following lemmas we show show that $\langle y_\varepsilon \mathbf{V}^{\frac{1}{2}}, \widehat{\theta} - \theta^*\rangle$ is small for all values of $y_\varepsilon$.

**Lemma 18.** *Let $x_1, x_2, \ldots, x_n$ be a sequence of fixed arm pulls (from a set $\mathcal{X}$) such that each arm $x$ in the support $\lambda$ from D-optimal design is pulled at least $\lceil \lambda_x \tau \rceil$ times. Consider $\mathbf{V} = \sum_{j=1}^{s} x_j x_j^\mathsf{T}$ and let $w$ be a vector such that $\|w\|_2 \leq 1$ and $\langle w \mathbf{V}^{\frac{1}{2}}, \theta^*\rangle \geq 6\sqrt{\frac{d}{\tau}}\log(\mathsf{T}|\mathcal{C}_\varepsilon|)$. Then, with probability greater than $1 - \frac{2}{\mathsf{T}|\mathcal{C}_\varepsilon|}$, we have,*

$$|\langle w\mathbf{V}^{\frac{1}{2}}, \theta^* - \widehat{\theta}\rangle| \leq \left(3\sqrt{\frac{nd}{\tau}}\log(\mathsf{T}|\mathcal{C}_\varepsilon|)\langle x^*, \theta^*\rangle\right)^{\frac{1}{2}}$$

*Proof.* We will make use of Lemma 5. We find the $\gamma$ parameter used in the lemma. We have

$$\left(w\mathbf{V}^{\frac{1}{2}}\right)^T \mathbf{V}^{-1} X_t \leq \left\|w\mathbf{V}^{\frac{1}{2}}\right\|_{\mathbf{V}^{-1}} \|\mathbf{V}^{-1} X_t\|_\mathbf{V}$$

$$\leq \|w\|_2 \|X_t\|_{\mathbf{V}^{-1}}$$

$$\leq \|X_t\|_{\mathbf{V}^{-1}} \qquad \text{(since } \|w\| \leq 1\text{)}$$

Let $A_\lambda$ be the optimal design matrix then we have $\mathbf{V} \succ \tau A_\lambda$. This gives us the following

$$\left(w\mathbf{V}^{\frac{1}{2}}\right)^T \mathbf{V}^{-1} X_t \leq \|X_t\|_{\mathbf{V}^{-1}}$$

$$\leq \|X_t\|_{\frac{1}{\tau}A_\lambda^{-1}}$$

$$\leq \sqrt{\frac{d}{\tau}} \qquad \text{(By Theorem 2)}$$

We use Corollary 8 with $\gamma = \sqrt{\frac{d}{\tau}}$ and $\delta = \left(6\sqrt{\frac{d}{\tau}}\frac{\nu \log(\mathsf{T}|\mathcal{C}_\varepsilon|)}{\langle w\mathbf{V}^{\frac{1}{2}}, \theta^*\rangle}\right)^{\frac{1}{2}}$. Note that $\delta \in [0,1]$ since $\langle w\mathbf{V}^{\frac{1}{2}}, \theta^*\rangle \geq 6\sqrt{\frac{d}{\tau}}\log(\mathsf{T}|\mathcal{C}_\varepsilon|)$. We have the following probability bound

$$\mathbb{P}\left\{|\langle w\mathbf{V}^{\frac{1}{2}}, \theta^* - \widehat{\theta}\rangle| \geq \left(6\sqrt{\frac{d}{\tau}}\nu \log(\mathsf{T}|\mathcal{C}_\varepsilon|)\langle w\mathbf{V}^{\frac{1}{2}}, \theta^*\rangle\right)^{\frac{1}{2}}\right\} \leq 2\exp\left(-\frac{6\sqrt{\frac{d}{\tau}}\frac{\log(\mathsf{T}|\mathcal{C}_\varepsilon|)}{\langle w\mathbf{V}^{\frac{1}{2}}, \theta^*\rangle}\langle w\mathbf{V}^{\frac{1}{2}}, \theta^*\rangle}{3\sqrt{\frac{d}{\tau}}}\right)$$

$$\leq \frac{2}{\mathsf{T}|\mathcal{C}_\varepsilon|}$$

We can get an upper bound on the term $\langle w\mathbf{V}^{\frac{1}{2}}, \theta^*\rangle$ as follows

$$\langle w\mathbf{V}^{\frac{1}{2}}, \theta^*\rangle \leq \|w\|_2 \left\|\mathbf{V}^{\frac{1}{2}}\theta^*\right\|_2$$

$$\leq \sqrt{\theta^{*T}\mathbf{V}\theta^*} \qquad \text{(since } \|w\| \leq 1\text{)}$$

$$= \sqrt{\left(\sum_{i \in [n]} \theta^{*T} x_i x_i^T \theta^*\right)} \qquad (\langle x_i, \theta^*\rangle \leq \langle x^*, \theta^*\rangle)$$

$$= \sqrt{n\langle x^*, \theta^*\rangle}$$

This proves the lemma. $\qquad \square$

**Lemma 19.** *Using the same notation as Lemma 18, If* $\langle w\mathbf{V}^{\frac{1}{2}}, \theta^*\rangle \leq 6\sqrt{\frac{d}{\tau}}\,\log\left(\mathsf{T}|\mathcal{C}_\varepsilon|\right)$

$$|\langle w\mathbf{V}^{\frac{1}{2}}, \theta^* - \widehat{\theta}\rangle| \leq 12\sqrt{\frac{d}{\tau}}\,\log\left(\mathsf{T}|\mathcal{C}_\varepsilon|\right)$$

*Proof.* We first use Lemma 9 to show $\langle w\mathbf{V}^{\frac{1}{2}}, \widehat{\theta}\rangle \leq 12\sqrt{\frac{d}{\mathsf{T}'}}$ by substituting $\delta = 1$, $\alpha = 6\sqrt{\frac{d}{\mathsf{T}'}}\,\log\left(\mathsf{T}|\mathcal{C}_\varepsilon|\right)$ and $\gamma = \sqrt{\frac{d}{\mathsf{T}'}}$. This trivially gives us $\langle w\mathbf{V}^{\frac{1}{2}}, \theta^* - \widehat{\theta}\rangle| \leq 12\sqrt{\frac{d}{\mathsf{T}'}}\log\left(\mathsf{T}|\mathcal{C}_\varepsilon|\right)$.

Next we Lemma 10 with $\delta = 1$ and $\alpha = 6\sqrt{\frac{d}{\mathsf{T}'}}\,\log\left(\mathsf{T}|\mathcal{C}_\varepsilon|\right)$ which gives $\langle w\mathbf{V}^{\frac{1}{2}}, \theta^* - \widehat{\theta}\rangle| \leq 6\sqrt{\frac{d}{\mathsf{T}'}}\log\left(\mathsf{T}|\mathcal{C}_\varepsilon|\right)$. $\qquad\square$

**Lemma 20.** *If the optimal arm satisfies* $\langle x^*, \theta^*\rangle \geq 196\sqrt{\frac{d^{2.5}}{\mathsf{T}}}\log\mathsf{T}$

$$\mathbb{P}\{G_1\} \geq 1 - \frac{3}{\mathsf{T}}$$

*and*

$$\mathbb{P}\{G_2\} \geq 1 - \frac{\log\mathsf{T}}{\mathsf{T}}$$

*Proof.* Recall, from (28) that we aim to get a bound on $\langle y_\varepsilon \mathbf{V}^{\frac{1}{2}}, \widehat{\theta} - \theta^*\rangle$ for all possible values of $y_\varepsilon$. The total number of arm pulls in Part I is equal to $\widetilde{\mathsf{T}}$. We will now apply Lemma 18. First, from Lemma 12 we have that the arms from the solution of the D-optimal design problem are selected (with probability greater than $1 - \frac{1}{\mathsf{T}}$) at least $\widetilde{\mathsf{T}}/3$ times, that is, $\tau = \widetilde{\mathsf{T}}/3$. Let us consider the case where $\langle y_\varepsilon \mathbf{V}^{\frac{1}{2}}, \theta^*\rangle \geq 6\sqrt{\frac{3d}{\widetilde{\mathsf{T}}}}\,\log\left(\mathsf{T}|\mathcal{C}_\varepsilon|\right)$. Taking union bound over $\mathcal{C}_\varepsilon$ we get that the following holds with probability greater than $1 - \frac{1}{\mathsf{T}}$

$$\left\|\widehat{\theta} - \theta^*\right\|_{\mathbf{V}} \leq \frac{1}{1-\varepsilon}\langle y_\varepsilon \mathbf{V}^{\frac{1}{2}}, \widehat{\theta} - \theta^*\rangle \qquad\qquad \text{(From (28))}$$

$$\leq \frac{1}{1-\varepsilon}\left(3\sqrt{\frac{\widetilde{\mathsf{T}}d}{\frac{\widetilde{\mathsf{T}}}{3}}}\,\log\left(\mathsf{T}|\mathcal{C}_\varepsilon|\right)\langle x^*, \theta^*\rangle\right)^{\frac{1}{2}} \qquad\qquad \text{(Using Lemma 18)}$$

$$\leq \frac{1}{1-\varepsilon}\left(3\sqrt{3d}\log\left(\mathsf{T}|\mathcal{C}_\varepsilon|\right)\langle x^*, \theta^*\rangle\right)^{\frac{1}{2}}$$

Since $|\mathcal{C}_\varepsilon| \leq \left(\frac{3}{\varepsilon}\right)^d$, choosing $\epsilon = 1/2$ gives us

$$\left\|\widehat{\theta} - \theta^*\right\|_{\mathbf{V}} \leq 7\left(d^{\frac{3}{2}}\log\left(\mathsf{T}\right)\langle x^*, \theta^*\rangle\right)^{\frac{1}{2}}$$

Now substituting $\tau = \mathsf{T}'/3$ in Lemma 19, if $\langle y_\varepsilon \mathbf{V}^{\frac{1}{2}}, \theta^*\rangle \leq 6\sqrt{\frac{3d}{\widetilde{\mathsf{T}}}}\,\log\left(\mathsf{T}|\mathcal{C}_\varepsilon|\right)$, we have

$$\left\|\widehat{\theta} - \theta^*\right\|_{\mathbf{V}} \leq \frac{1}{1-\varepsilon}\langle y_\varepsilon \mathbf{V}^{\frac{1}{2}}, \widehat{\theta} - \theta^*\rangle$$

$$\leq 24\sqrt{\frac{d^3}{\mathsf{T}'}}\,\log\left(\mathsf{T}\right) \qquad\qquad \text{(From Lemma 19 and substituting } \varepsilon = 0.5)$$

$$\leq 7\left(d^{\frac{3}{2}}\log\left(\mathsf{T}\right)\langle x^*, \theta^*\rangle\right)^{\frac{1}{2}}$$

The last inequality is due to the fact that $\langle x^*, \theta^*\rangle \geq 196\sqrt{\frac{d^{2.5}}{\mathsf{T}}}\log\mathsf{T}$ and $\mathsf{T}' = \widetilde{\mathsf{T}}/3 \geq \sqrt{\mathsf{T}d^{2.5}\log\mathsf{T}}$.

Similarly, for the event $G_2$, an identical use of Lemma 19 and Lemma 18 with $\tau = \mathsf{T}'$ shows that, for any fixed phase, the following holds with probability greater than $1 - \frac{1}{\mathsf{T}}$

$$\left\|\widehat{\theta} - \theta^*\right\|_{\mathbf{V}} \leq 7\left(d^{\frac{3}{2}}\log\left(\mathsf{T}\right)\langle x^*, \theta^*\rangle\right)^{\frac{1}{2}}$$

Taking a union bound over all phases (almost $\log\mathsf{T}$) of Part II gives us the required bound on $G_2$. $\quad\square$

**Corollary 21.** *If $G_1$ holds, the for all $x \in \mathcal{X}$, $\widehat{\theta}$ calculated at the end of Part I satisfies*

$$|\langle x, \widehat{\theta}\rangle - \langle x, \theta^*\rangle| \leq 7\sqrt{\frac{3\langle x^*, \theta^*\rangle d^{2.5} \log \mathsf{T}}{\widetilde{\mathsf{T}}}}$$

*Consider any phase $\ell$ in Part* II. *If $G_2$ holds, then for every arm in the surviving arm set $\widetilde{\mathcal{X}}$, $\widehat{\theta}$ calculated at the end of the phase satisfies*

$$|\langle x, \widehat{\theta}\rangle - \langle x, \theta^*\rangle| \leq 7\sqrt{\frac{3\langle x^*, \theta^*\rangle d^{2.5} \log \mathsf{T}}{2^\ell \widetilde{\mathsf{T}}}}.$$

*Proof.* First we use Hölder's inequality

$$|\langle x, \theta^* - \widehat{\theta}\rangle| \leq \|x\|_{\mathbf{V}^{-1}} \left\|\theta^* - \widehat{\theta}\right\|_{\mathbf{V}}. \tag{29}$$

Since $G_1$ holds, arms from the optimal design matrix are selected at least $\widetilde{\mathsf{T}}/3$ times; we have by Lemma 2

$$\|x\|_{\mathbf{V}^{-1}} \leq \sqrt{\frac{3d}{\widetilde{\mathsf{T}}}}.$$

Similarly, for every Phase in Part II with $\mathsf{T}' = 2^\ell \widetilde{\mathsf{T}}/3$ we have

$$\|x\|_{\mathbf{V}^{-1}} \leq \sqrt{\frac{d}{\mathsf{T}'}}.$$

Finally, using bound on $\left\|\theta^* - \widehat{\theta}\right\|_{\mathbf{V}}$ from events $G_1$ and $G_2$, we get the desired result. $\qquad\square$

**Corollary 22.** *If $\langle x^*, \theta^*\rangle \geq 196\sqrt{\frac{d^{2.5}}{\mathsf{T}}} \log \mathsf{T}$*

$$\frac{7}{10}\langle x^*, \theta^*\rangle \leq \max_{z \in \mathcal{X}}\langle z, \widehat{\theta}\rangle \leq \frac{13}{10}\langle x^*, \theta^*\rangle$$

*Proof.* Since $\mathsf{T}' \geq 2\widetilde{\mathsf{T}}/3$, via Lemma 21 any $\widehat{\theta}$ calculated in Part I or during any phase of Part II satisfies

$$|\langle x, \widehat{\theta}\rangle - \langle x, \theta^*\rangle| \leq 7\sqrt{\frac{3\langle x^*, \theta^*\rangle d^{2.5} \log \mathsf{T}}{\widetilde{\mathsf{T}}}}$$

We have

$$\max_{z \in \mathcal{X}}\langle z, \widehat{\theta}\rangle \geq \langle x^*, \widehat{\theta}\rangle$$

$$\geq \langle x^*, \theta^*\rangle - 7\sqrt{\frac{\langle x^*, \theta^*\rangle d^{2.5} \log \mathsf{T}}{\widetilde{\mathsf{T}}}}$$

$$\geq \langle x^*, \theta^*\rangle \left(1 - 7\sqrt{\frac{d^{2.5} \log \mathsf{T}}{\langle x^*, \theta^*\rangle \widetilde{\mathsf{T}}}}\right)$$

$$\geq \frac{7\langle x^*, \theta^*\rangle}{10} \qquad \text{(since } \langle x^*, \theta^*\rangle \geq 196\sqrt{\frac{d^{2.5}}{\mathsf{T}}} \log \mathsf{T} \text{ and } \widetilde{\mathsf{T}} = 3\sqrt{\mathsf{T}d^{2.5}\nu \log(\mathsf{T})})$$

Similarly for any $z \in \mathcal{X}$,

$$\langle z, \widehat{\theta}\rangle \leq \langle z, \theta^*\rangle + 7\sqrt{\frac{\langle x^*, \theta^*\rangle d^{2.5} \log \mathsf{T}}{\tau}}$$

$$\leq \langle x^*, \theta^*\rangle \left(1 + 7\sqrt{\frac{d^{2.5} \log \mathsf{T}}{\langle x^*, \theta^*\rangle \tau}}\right)$$

$$\leq \frac{13}{10}\langle x^*, \theta^*\rangle$$

$\qquad\square$

**Lemma 23.** *If events $G_1$ and $G_2$ hold then the optimal arm $x^*$ always exists in the surviving set $\widetilde{X}$ in every phase in Step II of Alg. 3.*

*Proof.* Let $\tau = \widetilde{T}/3$ for Part I and $\tau = T'$ for every phase of Part II. From Lemma 21 we have for $x \in \widetilde{\mathcal{X}}$

$$
\begin{aligned}
\langle x^*, \widehat{\theta} \rangle &\geq \langle x^*, \theta^* \rangle - 7\sqrt{\frac{\langle x^*, \theta^* \rangle d^{2.5} \log T}{\tau}} \\
&\geq \langle x, \theta^* \rangle - 7\sqrt{\frac{\langle x^*, \theta^* \rangle d^{2.5} \log T}{\tau}} \\
&\geq \langle x, \widehat{\theta} \rangle - 14\sqrt{\frac{\langle x^*, \theta^* \rangle d^{2.5} \log T}{\tau}} \\
&\geq \langle x, \widehat{\theta} \rangle - 16\sqrt{\frac{\max_{z \in \widetilde{\mathcal{X}}} \langle z, \theta^* \rangle d^{2.5} \log T}{\tau}} \qquad \text{(Using Lemma 22)}
\end{aligned}
$$

Hence, the best arm will never satisfy the elimination criteria in Alg. 3. $\qquad \square$

**Lemma 24.** *Given that events $G_1$ and $G_2$ hold, fix any phase index $\ell$ in Step II of Alg. 3. For the surviving set of arms $\widetilde{\mathcal{X}}$ at the beginning of that phase, we will have for $\widetilde{T} = \sqrt{d^{2.5} T \log(T)}$*

$$
\langle x, \theta^* \rangle \geq \langle x^*, \theta^* \rangle - 26\sqrt{\frac{3 d^{2.5} \nu \langle x^*, \theta^* \rangle}{2^\ell \cdot \widetilde{T}}} \text{ for all } x \in \widetilde{\mathcal{X}} \tag{30}
$$

*Proof.* From the second phase onwards, if an arm is pulled in a phase with phase length parameter $T'$, then it was not eliminated in the previous phase with phase length parameter $\frac{T'}{2}$. Additionally, since the best arm is always present in the surviving arm set $\widetilde{\mathcal{X}}$ (via Lemma 23), we have

$$
\begin{aligned}
\langle x, \widehat{\theta} \rangle &\geq \langle x^*, \widehat{\theta} \rangle - 16\sqrt{\frac{\max_{z \in \widetilde{\mathcal{X}}} \langle z, \widehat{\theta} \rangle d^{2.5} \nu \log(T)}{\frac{T'}{2}}} \\
&\geq \langle x^*, \widehat{\theta} \rangle - 26\sqrt{\frac{\langle x^*, \theta^* \rangle d^{2.5} \nu \log(T)}{T'}}
\end{aligned}
$$

$$\text{(via Lemma 22)}$$

Substituting $T' = 2^l \widetilde{T}/3$ in the above inequality proves the Lemma. $\qquad \square$

**Theorem 2.** *Consider the stochastic linear bandits problem over a horizon of $T$ rounds such that at every round $t \in [T]$, an arm $X_t \in \mathcal{X} \subset \mathbb{R}^d$ is selected and the corresponding reward $r_t$ is obtained satisfying equation (2). In this setting, Algorithm 3 achieves a Nash regret of*

$$

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

$\square$