# OpenReview forum: "Nash Regret Guarantees for Linear Bandits"
_NeurIPS.cc/2023/Conference — NeurIPS 2023 poster_

### Official Review · Reviewer_K5vq · 2023-06-30

**Soundness:** 3 good
**Presentation:** 3 good
**Contribution:** 3 good
**Rating:** 6
**Confidence:** 3

**Summary:**

In this research, the authors investigate the concept of Nash regret in the context of stochastic linear bandits. Nash regret represents the disparity between the optimal reward and the geometric mean of expected rewards, reflecting fairness considerations across different time instances within the Nash social welfare framework. The study assumes a fixed set of arms denoted as $\mathcal{X}$, and an unknown vector $\theta^*$. When an arm x is chosen, the agent observes a sub-Poisson $r_t$ with a mean of $<x,\theta^*>$. The nash regret is defined as $max_x<x,\theta^*>-(\prod_t \mathbb{E}[<X_t,\theta^*>])^{1/T}$ where $X_t$ is a selected arm at time t.

To minimize Nash regret, the authors propose an algorithm that incorporates elimination phases. Firstly, to ensure the lower bound of expected rewards, they employ sampling via the Jon ellipsoid technique, as in Lemma 4. Additionally, they employ a G-optimal design to construct a probability distribution over arms, aiming to minimize the confidence interval width of estimated rewards. By combining these strategies, a sequence of arms is generated, and elimination phases are employed to remove arms with suboptimal mean rewards, utilizing upper confidence bounds (UCB) and lower confidence bounds (LCB) with a doubling time horizon for exploration.
To construct the UCB and LCB, a Nash confidence bound is utilized, which contains the mean reward value. Larger mean rewards correspond to larger confidence intervals, as defined in equation (8).

The theoretical analysis demonstrates that for a finite number of arms, the proposed algorithm achieves a nash regret of approximately $\tilde{O}(\sqrt{d/T})$. In the case of an infinite number of arms, the regret is $\tilde{O}(d^{5/4}/\sqrt{T})$ with a modified confidence width.

**Strengths:**

- This paper addresses the concept of nash regret in the context of stochastic linear bandits.
- It presents a novel algorithm based on elimination phases and provides a detailed regret analysis.

**Weaknesses:**

- The paper does not discuss the computational aspects of finding the G-optimal distribution and sampling via the John ellipsoid method.
- It does not include experimental results.
- In the case of infinite arms, the achieved regret bound may not be sufficiently tight, as it involves a factor of $d^{5/4}$ instead of just $d$.

**Questions:**

- I'm wondering what is the reason for using an elimination-based algorithm rather than UCB.
- wondering what is the reason to consider sub Poisson distribution rather than sub gaussian.
- In line 504 (proof of Theorem 1 in appendix), when $\langle x^*,\theta^* \rangle\le \sqrt{d\nu/T}$, what is the reason that the NR bound, which contains $\langle x^\star,\theta^\star \rangle$ in the bound,  trivially holds?

**Limitations:**

- This problem deals with a fixed arms set rather than a contextual linear setting.
- The regret bound of $d^{5/4}$ in the case of infinite arms may not be tight, and it would be better to elaborate on why the factor of $5/4$ appears. ( $\sqrt{d\log|\mathcal{X}|}$ regret bound for the finite case seems to imply $d$ when we consider $|\mathcal{X}|=2^d$ for infinite case.)
- There are no experimental results and discussion on the computational cost, which raises concerns about practical aspects

Minor comments:

- line 441 in Appendix: $d$-> $\sqrt{d}$
- line 449 in Appendix: $\langle x,\hat{\theta}\rangle \le 20$ --> $\langle x,\hat{\theta}\rangle \ge 20$
- Lemma 7, equation (10) in Appendix: $P()\le 4T^{-1}\log T$ -->$P()\ge 1-4T^{-1}\log T$  ?

---

> ### Author Rebuttal · Authors · 2023-08-09
>
> ***The paper does not discuss the computational aspects of finding the G-optimal distribution and sampling via the John ellipsoid method.***
>
> Thank you for this relevant question. Our algorithm (LinNash) is in fact computationally efficient. A detailed runtime analysis is provided in the common response (addressed to all the reviewers).
>
>
> ***It does not include experimental results.***
>
> Kindly see the common response (addressed to all the reviewers); it details experiments supporting our theoretical guarantees.
>
>
> ***I'm wondering what is the reason for using an elimination-based algorithm rather than UCB.***
>
> Note that, in the finite arms setting for linear bandits and even for cumulative regret, theoretical guarantees of UCB/OFUL and Phased Elimination have a gap of $\sqrt{d}$; see Chapter 22 of [Lattimore and Szepesvári, 2022].
>
> In fact, in Chapter 25 of [Lattimore and Szepesvári, 2022] demonstrates the shortcomings of UCB-based algorithms when information is being shared across arms. For these reasons, we opted to focus on an elimination-based algorithm (rather than UCB). Furthermore, note that the results obtained in this work do not make any assumption on the set of arms.
>
> ***wondering what is the reason to consider sub Poisson distribution rather than sub gaussian.***
>
> The main purpose of using sub-Poisson random variables is to exploit tighter multiplicative forms of concentration bounds that are mean dependent. Having said that, many natural random variables (for instance bounded random variables) are jointly sub-Poisson and sub-Gaussian. Kindly see the common response for a detailed discussion on connections between sub-Poisson and sub-Gaussian distributions.
>
> ***In line 504 (proof of Theorem 1 in appendix), when $\dots$***
>
> There is a typo in the Nash regret statement that caused this confusion. The Nash regret guarantees hold for the scaling factor $\max( \{ 1, \langle x^*, \theta^* \rangle \})$, in place of $\langle x^*, \theta^* \rangle$. This fact has been explained in Lines 294-296, but we have missed placing the scaling factor, as such, in the final expression.
>
> We thank the reviewer for the careful reading and will address it in the next version.
>
> Note that if we have the $f = \max (\{ 1, \langle x^*, \theta^* \rangle\})$ within the Nash Regret bound, then L504 is easy to follow: the Nash Regret by definition is less than $ \langle x^*, \theta^* \rangle$ and therefore, when $ \langle x^*, \theta^* \rangle = \widetilde{O}(\sqrt{d\nu/T})$, the Nash regret must also be smaller. Hence, proving the WLOG statement.

---

> > ### Comment · Reviewer_K5vq · 2023-08-11
> > **Thank you for your response**
> >
> > I appreciate your reply. I would like to preserve my score due to its specific focus on sub-poison aspects and the omission of a discussion regarding the dependency on $d$ for infinite arms.

---

> > > ### Author Response · Authors · 2023-08-11
> > > **Apologies for accidentally missing the question. We provide detailed answer below:**
> > >
> > > Our sincere apologies for having accidentally missed the question on the dependency of $d$ for infinite arms. We hope the following answers change the reviewer's mind regarding the score.
> > >
> > > ***The regret bound of $d^{5/4}$ in the case of infinite arms may not be tight***
> > >
> > > Please note that the factor $\log \left|\mathcal{X}\right|$ is not within the square root in Theorem 1. For $\left|\mathcal{X}\right|=2^d$, our first regret bound for finite arms (Theorem 1) of the form $\sqrt{d}\log \left|\mathcal{X}\right|$ results in a Nash regret that has a scaling factor of $d^{3/2}$. Thus Theorem 2 for infinite arms improves upon the dependence of $d$ in Theorem 1 guarantee when  $\left|\mathcal{X}\right|=\omega(2^{d^{3/4}})$. The dependence of $d^{5/4}$ in Theorem 2 is mainly an artifact of our proof techniques that arises from using multiplicative versions of Hoeffding bounds that are crucial to the regret guarantees. We conjecture that this dependence can be removed (as mentioned in Conclusion section) - but this will require a completely novel set of proof techniques. Finally, we emphasize that as stipulated by NeurIPS, we have already acknowledged this limitation in the Conclusion section and Limitations paragraph in Appendix. We will add a more detailed discussion on this in the final manuscript with more available space.
> > >
> > > ***specific focus on sub-poison aspects***
> > >
> > > We urge the reviewer to read the global response. We have provided a detailed equivalence connection between sub-Poisson and sub-Gaussian random variables. In particular, we can prove the following statements (note that positiveness of reward random variables are crucial to the definition of Nash Regret):
> > >
> > > 1) Bounded, positive random variables are both sub-Gaussian and sub-Poisson (also discussed in Lemma 1).
> > >
> > > 2) The half-normal random variable (https://en.wikipedia.org/wiki/Half-normal_distribution), with a variance of $\sigma$, is also a $C\sigma$
> > >  sub-Poisson random variable, where $C$ is a constant independent of distribution parameters.
> > >
> > > 3) For many positive sub-Gaussian random variables, including truncated and folded Normal distributions, the sub-Poisson parameter is small.

---

> > > > ### Author Response · Authors · 2023-08-16
> > > > **Can we expand on the explanations?**
> > > >
> > > > Our apologies again for having missed a question in the original response.
> > > >
> > > > In the above follow-up comment, we have provided a detailed response on both 1) dependence of $d$ in the case of infinite arms 2) connections between sub-Poisson and sub-Gaussian random variables.
> > > >
> > > > Kindly, please let us know if we can expand on the explanations or if there is something that we can clarify further in our response.

---

> > > > > ### Comment · Reviewer_K5vq · 2023-08-17
> > > > > **Thank you**
> > > > >
> > > > > Thank you for your response.
> > > > > According to your global rebuttal, sub-gaussian X is $\sigma^2/\mu$ sub poisson. However, $\mu$ is latent, so I'm wondering how the proposed algorithm can handle the sub-gaussian distribution.

---

> > > > > > ### Author Response · Authors · 2023-08-17
> > > > > > **Response**
> > > > > >
> > > > > > We thank the reviewer for the nice follow-up question. We provide our answer below:
> > > > > >
> > > > > > In the well-studied case of sub-Gaussians, one needs an upper bound on the sub-Gaussian norm. Along the same lines, our algorithms require only an upper bound on the sub-Poisson norm.  (The sufficiency of an upper bound follows from the direct observation that, for any $\nu < \nu’$, a $\nu$ sub-Poisson random variable is also $\nu’$ sub-Poisson.)
> > > > > >
> > > > > > **Therefore, an upper bound on the stated ratio suffices.**
> > > > > >
> > > > > > We reiterate that, with relevant upper bounds in hand, our results can be used **as is** for the following well-known variants of the Gaussian random variable: half-normal random variables, positively truncated Gaussians, and rectified Gaussians.
> > > > > > Note that under these variants, and as required, the resulting random variables (rewards) are positive.

---

> > > > > > > ### Comment · Reviewer_K5vq · 2023-08-18
> > > > > > > **Thank you**
> > > > > > >
> > > > > > > Thank you for your response. When $\mu$ is small enough,  it displays an unexpectedly large regret, which is unusual in bandits with sub guassian distribution. How could we interpret this?

---

> > > > > > > > ### Author Response · Authors · 2023-08-18
> > > > > > > > **Thank you for the question**
> > > > > > > >
> > > > > > > > We thank the reviewer for the question.
> > > > > > > >
> > > > > > > > A small $\mu$ doesn't necessarily imply a large sub-Poisson parameter. It is relevant to note that, a positive sub-Gaussian random variable with small mean will tend to have small variance (and, hence, a small sub-Poisson parameter). Though, there can be exceptions.
> > > > > > > >
> > > > > > > >  For **all** positive sub-Gaussian random variables, and with an additional factor of $\log^{¼} T$, a Nash regret bound can be obtained as follows:
> > > > > > > >
> > > > > > > > Consider a positive random variable that we observe (on pulling an arm at a certain round) and suppose it has a sub-Gaussian norm (variance proxy) of $\sigma^2$. In that case, by standard concentration bound, with probability at least $1-o(T^{-2})$, the random variable will be bounded from above by $4\sigma\sqrt{\log T}$. Conditioned on this high probability event, the random variable is a $4\sigma\sqrt{\log T}$ sub-Poisson random variable (Lemma 1 in paper for bounded random variables).  Then, a union bound leads to boundedness for all observed rewards with high probability. Our entire analysis goes through, and this idea only introduces a additional multiplicative $\log^{¼} T$ factor in the regret bound.
> > > > > > > >
> > > > > > > >
> > > > > > > >
> > > > > > > > Given the interest in sub-Gaussians, we will include this observation in the updated version.
> > > > > > > >
> > > > > > > >
> > > > > > > >
> > > > > > > > However, we reiterate that for natural classes of positive random variables have the same sub-Gaussian and sub-Poisson parameters modulo constants. For these random variables, the additional multiplicative log factor does not even come up.

---

> > > > > > > > > ### Comment · Reviewer_K5vq · 2023-08-19
> > > > > > > > > **Thank you for your response**
> > > > > > > > >
> > > > > > > > > Thank you for your response.
> > > > > > > > > Small mean in subguassian seems to have large variance according to your rebuttal since $\sigma^2/\mu$ sub poisson. Also, I have a concern about whether considering the event of bounded random variables influences the mean of the positive sub-guassian, which makes it difficult to analyze regret.

---

> > > > > > > > > > ### Author Response · Authors · 2023-08-19
> > > > > > > > > > **Thank you for the follow-up**
> > > > > > > > > >
> > > > > > > > > > We understand the reviewer’s interest in sub-Gaussian random variables and provide detailed answers to the raised questions below.
> > > > > > > > > > However, we also feel that this discussion might be taking away the focus from our contributions. The paper is aimed at establishing regret for sub-Poisson rewards. Furthermore, in and of itself, sub-Poisson is an encompassing and important class. Of course, as future work, one can consider extensions to other distribution classes, such as sub-Gaussian, sub-exponential, and heavy tailed.
> > > > > > > > > >
> > > > > > > > > > Here are the responses to the raised questions:
> > > > > > > > > >
> > > > > > > > > > 1)	Small mean: We reiterate that the sub-Poisson parameter is a ratio of the variance and the mean ($\sigma^2/\mu$) of a positive sub-Gaussian random variable. Hence, the reviewer’s instinct that a small mean will invariably drive up the sub-Poisson parameter is not universally correct.
> > > > > > > > > > The ratio can continue to be small even with a small denominator. This follows from the intuition that (specifically for **positive sub-Gaussian** random variables) a small mean also suggests a small variance. This intuition is clearly validated in the case of well-studied modifications of standard Gaussians (truncated, restricted, folded Gaussians), which we have pointed out earlier.
> > > > > > > > > >
> > > > > > > > > > 2)	Mean Shift: Yes, there will be a mean shift after conditioning. Though, the entire proof goes through for identical sub-Gaussian noise. This follows from the fact that the shift, say $\delta$, is identical across all arm pulls. That is, if an arm $x$ initially has mean $<x, \theta^*>$, then it will now have mean $<x, \theta^*> + \delta$. This updated setting can be viewed as yet another linear bandit instance, though now in dimension $d+1$: each arm $x$ is appended with a component equal to one, and $\theta^*$ is appended with a component equal to $\delta$. Everything else follows as in the paper.

---

> > > > > > > > > > > ### Comment · Reviewer_K5vq · 2023-08-20
> > > > > > > > > > > **Thank you for your response.**
> > > > > > > > > > >
> > > > > > > > > > > I appreciate your explanation. I understand that the sub-Poisson assumption is versatile enough to encompass various scenarios, including specific sub-Gaussian distributions as you pointed out. (I'm personally interested in understanding how to handle the value of δ, given that an assumption is needed for bounding the norm of θ*.)
> > > > > > > > > > >
> > > > > > > > > > > In addition to the noise distribution assumption, I also have a concern regarding the inherent constraint of the linear model, where there is a predetermined set of arms with 𝒳, as opposed to a contextual linear model.
> > > > > > > > > > >
> > > > > > > > > > > Now that my part of initial concerns have been addressed, I've adjusted my rating to 6.

---

> > > > > > > > > > > > ### Author Response · Authors · 2023-08-21
> > > > > > > > > > > > **Thank you.**
> > > > > > > > > > > >
> > > > > > > > > > > > We thank the reviewer for updating the score.
> > > > > > > > > > > >
> > > > > > > > > > > > A quick note: $\delta$ will be a small ($o\left(\frac{\sigma}{T} \right)$). Hence the norm of the new $\theta^*$ will be, at most, the initial one plus a small additive term.
> > > > > > > > > > > >
> > > > > > > > > > > > For contextual bandits, kindly see the reply to reviewer Cioh.

---

### Official Review · Reviewer_zbVi · 2023-07-04

**Soundness:** 3 good
**Presentation:** 3 good
**Contribution:** 3 good
**Rating:** 6
**Confidence:** 3

**Summary:**

The authors consider stochastic linear bandits, and the goal is to minimize Nash regret, which is a stronger notion than the standard regret, by replacing the arithmetic mean reward of the learner over time with the geometric one in the definition.

**Strengths:**

The paper is well written. I didn't check the proof but I didn't spot any critical errors in the main text, either. The setting, the algorithms, and the proof sketch all make sense to me. I agree that indeed in some scenarios we may need to consider fairness across rounds and the definition of Nash regret in this paper is probably a good notion to consider in this case.

**Weaknesses:**

This setting is rarely known. It doesn't mean this problem is unimportant, but I think it would be better to provide more context for it to better evaluate the contributions. In the related work, the authors only mention that this setting differs from prior works because they all consider fairness across arms while this paper focuses on fairness across rounds. However, I would expect more discussions about this like whether previous techniques can be applied here. Also, there is no discussion about the Nash regret of multi-armed bandits, which is an important special case. Although this setup may not be meaningful in practice, it may provide insights into handling the linear case when designing algorithms or proving regret bounds.

**Questions:**

- The current algorithm requires $T$ as input. Do the authors think this is necessary? Part I in Algorithm 2 is somewhat non-standard for a successive elimination algorithm, so I guess it might be difficult to apply a doubling trick to remove the prior knowledge of $T$.
- Does the sub-Poisson assumption include Gaussian as a special case? If not, can the analysis go through under sub-Gaussian?

---

> ### Author Rebuttal · Authors · 2023-08-09
>
> ***no discussion about the Nash regret of multi-armed bandits, which is an important special case.***
>
>  Perhaps the reviewer missed this, but Nash Regret in the context of stochastic multi-armed bandits (MAB) has already been addressed in [4]. The writeup extensively cites this prior work (see Lines 69-70, 89, 111, 180, 254). We have also discussed why the techniques for MAB in [4] fail in the context of linear bandits (see Sections 3.1 and 3.2 for a detailed discussion).
>
> ***whether previous techniques can be applied here***
>
> Previous techniques for analysing fairness across arms (for instance in [18]) are completely unrelated and are based more on combinatorial tools. To support this point, we will add a discussion in the next version of the manuscript.
>
> ***The current algorithm requires T as input. Do the authors think this is necessary? Part I in Algorithm 2 is somewhat non-standard for a successive elimination algorithm, so I guess it might be difficult to apply a doubling trick to remove the prior knowledge of T.***
>
> Indeed, Part I requires the knowledge of T and resists the use of the doubling trick. Developing an anytime algorithm for Nash regret in the linear bandits context is an interesting problem and requires a different set of ideas.
>
> ***Does the sub-Poisson assumption include Gaussian as a special case? If not, can the analysis go through under sub-Gaussian?***
>
> Kindly see the common response for a detailed discussion on connections between sub-Poisson and sub-Gaussian distributions.

---

> > ### Comment · Reviewer_zbVi · 2023-08-18
> >
> > I thank the authors for their response. It's my bad not noticing [4] as a highly related work. However, the algorithm looks similar to the one in [4]. It would be good to discuss the common idea better and highlight the key difference and challenges. Overall my concerns were addressed and I increased my score.

---

> > > ### Author Response · Authors · 2023-08-19
> > > **Thank you for your response**
> > >
> > > We appreciate the reviewer's feedback. We'll include a brief comparison with the algorithm in [4] in the updated version.

---

### Official Review · Reviewer_Cioh · 2023-07-04

**Soundness:** 4 excellent
**Presentation:** 4 excellent
**Contribution:** 3 good
**Rating:** 7
**Confidence:** 3

**Summary:**

The paper extends algorithmic techniques for the linear bandits problem to the Nash Social Welfare objective function which ensures a level of fairness rather than merely the utilitarian welfare. Their work provides essentially optimal regret guarantees as compared to the traditional linear bandits problem (as it is well known that maximizing the geometric mean is necessarily stronger than the arithmetic mean). These results work for both a finite and infinite set of arms with slightly different regret guarantees.

**Strengths:**

The paper takes a novel approach to the linear bandits problem by adapting the objective function to one that ensures some level of fairness across agents in the given system. Moreover, all results appear sound and presented in a nice and compact form within the main text -- the sketches and intuition are clearly depicted with full proofs (not all of which were verified by this reviewer) in the appendix. In addition to near optimal regret guarantees on this modified problem instance, the authors provide various tail bounds that are of independent research to the community.

**Weaknesses:**

The paper lacks a comprehensive discussion on the related work surrounding this problem. Though the problem at hand is novel, it does hurt the submission to not include such context for readers. Specifically, there is considerable work on online NSW maximization which is naturally a similar problem. Moreover, discussion of the linear bandits problem and the best known results there would be very useful (in addition to the brief mention of Nash regret being a stronger guarantee). Some discussion on these results for finite and infinite arms would also potentially alleviate the issue of the gap between upper and lower bounds in the latter setting. As noted by the paper, improving the guarantee for the infinite arm problem setting would naturally strengthen the submission considerably.

**Questions:**

Can you provide a brief sketch of the final theorem (infinite arms) regret guarantee? I understand that the submissions page limit prevents this but it would be helpful to have one in the extended version of the paper.

Have you considered the regret guarantees of the contextual bandits problem with a NSW objective?

Have you considered the multiplicative regret on this multiplicative objective function as well?

---

> ### Author Rebuttal · Authors · 2023-08-09
>
> ***The paper lacks a comprehensive discussion on the related work surrounding this problem.***
>
> As suggested in the review, we will add some discussion on online resource allocation (including online algorithms for NSW maximization).
>
>
> ***Can you provide a brief sketch of the final theorem (infinite arms) regret guarantee? I understand that the submissions page limit prevents this but it would be helpful to have one in the extended version of the paper.***
>
> With more available space in the final version, we will include a proof sketch for Theorem 2. Here is an outline:
> ​​
> To analyse the infinite-arms case, we bound the norm difference between the computed least squares estimator, and the true (unknown) parameter vector. In particular, using the selection criterion for the set of surviving arms (see equation 11 in the paper) and an $\epsilon$-net argument, we bound the error of the OLS estimate in terms of the reward of the best arm. The main technical novelty in this case is the $\epsilon$-net argument which needs to be done carefully for this setting. Furthermore, since we sample following the D-optimal design, the confidence widths in the reward estimates of the surviving arms reduces exponentially in every phase. Combining these, for all surviving arms, we can bound the difference in reward with the best arm at every phase and, hence, bound the Nash regret overall. This completes the proof sketch.
>
>
>
>
> ***Have you considered the regret guarantees of the contextual bandits problem with a NSW objective?***
>
> The extension to contextual bandits requires independent ideas. Here, it is worth considering the recent result of Hanna, Yang, and Fragouli (COLT’23) that reduces linear, contextual bandits to standard linear bandits.  Note that, even for cumulative regret, contextual bandits (in the linear setting with changing arm sets) requires involved ideas;  see, e.g., the SupLinUCB paper by Chu et al. (2011). As mentioned in Section 5, we consider contextual bandits as an interesting direction of future work.
>
>
> ***Have you considered the multiplicative regret on this multiplicative objective function as well?***
>
> This is an interesting question. We have considered the multiplicative regret. However, the techniques and results are orthogonal to the current, standard treatment of regret.

---

> > ### Comment · Reviewer_Cioh · 2023-08-16
> >
> > Thanks for the response, I have no further questions at this time and will keep my 7 score.

---

### Official Review · Reviewer_XR4M · 2023-07-11

**Soundness:** 3 good
**Presentation:** 2 fair
**Contribution:** 2 fair
**Rating:** 6
**Confidence:** 3

**Summary:**

This paper considers the problem minimize 'Nash regret' in linear bandit, where Nash regret here is defined as the difference between the optimum and the geometric mean of expected rewards induced by the bandit algorithm. The stochastic reward in this paper is assumed sub-Poisson. The authors give algorithm to achieve Nash regret of $O(\sqrt{d/T}log(T/|\mathcal{X}|)$ when the set of arms $\mathcal{X}\in \mathbb{R}^d$ is finite and $O(\frac{d^{5/4}}{\sqrt{T}}logT)$ when the set of arms $\mathcal{X}$ is infinite.

**Strengths:**

- This paper exhibits a good technical foundation. Specifically, in the finite-arm case, the upper bound is tightly bounded, even considering logarithmic terms.

- The notion of regret explored in this paper is novel and is stronger compared to external regret.

**Weaknesses:**

The paper lacks experimental results. It would be benefical to compare other bandit algorithm with the algorithm in the paper specifically in terms of this kind of Nash regret.

**Questions:**

- The term "Nash regret" has been previously used in [1] and [2], but with a different definition from the one presented in this paper. The current name "Nash regret" in this paper does not capture the characteristic of the geometric mean.

- Regarding the definition of Nash regret in this paper, the expectation is taken within the geometric mean instead of first taking the geometric mean and then the expectation. Is there any specific reasons or motivations behind the choice of this order?

- In equation (3), the solution obtained provides the optimal sequence of arm pulls. What is the meaning of 'optimal' here? Further explanation and intuition behind equation (3) can be provided by the authors to better understand the optimality criteria and its implications.

- In line 439 in the appendix, there's a wrong ref (Algorithm ??)

[1] Cui, Qiwen, et al. "Learning in congestion games with bandit feedback." Advances in Neural Information Processing Systems 35 (2022): 11009-11022.

[2] Tay, Sebastian Shenghong, et al. "No-regret Sample-efficient Bayesian Optimization for Finding Nash Equilibria with Unknown Utilities." International Conference on Artificial Intelligence and Statistics. PMLR, 2023.

**Limitations:**

The authors have partially addressed the limitations of their work, though there is space for improvement (see the section Weaknesses and Questions).

---

> ### Author Rebuttal · Authors · 2023-08-09
>
> ***The term "Nash regret" has been previously used in [1] and [2], but with a different definition from the one presented in this paper. The current name "Nash regret" in this paper does not capture the characteristic of the geometric mean.***
>
> The papers [1] and [2] consider Nash equilibrium and, hence, refer to “Nash” in their regret analysis. In contrast, the current work builds upon the notion of Nash social welfare (also studied by John Nash). Nash social welfare is defined as the geometric mean of agents’ valuations.
>
> ***Regarding the definition of Nash regret in this paper, the expectation is taken within the geometric mean instead of first taking the geometric mean and then the expectation. Is there any specific reasons or motivations behind the choice of this order?***
>
> Kindly see Lines 108 to 114 for a full discussion on this very point.
> To reiterate, taking the geometric mean of expectations corresponds to an ex ante evaluation of welfare: For each of the T agents (arriving one per round), we consider the expected reward (i.e., ex-ante utility) and then compute the welfare (apply geometric mean). Such ex ante assessments are standard in mathematical economics.
>
> On the other hand, Jensen’s inequality (in the multivariate form) implies that the variant wherein one considers expected value of the geometric is a more challenging objective. However, as discussed in [4], it is not possible to obtain non-trivial guarantees for the variant in general.
>
>
> ***In equation (3), the solution obtained provides the optimal sequence of arm pulls. What is the meaning of 'optimal' here? Further explanation and intuition behind equation (3) can be provided by the authors to better understand the optimality criteria and its implications.***
>
> Thank you for pointing out the need for further details. A thorough and mathematically rigorous answer to the question can be found in Chapter 21 of [Lattimore and Szepesvári, 2020].
>
> At a high level, given a budget B on the number of arm pulls, there are many possible sequences of B arms that can be selected. Here, an fundamental objective is to identify a sequence of B arms pulls such that the resulting least squares estimate $\widehat{\theta}$ (computed from the noisy rewards) minimizes the maximum confidence width in the estimated mean reward over the arms.
>
> The optimality is in this exact sense. We will clarify this point further in updated version.
>
> ***In line 439 in the appendix, there's a wrong ref***
>
> Thank you for pointing the typo. We will fix it.

---

> > ### Comment · Area_Chair_69Ca · 2023-08-21
> >
> > Dear Reviewer XR4M,
> >
> > Could you confirm that you have read the authors' rebuttal, and whether your assessment has any change based on the rebuttal?
> >
> > Many thanks.
> >
> > Area Chair

---

> > ### Comment · Reviewer_XR4M · 2023-08-21
> >
> > Thank you for the response. The authors have addressed my concerns. So I will increase my score to 6.

---

### Official Review · Reviewer_4BdU · 2023-07-27

**Soundness:** 3 good
**Presentation:** 2 fair
**Contribution:** 2 fair
**Rating:** 6
**Confidence:** 1

**Summary:**

This paper proposes an approach based on the Nash regret to extend the stochastic linear bandits framework with fairness and welfare considerations. The Nash regret relies on the geometric mean of expected rewards induced by the linear bandit algorithm, which tends to incentivize uniformly high rewards as opposed to the average regret which promotes cumulatively high rewards. In this setting, the authors derived tight upper bounds on this strengthened notion of regret, thereby providing fairness guarantees as the geometric mean of rewards corresponds to the Nash social welfare (NSW) function.

Building upon a novel sampling arms strategy based on John ellipsoid and optimal design, as well as tailored concentration bounds, the authors propose an algorithm that achieves $O\left(\sqrt{\frac{d\nu}{T}}log(T|\chi|)\right)$, for a finite set of arms $\chi$, and $O\left(\frac{d^{5/4}\nu^{1/2}}{\sqrt{T}} log(T)\right)$, in the infinite case.

**Strengths:**

The paper is technically solid and well-motivated. I really like the idea of inherently imposing fairness to the bandit algorithm via the Nash regret. To the best of my knowledge, the authors' result is new and provides valuable insight into incorporating fairness and welfare considerations into bandit algorithms. In addition, the authors propose tail bounds that may be of broader interest.

**Weaknesses:**

Although the paper is technically well-set, the contribution seems incremental. The paper is also suffering from a lack of comparisons with existing approaches. The related work section seems a bit short. Having a richer picture of the literature would be beneficial for understanding the context of the paper’s contribution.

Lastly, I feel that the paper could benefit from numerical experiments that support the derived theoretical results.

**Questions:**

What is the computational efficiency of the LinNash algorithm?

**Limitations:**

The limitations of the proposed framework are not entirely clear.

---

> ### Author Rebuttal · Authors · 2023-08-09
>
> ***Although the paper is technically well-set, the contribution seems incremental.***
>
> The work brings forth (1) important conceptual ideas and (2) entails substantive technical steps. We do not find these results to be incremental and believe the current paper will inspire future research.
>
> 1) On the conceptual front, the current work is the first to address Nash regret in the important context of linear bandits. Nash regret systematically incorporates Nash social welfare, which, in turn, upholds fairness axioms, such as Pigou-Dalton transfer principle, scale invariance, and independence of unconcerned agents. Indeed, this principled approach of addressing fairness in bandit algorithms avoids ad hoc criteria and adjustments.
>
> 2) On the technical front, the paper develops multiple, novel ideas for bounding the strengthened notion of regret; this includes, (i) sampling from a distribution whose mean is the centre of the John Ellipsoid induced by the arm vectors and (ii) tighter multiplicative forms of concentration bounds that are mean dependent using properties of sub-Poisson random variables. As mentioned in the writeup, these techniques will likely be of independent interest.
>
>
> ***The paper could benefit from numerical experiments that support the derived theoretical results.***
>
> Kindly see the common response (addressed to all the reviewers); it details experiments supporting our theoretical guarantees.
>
>
> ***What is the computational efficiency of the LinNash algorithm?***
>
> Thank you for this relevant question. Our algorithm (LinNash) is in fact computationally efficient. A detailed runtime analysis is provided in the common response (addressed to all the reviewers).

---

> > ### Comment · Area_Chair_69Ca · 2023-08-21
> >
> > Dear Reviewer 4BdU,
> >
> > Could you confirm that you have read the authors' rebuttal, and whether your assessment has any change based on the rebuttal?
> >
> > Many thanks.
> >
> > Area Chair

---

> > > ### Comment · Reviewer_4BdU · 2023-08-22
> > >
> > > Thank you for your response. My concerns and questions have been addressed. I have increased my score to 6.

---

### Author Rebuttal · Authors · 2023-08-09

***Numerical Simulations***

Some reviewers have asked about numerical simulations. In the limited rebuttal time period, we have compared our algorithm LinNash with Thomson Sampling on synthetic data. Note that we chose Thomson Sampling (Algorithm 1 in [1]) instead of UCB/OFUL because it is crucial in our setting to use a randomized algorithm in order to achieve non-trivial Nash Regret guarantees.

We tune parameters of both algorithms to the best of our ability and compare them in the following setting - we set the ambient dimension to be 80, number of arms to be 10000 and number of rounds to be 50000. Both the unknown parameter vector and the arm embeddings are drawn from a multi-variate gaussian. The arm embeddings are then shifted and normalized so that all true rewards are non-negative and the maximum true reward is 0.5. On pulling an arm, we observe a bernoulli random variable with probability corresponding to its true reward.

In this setting, we observe that LinNash outperforms Thomson Sampling by a significant difference. Another advantage of LinNash that is apparent from this experiment is successive elimination - the variance in the quality of arms pulled reduces with rounds. This is because sub-optimal arms are being eliminated in bulk at regular intervals.
In contrast, Thomson Sampling incurs a large variance in quality of arms being pulled even after several rounds - since no arms are being eliminated at any point.

[1] Shipra Agrawal and Navin Goyal. Thompson Sampling for Contextual Bandits with Linear Payoff (2014).


***Computational efficiency of the LinNash algorithm***

Some reviewers have asked this relevant question. Our algorithm (LinNash) is in fact computationally efficient. Here, we provide a detailed breakdown of the runtime analysis.

Time Complexity of the developed algorithm: Note that LinNash calls the subroutine GenerateArmSequence in Part 1 (that involves computing the John Ellipsoid). Also, the algorithm solves the optimal design problem once in Part 1 and at most O(log T) times in Part 2.

We note that each of these modules execute in time polynomial in the size of the arm set.
Optimal design - In Lines 150-163, we have described that the G-optimal design problem (eq. (3)) and the D-optimal design problem (eq. (4)) share the same optimal solution. The D-optimal design problem is a concave function and the Frank-Wolfe algorithm with rank-1 updates can be used to find the optimal solution efficiently (see Note 3 in Chapter 21 of [Lattimore and Szepesvári, 2022]). More specifically, as discussed in Chapter 3 of [Todd, 2016], the runtime of each iteration is $O(|\mathcal{X}|^2)$ and the number of iterations is at most $O(d)$. Here $|\mathcal{X}|$ is the number of arms and $d$ is the ambient dimension of the arm vectors.
Sampling via John Ellipsoid - Computing the John Ellipsoid given a set of arm vectors is equivalent to the D-optimal design problem (c.f., [Todd 2016]). Furthermore, approximate versions of John Ellipsoids (which also suffices for our purposes) can be computed much faster (see, e.g., [Cohen et al., 2020]) with a runtime of $O(|\mathcal{X}| d^2)$.

In summary, for each round , the runtime of our algorithm is quadratic in the number of arms and the ambient dimension.

References:
Minimum Volume Ellipsoids (Theory and Algorithms) by Michael J. Todd.
A near-optimal algorithm for approximating the John Ellipsoid by Michael B. Cohen, Ben Cousins, Yin Tat Lee, Xin Yang (2020).


***Connection between sub-Poisson and sub-Gaussian random variables***


We will focus exclusively on positive random variables so that the Nash Regret is well defined. To provide a general connection between sub-Poisson and sub-Gaussian random variables, we can show the following Lemma:

Lemma: Any positive $\sigma$ sub-Gaussian random variable, with mean $\mu$ is also  $\frac{\sigma^2}{\mu}$ sub-Poisson.

Before going to the proof, we make the following noteworthy observations using the above lemma:
Bounded, positive random variables are both sub-Gaussian and sub-Poisson (also discussed in Lemma 1).
The half-normal random variable (https://en.wikipedia.org/wiki/Half-normal_distribution), with a variance of $\sigma$, is also a C$\sigma$ sub-Poisson random variable, where C is a constant independent of distribution parameters.
For many positive sub-Gaussian random variables, including truncated and folded Normal distributions, the sub-Poisson parameter is small.

We leave a more in-depth analysis into the connections as future work. For the interested reader, we provide a proof of the above lemma below:

Proof: Let $X$ be any positive $\sigma$ sub-Gaussian random variable. For any positive $s$ and any $K \geq \frac{\sigma}{\mu}$, we have
 $ E[e^{sX}] \leq \texttt{exp} \left( s\mu + \frac{(s \sigma)^2 }{2}\right) \leq \texttt{exp} \left(\frac{\mu}{ K \sigma } \left(K s \sigma + \frac{(K s \sigma^2 )}{2} \frac{\sigma}{\mu K}\right)\right)$.

Now, since $K \geq  \frac{\sigma}{\mu}$ and $e^x \geq 1+ x+ \frac{x^2}{2}$ for non-negative $x$, we obtain $ E[e^{sX}] \leq \texttt{exp} \left(\frac{\mu}{ K \sigma } \left(e^{K s \sigma } - 1 \right)\right)$.
For negative values of $s$, there exists a bounded random variable $Y$ which takes value in $[0, K \sigma]$ such that $E[e^{sX}] \leq E[e^{sY}]$. Now using Lemma 1 for bounded random variables we obtain $ E[e^{sX}] \le \texttt{exp} \left(\frac{\mu}{ K \sigma } \left(e^{K s \sigma } - 1 \right)\right)$. Finally, substituting $K = \frac{\sigma}{ \mu }$ proves that $X$ is $\frac{\sigma^2}{\mu}$ sub-Poisson.

---

> ### Author Response · Authors · 2023-08-15
> **Any further clarifications**
>
> Dear Reviewers,
>
> Thank you for your insightful reviews. Do let us know if further clarifications are required.

---

### Decision · Program_Chairs · 2023-09-21

**Decision:**

Accept (poster)

**Comment:**

The submission is a study on a stronger regret form of Nash regret for linear bandits that has fairness implications. The authors design tailored techniques to achieve good regret bounds for this case. All reviewers are positive towards the paper (with scores 7, 6, 6, 6, 6). The main issues raised by the reviewers, such as experimental evaluations and computational efficiency, have been addressed by the authors' rebuttal, and I am confident that they will be incorporate properly into the revised version of the paper. The reviewers also seemed to be satisfied by the rebuttals and some increases their scores as the result. I would like to recommend acceptance to the paper, and encourage the authors to incorporate all constructive comments from the reviewers into the final version of the paper.